# The Whole Antarctic Ocean Model (WAOM v1.0): Development and Evaluation

Ole Richter[1,2], David E. Gwyther[1,3], Benjamin K. Galton-Fenzi[4], and Kaitlin A. Naughten[5]

[1]Institute for Marine and Antarctic Studies, University of Tasmania, Private Bag 129, Hobart, TAS, 7001, Australia.
[2]Geography & Spatial Sciences, School of Technology, Environments and Design, University of Tasmania, Hobart, TAS, 7001, Australia.
[3]Coastal and Regional Oceanography Laboratory, School of Mathematics and Statistics, University of New South Wales, Sydney, NSW, 2052, Australia.
[4]Australian Antarctic Division, Kingston, TAS, 7050, Australia.
[5]British Antarctic Survey, High Cross, Madingley Road, Cambridge, CB3 0ET, United Kingdom.

**Correspondence:** Ole Richter (ole.richter@utas.edu.au)

**Abstract.** The Regional Ocean Modeling System (ROMS), including an ice shelf component, has been applied on a circum-Antarctic domain to derive estimates of ice shelf basal melting. Significant improvements made compared to previous models of this scale are the inclusion of tides and a horizontal spatial resolution of 2 km, which is sufficient to resolve onshelf heat transport by bathymetric troughs and eddy scale circulation. We run the model with ocean-atmosphere-sea ice conditions from the year 2007, to represent nominal present day climate. We force the ocean surface with buoyancy fluxes derived from sea ice concentration observations and wind stress from ERA-Interim atmospheric reanalysis. Boundary conditions are derived from the ECCO2 ocean state estimate and tides are incorporated as sea surface height and barotropic currents at the open boundary. We evaluate model results using satellite derived estimates of ice shelf melting and established compilations of ocean hydrography. WAOM qualitatively captures the broad scale difference between warm and cold regimes and many of the known characteristics of regional ice-ocean interaction. We identify a cold bias for some warm water ice shelves and a lack of HSSW formation. We conclude that further calibration and development of our approach is justified. At its current state, the model is ideal for addressing specific, process-oriented questions, e.g. related to tide-driven ice shelf melting at large scales.

## 1 Introduction

Modelling of Antarctic ice shelf-ocean interaction is critical to predicting future changes in sea level and climate. Antarctic glaciers drain into floating ice shelves and melting or marine ice accretion at the base of these ice shelves changes their ability to buttress inland ice sheet discharge (e.g. Dupont and Alley, 2005; Gudmundsson, 2013; Pritchard et al., 2012). In turn, glacial melt water impacts the surrounding oceans with consequences for global ocean circulation and climate (e.g. Jacobs, 2004; Purkey and Johnson, 2013). Ocean models that include an ice shelf component are playing a key role in estimating

the current state of ocean-ice shelf interaction (e.g. Galton-Fenzi et al., 2012; Gwyther et al., 2014; Hattermann et al., 2014), understanding the underlying mechanisms of ice shelf melting (e.g Makinson et al., 2011; Hattermann, 2018; Gwyther et al., 2018) and predicting future changes (Kusahara and Hasumi, 2013; Mueller et al., 2018; Naughten et al., 2018a). Within these models, Antarctic-wide applications are of particular interest, as they resolve ice shelf teleconnections (Gwyther et al., 2014; Silvano et al., 2018) and smaller ice shelves with less research focus, all around the continent (Timmermann et al., 2012).

Further, consistent model design and parameter choices in large scale models make it easier to compare different regions, and coupled ice sheet-ocean models for climate predictions will ultimately need Antarctic-wide domains (Asay-Davis et al., 2017). The first realistic, coupled models are now becoming available (Timmermann and Goeller, 2017; Naughten et al., 2021).

The accuracy of circum-Antarctic ocean-ice shelf models, however, suffers from incomplete model dynamics and poorly represented subgrid scale processes. Many ocean models with pan-Antarctic coverage have either been designed with cavities

from the beginning (Beckmann et al., 1999; Timmermann et al., 2002; Hellmer, 2004) or augmented by an ice shelf component at a later stage (e.g. Timmermann et al., 2012; Kusahara and Hasumi, 2013; Dinniman et al., 2015; Schodlok et al., 2016; Mathiot et al., 2017; Naughten et al., 2018b). There also exist stand alone ocean models without explicit or even parameterised ice shelf interaction (e.g. Mazloff et al., 2010) and most earth system models used for state-of-the-art climate projections do not include an ice shelf component (see, e.g., Griffies et al., 2016; Dinniman et al., 2016). Reviews about ocean-ice shelf

modelling are presented by Dinniman et al. (2016) and Asay-Davis et al. (2017). Their results for present day conditions, however, often disagree with available observations and vary widely between models (e.g., see Fig. 6 for estimates of basal mass loss from major ice shelves). Part of these discrepancies originate from boundary conditions and model design (e.g. Dinniman et al., 2015; Naughten et al., 2018b). The integrity of model dynamics, however, is also questionable, since certain physical processes that have been identified as critical in regional studies, have not yet been included in large scale applications

(Dinniman et al., 2016). One of these critical processes is ocean tides, which interact with ice shelves in several ways, most importantly through ice shelf basal melting (Padman et al., 2018). Regional studies have shown that tidal currents can heavily modulate local melt rates (e.g Makinson et al., 2011; Mueller et al., 2012, 2018), but, to our best knowledge, tides have not yet been included in Antarctic-wide ocean-ice shelf models. Further, large scale models are typically run at coarse horizontal resolutions (10-20 km), which are not sufficient to resolve bathymetric troughs and eddies on the continental shelf (Dinniman

et al., 2016). Both of these features, however, have been identified to transport heat from the deep ocean shoreward and not resolving them leads to underestimates of ice shelf melting in some regions (for the importance of troughs see Thoma et al., 2008; Assmann et al., 2013; for eddies see Stewart and Thompson, 2015; Stewart et al., 2018). Resolving tides and eddies is expensive, as they require a fine temporal and spatial discretisation, but including them in large scale models is seen as a major step towards more accurate representations of the polar regions.

Model evaluation and efficient tuning is hindered by sparse in situ observations, both beneath ice shelves and on the continental shelf. Model parameters in regional studies can be calibrated (e.g. Nakayama et al., 2017), but to approach similar efforts with large scale models, suitable Antarctic-wide observations need to be compiled first. Nevertheless, evaluation of selected quantities helps to identify large biases and evaluate model performance. For this purpose, previous studies have utilised ice shelf melt rates derived from satellite observations and models of firn processes (e.g. Schodlok et al., 2016), and selected

Southern Ocean quantities from observations and reanalysis products (e.g. Naughten et al., 2018b). These measures, however, have limitations. For example, while satellite studies provide uncertainty bounds for melt rates averaged over ice shelves or ice flow lines (as in Rignot et al., 2013; Depoorter et al., 2013; Liu et al., 2015), the uncertainty of high resolution data is unknown. Further, compilations of ocean observations, such as the World Ocean Atlas 2018 (WOA18), include most of the available data from ships, Argo Floats, gliders and elephant-seals, but the interpolated temperature and salinity fields are only available as

climatologies with up to decadal resolution and observations in sea-ice covered regions are sparse, implying large uncertainties on the Antarctic continental shelf.

Here we describe the development and evaluation of a new circum-Antarctic ocean-ice shelf model that aims to overcome some of the shortcomings of previous studies. The Whole Antarctic Ocean Model (WAOM v1.0) includes tides and an eddy-resolving horizontal resolution of 2 km, both known to be critical to resolve accurate ice shelf-ocean interaction. We compare

model results against a selection of established estimates of Southern Ocean quantities and ice shelf melting for the chosen period of 2007. This way, we aim to convince the reader that this first version of WAOM is realistic enough to be applied to specific, process oriented studies and to justify further development of our approach.

The following section (Sect. 2) describes the model, the experiments performed in this study and our evaluation strategy. In Section 3, we present tidal accuracy, investigate resolution effects and compare model results against estimates of ice shelf-

ocean interaction from satellite observations and regional studies, as well as selected hydrography from WAO18 and Schmidtko et al. (2014). This is followed by a discussion of WAOM's key strengths, biases and limitations, as well as future development and research questions suitable for exploration with the model at its current state (Sect. 4). The last section (Sect. 5) summarises and concludes this study.

## 2   Model Description

### 2.1   General Approach

The code that is underlying WAOM has been developed over a decade by our research group in Tasmania and established its integrity in the wider community in many regional and idealized applications (Galton-Fenzi et al., 2012; Cougnon et al., 2013; Gwyther et al., 2014, 2016). In this study we use our experience to upscale the code to a circum-Antarctic domain. WAOM v1.0 (Richter, 2020a) and the scripts used for pre- and post-processing (Richter, 2020b) are open source and can be downloaded

and developed on github.

### 2.2   ROMS and Ice-Ocean Thermodynamics

WAOM's backbone is the Regional Ocean Modeling System (ROMS v3.6). ROMS is a free-surface, terrain-following, primitive equations ocean model framework (Shchepetkin and McWilliams, 2005) that allows treatment of advection and diffusion in the ocean in a multitude of ways and on different grid configurations. For WAOM we use a curvilinear coordinate grid (south

polar projection) and solve, for example, horizontal and vertical tracer advection using the 4th-order Akima advection scheme,

while closing turbulent vertical mixing with the scheme from Large et al. (1994) (see Tab. C1 and C2 for all activated options and key parameter choices, respectively).

For ice ocean-thermodynamics, we use the 3-equation melt parameterisation developed by Hellmer and Olbers (1989), refined by Holland and Jenkins (1999) and implemented into ROMS by Galton-Fenzi et al. (2012). The parameterisation accounts
for thermal and haline driving across the ice-ocean boundary layer, velocity dependent exchange coefficients following McPhee (1987) and the case of molecular diffusion alone Gwyther et al. (2016). The exact equations used for ice-ocean interaction in WAOM are described in Gwyther et al. (2016).

## 2.3 Domain, Topography and Spatial Discretisation

The rectangular domain is shown in Figure 1 and covers all of the Antarctic ice shelf cavities and adjacent continental shelf
regions. Spatial discretisation in the vertical uses 31 terrain-following layers with enhanced resolution at top and bottom and results in top layer thicknesses under the ice varying from 0.5 m to 8.3 m (stretching function and parameters used in transformation equations shown at Tab. C2). In the horizontal we apply uniform grid spacing with resolutions of 10 km, 4 km and 2 km, which results in 530x630, 1325x1575 and 2650x3150 horizontal cells, respectively. We note that the design of WAOM requires masking of about 36 % of the cells due to land area.

The ice draft and bottom topography south of 60 °S have been derived from Bedmap2 (Fretwell et al., 2013) and north of 60 °S (outside the Bedmap2 boundaries) have been taken from RTopo-2 (Schaffer et al., 2016). Calculating the horizontal pressure gradient at steep sloping topography in terrain-following coordinates is known to generate spurious currents and mixing (Mellor et al., 1994, 1998). ROMS is designed to minimise this issue by applying the splines density Jacobian method for the calculation of the pressure gradient force (Shchepetkin and McWilliams, 2003). Nevertheless smoothing of bathymetry
and ice draft is recommended (e.g. Sikirić et al., 2009), in particular considering the almost vertical cliff face at the ice shelf front (also discussed in Naughten et al., 2018b). We apply the Mellor-Ezer-Oey algorithm (Mellor et al., 1994), which is well established for bathymetry smoothing (Sikirić et al., 2009). We smooth the bathymetry and water column thickness directly until a maximum slope factor $r = (h_i - h_{i+1})/(h_i + h_{i+1}) \leq 0.3$ is satisfied for both. The ice draft is then redefined as the superposition of bed and water column thickness. This is a well established procedure to minimise spurious currents in regional
ice shelf-ocean configurations (Galton-Fenzi et al., 2012; Cougnon et al., 2013; Gwyther et al., 2014). An experiment at 10 km resolution with uniform stratification and no forcing produced negligible currents in most regions. The only spurious currents of note on or near the continental shelf are along part of the Amundsen-Bellingshausen shelf break, which may explain some of the discrepancy in hydrographic conditions in this region (described in Sec. 3.4). Further, for numerical stability, we artificially deepen the bathymetry in shallow ice shelf grounding zones to obey a minimum water column thickness of 20 m. While this
step might impact local ice shelf ocean-interaction, it has been shown not to affect ice shelf average melt rates and 20 m is considered small within reasonable stability constraints (Schnaase and Timmermann, 2019).

Table 1 summarises the computational costs associated with running the model on the Australian National Computing Infrastructure (NCI) supercomputer Gadi. On the resulting grids with 10 km, 4 km and 2 km resolution the 3-D equations

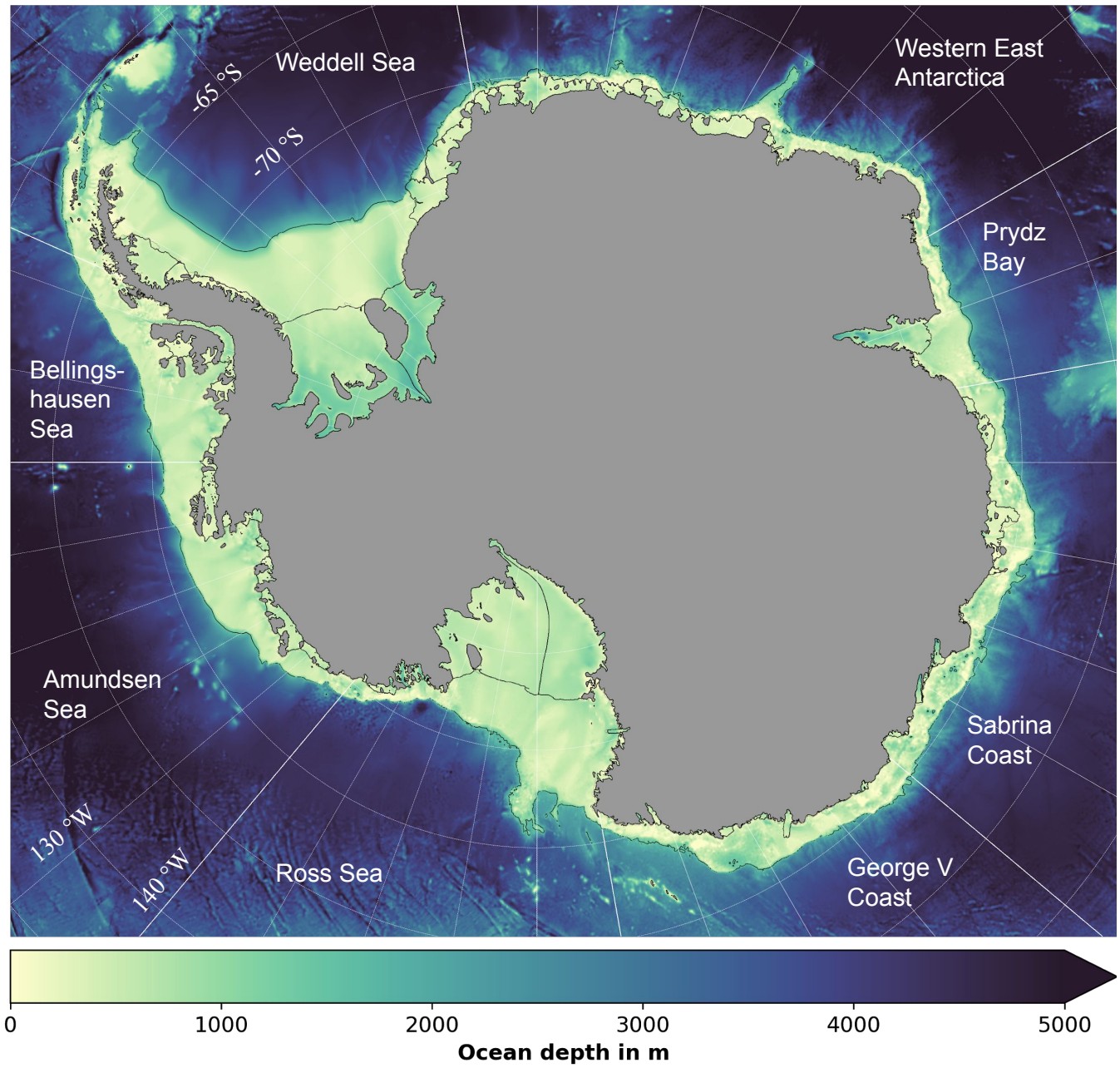

**Figure 1.** Model domain and bathymetry. Figure boundaries denote the model domain and colors show sea floor depth (also inside the sub-ice shelf cavities). Thin black lines are boundaries for the ice shelves and the continental shelf used in this study. Thin white lines are longitudes and latitudes. Labels denote ocean sectors, while bold white lines indicate their boundaries.

integrate stably with timesteps of, respectively, 900 s, 360 s and 180 s. This leads, for example, to a cost of 1,877 CPU hours for 1 year of simulated period at 4 km resolution. We note that initialization and Input/Output require additional resources.

| Model Resolution | 10 km | 4 km | 2 km |
|---|---|---|---|
| Period simulated | 1 year | 1 year | 1 year |
| CPU hours | 93 h | 1,877 h | 16,983 h |
| Number of CPUs | 288 | 2304 | 5184 |
| Memory | 45 GB | 390 GB | 1.53 TB |
| Walltime | 0.5 h | 1.2 h | 4.4 h |
| Storage for 1 3D field | 40 MB | 250 MB | 1 GB |

**Table 1.** Computational requirements at different resolutions. WAOM has been run on the supercomputer Gadi from the National Computing Infrastructure (NCI) in Australia. The architecture consists of 2x24-core Intel Xeon Platinum 8274 (Cascade Lake) 3.2 GHz CPUs per node with 192 GB RAM per node. Listed times are for time-stepping only, that is without initialization or Input/Output.

## 2.4 Forcing and Boundary Conditions

At the surface, we apply daily heat and salt fluxes derived from sea ice concentration observations (Tamura et al., 2011) and daily wind stress calculated from ERA-Interim 10-m winds and bulk flux formula (Dee et al., 2011). Accurate coastal polynyas that form in the lee of fast ice and icebergs are critical to resolve accurate ice shelf melting in cold regimes (see Mode 2 melting described in Jacobs et al., 1992). While flaw leads in the vast pack ice are likely to add more salt into the ocean in total (shown for the Weddell See region, see Haid and Timmermann, 2013), coastal polynyas play a more critical role for regional ice shelf interaction due to their stationary character. Small scale katabatic winds and grounded icebergs play an important role for these polynyas (Kusahara et al., 2010; Mathiot et al., 2010), but both (small-scale winds and ice bergs) are not well represented in current generation sea-ice models. Hence, prescribed surface buoyancy fluxes rather than including a sea ice model, is more likely to capture the position and strength of coastal polynyas. Decoupling sea ice-ocean fluxes from the ocean state is known to create artificial water masses. A similar approach, for example, is known to overestimate heat flux into the ocean by up to 51 % (Jendersie et al., 2018). To reduce such effects, we tune the surface forcing by reducing positive heat flux into the ocean to half its original value, omit brine injection when the ocean is warmer than the freezing point and relax surface temperatures towards freezing when they are being forced below freezing. Further, to avoid model drift, the surface ocean is relaxed to the solution from SOSE (Mazloff et al., 2010), using a surface net heat flux sensitivity to SST of 40 $\mathrm{W\,m^{-2}\,{}^{\circ}C^{-1}}$ and a salinity relaxation timescale of one month. We do not account for the effect of sea ice on wind stress or frazil ice formation (as in, e.g. Galton-Fenzi et al., 2012). While wind stress modulation by sea ice has been shown to play an important role for the circulation and hydrography in the Arctic (Meneghello et al., 2018a, b), its importance for Antarctic ocean-ice shelf interaction has yet to be constrained (as discussed in Jendersie et al., 2018).

Open boundary conditions are taken from the ECCO2 reanalysis (Menemenlis et al., 2008) and consists of monthly values for sea surface height, barotropic and baroclinic velocities, and temperature and salinity. We nudge inflow and outflow with timescales of 1 day and 1 year, respectively. Initial ocean temperatures and salinities for January 2007 are also derived from ECCO2 and values under the ice shelves are extrapolated from the ice front. Thirteen major tidal constituents (M2, S2, N2, K2, K1, O1, P1, Q1, MF, MM, M4, MS4, MN4) are derived from the global tidal solution TPXO7.2 (Egbert and Erofeeva, 2002) and also introduced along the northern boundaries of WAOM using sea surface height and barotropic currents.

## 2.5 Spin Up and Experiments

For this study we simulate the year 2007. Forcing with single year conditions captures daily to seasonal variability, while allowing us to run the model close to quasi-equilibrium with our given supercomputing resources. At the time of development, all data products used to force the model covered the period from 2005 to 2011 and we found that sea ice buoyancy fluxes and wind stress from the year 2007 are a non-anomalous representation of the period from 1992 to 2011.

To further save computational costs we perform most of the spin up at lower horizontal resolutions. This idea takes advantage of the fact that the temporal and spatial scales of ocean processes are correlated, that is the largest spatial features, such as the Weddell Sea gyre, also take the longest time to develop. Figure 2 visualizes our spin-up procedure. The 10 km version of the model is integrated for 5 years, before the on shelf ocean is near to a quasi equilibrium and its solution is used to initialise the 4 km run. Analogously, the 4 km run is stepped forward in time for 2 years before the final 2 km simulation is initiated and integrated for another year and three months. Interannual monthly mean melting at each resolution drifts by less than 3% at the end of the integration period, which we rate as acceptable for the purpose of this study. Interpolation of lower resolution solutions to the higher resolution grids is performed using a nearest neighbour method. This can result in artificially large pressure gradients between neighbouring cells, causing model instability. We address this issue by running the first day of each high resolution simulation with a reduced timestep. The ocean state after one day is then used to initiate the actual high resolution run. Diffusivity and viscosity coefficients have been reduced in proportion to to grid refinement (see C2). The main results are taken from the final year of the 2 km run. The instantaneous drops in melting upon re-initialization is caused by geometrical effects of the grid refinement. The ice shelf area reduces with increasing resolution (e.g. 11 % between 10 km and 4 km), predominantly in ice shelf frontal regions, where melt rates are elevated.

## 2.6 Model Evaluation

In this study we present a tool for the community that can ultimately be used to address many different questions related to ocean-ice shelf interaction. Future studies intending to apply WAOM will need to tune and evaluate the model to their specific needs. In this manuscript we focus on ice shelf basal melting and, hence, have focused our evaluation strategy on this quantity. Also, melt rates contain the integrated history of the upstream ocean and their evaluation implies insights into the hydrography of sub-ice shelf cavities and the adjacent continental shelf. In addition, we directly compare ocean hydrography against observations to provide a first estimate of the biases. This helps to better explain the predicted melt rates and provides a starting point for future studies with different focus.

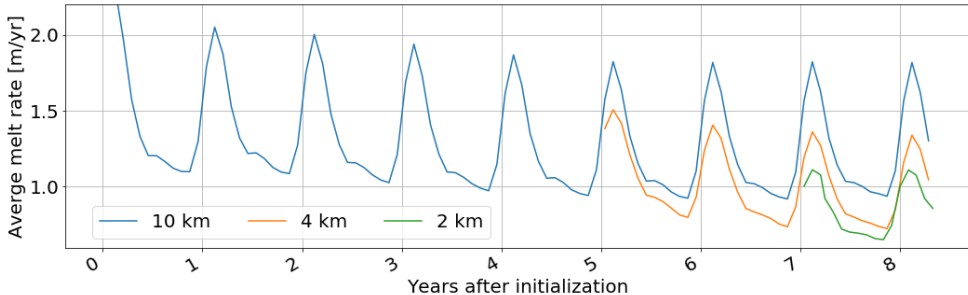

**Figure 2.** Spin up procedure. We spin up large scale processes at lower resolution and find that ice shelf average melting is a good diagnostic for the state of the continental shelf ocean. This way, the total spin up time for the final year of the 2 km-resolution solution is 7 years and 3 months. Model output is plotted as average of every month.

We compare annual mean ice shelf mass loss averaged over individual regions and for total Antarctica against satellite observations from Rignot et al. (2013), Depoorter et al. (2013) and Liu et al. (2015). Uncertainties for satellite derived ice
shelf-ocean interaction at high resolution are unknown (discussed earlier). In this regard, we showcase models results and compare predictions against theory, regional studies and satellite estimates in the text. To calculate basal mass loss from individual ice shelves we use ice shelf boundaries from the MEaSURES Antarctic boundaries dataset (Mouginot et al., 2016). This dataset reflects the 2007 state, while Bedmap2 ice thickness data is mostly based on laser altimetry data from 1994 to 1995. Restricting the ice shelf area to the intersect of Bedmap2 and MEaSURES excludes parts of the ice shelf front in some
regions and a narrow frame of thin ice along the open coastlines (see Fig. B1).

For the ocean evaluation, we have chosen to use WOA18 climatologies and estimates of on-shelf bottom layer hydrography from Schmidtko et al. (2014). WOA18 is most accurate in summer, when sea ice has its minimum extent and the vast majority of observations are taken. The deep ocean is expected to show little seasonality though. Observations on the shelf are sparse and often concentrated along repeated ship tracks (see Fig. D1). The ocean state on the shelf is critical in determining the
circulation and melt rates in the ice shelf cavities. Here, bottom layer hydrography is of particular interest, as it provides information about CDW intrusions and dense water formation. Schmidtko et al. (2014)) provides a comprehensive compilation of on-shelf bottom layer hydrography from CTD measurements. The northern extents of the off-shelf ocean should be seen as a sponge layer, likely affected by ECCO2 boundary and initial conditions and not fully spun-up using our procedere. The flux-forced approach at the surface decouples sea ice conditions from the underlying ocean. This is known to create artificial
water masses in the uppermost layers of the model. Hence, the top 15 m (equivalent to the uppermost 2 sigma layers in most regions) should be seen as a boundary and are excluded from this analysis.

On the shelf (south of the 1500 m isobath), we compare the summer mean (December, January and February) of the WOA18 climatology from 2005 to 2017 against the summer mean of 2007 as predicted by WAOM. We use Temperature-Salinity (TS) diagrams to assess the water masses and longitudinal transects for the stratification. For the TS-diagrams, both products

have been sampled on their original grid. The transects are taken where CTD data underlies the WOA18 product (along ship repeat tracks and on the Amundsen Sea continental shelf, see Fig. D1) and WAOM's estimates have been interpolated to the WOA18 grid (1/4° and up to 102 depth levels) using a nearest neighbour scheme. Further, we compare multidecadal means of bottom layer hydrography from Schmidtko et al. (2014) against the 2007 mean from WAOM. The CTD locations have been interpolated onto the model grid using nearest neighbour interpolation. Then, model data has been interpolated to the depth of the observations using the nearest neighbour scheme. We augment these comparisons by showcasing high resolution transects and regional TS-diagrams that include the cavities from WAOM and compare these results against regional studies in the text. We define the off-shelf ocean as south of 65°N and north of the 1500 m isobath. Here we compare the summer mean of the 2005-2017 climatology from WOA18 against the 2007 summer prediction of WAOM. We also include the prediction of the 2007 summer mean from ECCO2, which provides the initial and boundary conditions for WAOM. Differences in bottom layer hydrography between WOA18, ECCO2 and WAOM are assessed using annual means, as we expect little seasonality at such great depths. All observational estimates have been converted to model quantities (potential temperature and practical salinity).

## 3   Results

### 3.1   Tides Evaluation

Following King and Padman (2005), we assess the accuracy of tides in the model by comparing tidal height signals against 69 Antarctic Tide Gauge (ATG) station data, including observations from tide gauges, gravimetric data and GPS records of ice shelf surface elevation. For this we use 365 days of hourly sea surface elevation model output from the 10 km-horizontal resolution simulation. Evaluating tides at higher resolution would have taken considerably more resources and we expect the improvement of accuracy with finer grid spacing to be incremental. We interpolate the model data to the coordinates of each of the 69 tide gauge stations using nearest neighbour interpolation. For the four major tidal constituents M2, S2, K1 and O1, we recover amplitudes $H$ and phases $G$ from the sea surface height time series using classical tidal harmonic analysis (Pawlowicz et al., 2002), and then calculate the complex amplitude $Z = H(\cos G + i \sin G)$ as a representation of the tidal energy. We disregard stations for a certain constituent if no ATG data is available, the nearest ocean cell is further than 50 km (5 grid cells) away or the tidal harmonic analysis fails to converge. We also disregard 3 stations, which are noted as partially grounded and show non-sinusoidal and complex behaviour (70 Amery IS, 43 Rutford ISTR, 106 Evans ISTR). The Antarctic-wide accuracy of complex amplitudes for each constituent is assessed using root-mean-square (RMS) errors (defined as $\sigma_x$) as follows:

$$\sigma_x = \sqrt{\frac{1}{2N} \sum_{j=1}^{N} \left| Z_j^m - Z_j^o \right|^2}, \tag{1}$$

whereby $m$ and $o$ superscripts denote modelled and observed, respectively, and $N$ is the number of stations. To get a single measure for model bias in tidal energy, the combined RMS error is calculated as

$$\sigma_{comb} = \sqrt{\frac{1}{2N}\sum_{k=1}^{4}\sum_{j=1}^{N}\left|Z_{k,j}^{m} - Z_{k,j}^{o}\right|^{2}}, \tag{2}$$

where the differences are also summed over all four constituents $k$ = [M2,S2,O1,K1].

Table 2 summarizes the outcomes of the tidal height accuracy analysis. The model has a combined RMS error of 20 cm, which is within the accuracy of 2D Antarctic tide models (assessed by King and Padman, 2005). Similar to these models, most of our bias comes from sites at the grounding line deep under the large ice shelves (see Appendix Fig. D4). In these shallow regions, semidiurnal tides reach maximum amplitudes of 3 metres (e.g. Griffiths and Peltier, 2008), while bathymetry and ice
draft are very uncertain. Tidal strength is sensitive to water column thickness and, thus, we attribute most of the tidal bias in WAOM to uncertainties in the sub-ice shelf cavity geometry of Bedmap2. Also, some bias might originate from the imposed 20 m minimum water column thickness in shallow regions (see Sect. 2.3).

Figure 3 shows the relative differences in tidal height amplitude. WAOM systematically overestimates tidal strength of the semi-diurnal constituents in the Ross Sea with differences often exceeding 80%. In contrast, diurnal tides are generally
underestimated and deviations are more balanced around the coast. For the diurnal bands most stations feature differences below 35%.

|  | M2 | S2 | O1 | K1 |
|---|---|---|---|---|
| Number of ATG stations | 98 | 91 | 87 | 79 |
| $\sigma_x$ [m] | 0.14 | 0.11 | 0.06 | 0.08 |
| $\sigma_{comb}$ [m] | | 0.20 | | |

**Table 2.** Summary of tidal height comparison against Antarctic Tide Gauge Records using Root-Mean-Square-Differences (RMSD). RMSD amplitude relative to ATG is also included.

### 3.2 Resolution Effects

The model solution of the continental shelf ocean converges with increasing resolution. We assess the impact of horizontal resolution on the continental shelf ocean by analysing changes in annual mean ocean temperature and average ice shelf melting.
To ensure consistency, we compare the 2 km result against lower resolution solutions with equivalent overall simulation time, that is 365 days after 7 years and three months (the overlap in Fig. 2). The results of the grid convergence study are shown in Figure 4. We find that ocean temperatures as well as melt rates converge when increasing the grid resolution first from 10 km to 4 km (equivalent 150 %) and then to 2 km (equivalent 400 %). We note that several aspects related to model resolution have been changed simultaneously (bathymetry, ice draft topography, horizontal viscosity, horizontal diffusion, the model's ability

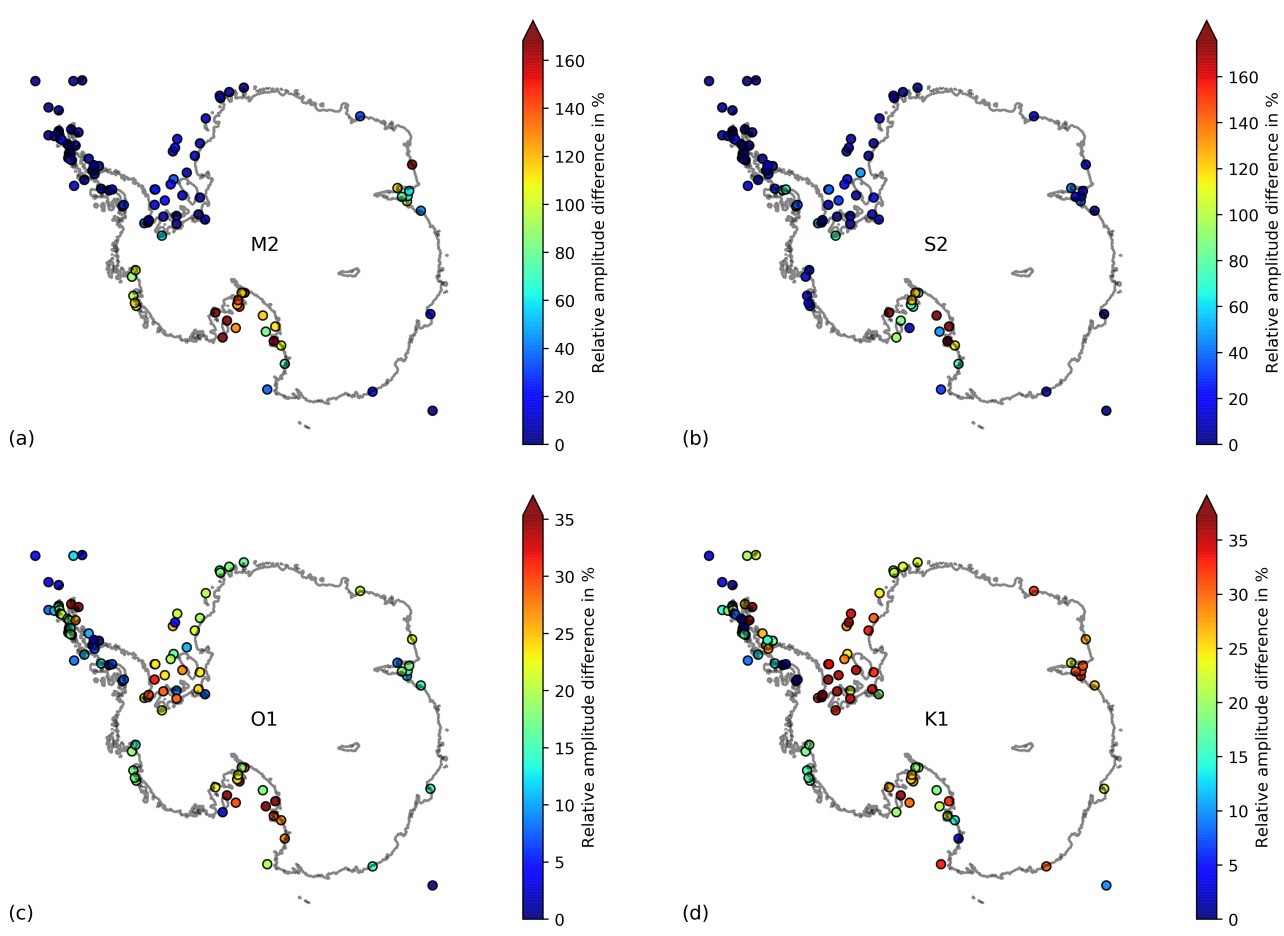

**Figure 3.** Spatial distributions of tidal height accuracy. Relative amplitude differences between the model solution and Antarctic Tide Gauge records ($[H_{WAOM} - H_{ATG}]/H_{ATG}$) are shown for the major tidal constituents (a) M2, (b) S2, (c) K1 and (d) O1. The colorbar has been truncated at the 95% quantile.

to resolve physical processes such as internal tides and eddies). Thus, we use the term "model" in its widest possible sense here, referring to all these aspects together. From 10 km to 1 km, we expect the model solution to be less dependent on resolution, as we start resolving the processes most critical to our problem. Demonstrating convergence of WAOM as a whole is an important first step, proving consistency between our understanding and the models behaviour. Attribution of change to the individual resolution dependent aspects is also important, but out of the scope of this study, as it would require several additional series

of experiments (discussed later). We note that we do not necessarily expect the model to converge towards the observations, without further calibration.

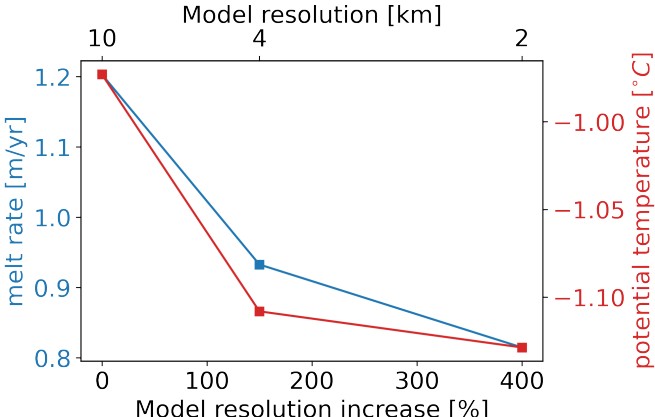

**Figure 4.** Grid convergence. Annual means of average melt rate and continental shelf potential temperature resolved at the different horizontal resolutions: 10 km (0 % increase), 4 km (150 % increase) and 2 km (400 % increase). Continental shelf temperatures have been calculated for depths shallower than 1500 m and including the ice shelf cavities (see Fig. 1). Continental shelf processes converge when grid spacing is refined.

When increasing the grid resolution from 10 km to 4 km, the shelf ocean cools at many places. We find that resolution-induced changes in depth averaged temperature are governed by changes in the bottom sigma layer (not shown). Figure 5 shows how bottom sigma layer temperatures change with increasing resolution. The ocean cools at many places when refining

the horizontal grid spacing from 10 km to 4 km (Fig. 5a). Differences exceed 1 °C in the eastern Bellingshausen Sea and in the eastern Ross Sea, and are on the order of 0.25 °C in the Amundsen Sea and around the East Antarctic coastline.

In contrast, increasing the resolution further (from 4 km to 2 km, see Fig. 5b) leads to a warming of the Amundsen-Bellingshausen Seas continental shelf. Even though the shelf temperature of the total domain still decreases slightly at the second resolution step, the Amundsen-Bellingshausen Seas is warming. As mentioned earlier, this phenomenon is often asso-

ciated with shoreward heat transport by eddies that need a grid spacing on the order of 1 km to be resolved by ocean models (Dinniman et al., 2016; Mack et al., 2019) and the representation of narrow troughs at the continental shelf break (Nakayama et al., 2014). The cooling north of Nickerson, Sulzberg and Swinburne Ice Shelves might be a consequence of this warming,

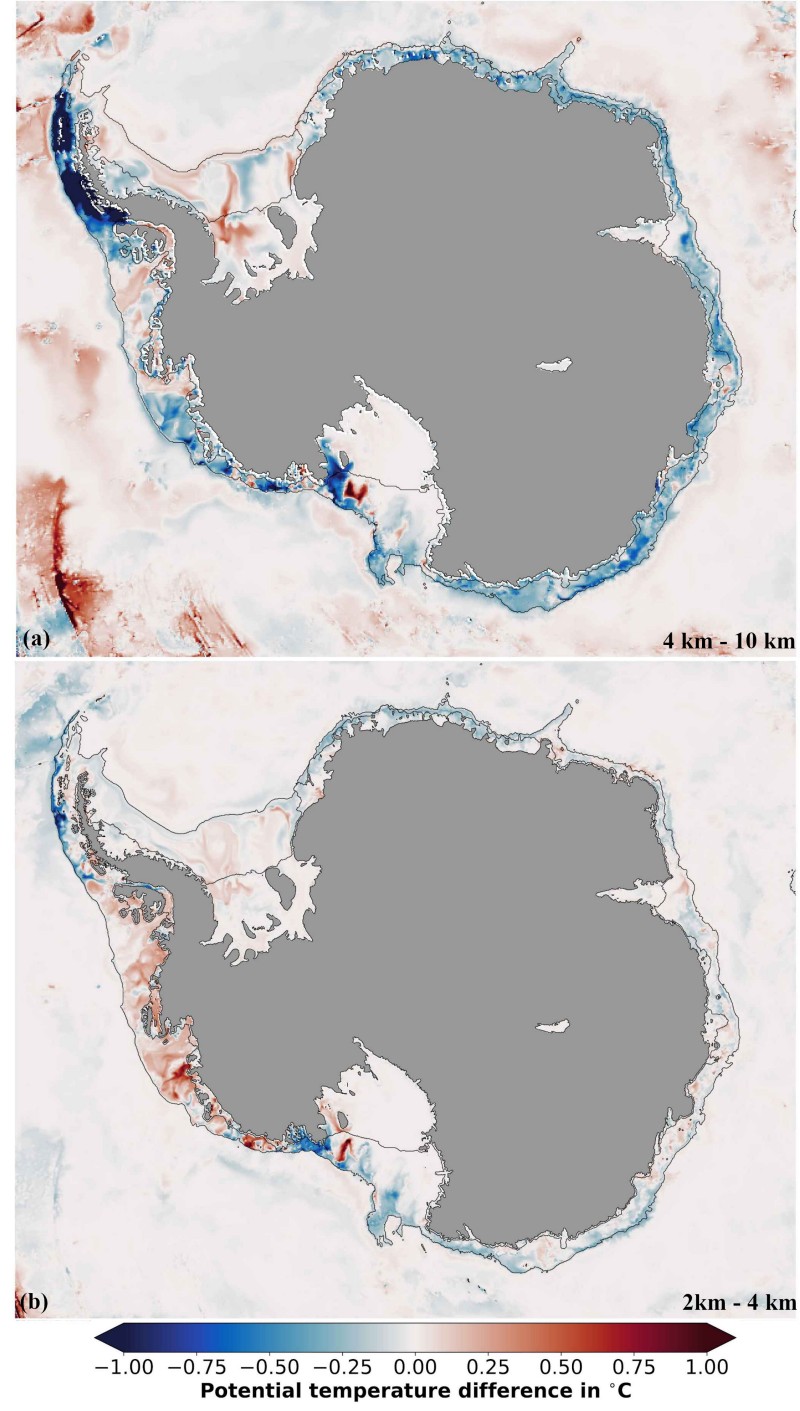

**Figure 5.** The effect of model resolution on bottom layer temperature. Change in annual average bottom sigma-layer potential temperature when increasing the horizontal model resolution from (a) 10 km to 4 km and (b) 4 km to 2 km. Black contour lines indicate the 1500 m isobath and ice shelf fronts.

as the continental shelf current drives melt water from the Amundsen-Bellingshausen Seas mostly westward (Nakayama et al., 2017).

## 3.3 Ice Shelf Melting

Estimates of ice shelf basal mass loss generally agree with satellite observations in many regions. Figure 6 compares mass loss estimates for major ice shelves and Antarctica in total from this study against estimates from satellite observations and other ocean models (see Tab. A1 for underlying data). Using all ice in the model (according to Bedmap2, see Section 2.6), we calculate a total mass loss of 1209 Gt/yr (equivalent to an average melt rate of 0.82 m/yr). This is only 4 % below the range of estimates based on remote sensing data and models of surface processes (1263 Gt/yr to 1737 Gt/yr; Rignot et al., 2013; Depoorter et al., 2013; Liu et al., 2015). Regionally, the model and data show larger differences for some ice shelves (Pine Island, Getz, combined Brunt and Riiser-Larsen, Shackleton, combined Totten and Moscow University), but are in agreement or close to others (George VI, Abbot, combined Fimbulisen and Jelbart, Filchner-Ronne, Larsen C, Ross, Amery). In most regions of disagreement (Pine Island, Getz, Shackleton, combined Totten and Moscow University), satellite estimates suggest higher melting consistent with results from regional studies (e.g. Gwyther et al., 2014, for Totten and Moscow University Ice Shelves; Dutrieux et al., 2013, and Shean et al., 2018, for Pine Island Ice Shelf; Jacobs et al., 2013, for Getz Ice Shelf).

Melting and refreezing at high resolution shows that WAOM resolves many of the key features known from observations. Figure 7 presents ice shelf basal melt rates and bottom layer temperature around Antarctica from this study. In cold regimes, for example, HSSW often drives strong melting along deep grounding lines followed by refreezing along western outflows (defined as Mode 1 melting by Jacobs et al., 1992). WAOM's melt rates resemble this pattern at many places under the large cold water ice shelves in agreement with regional studies (e.g. under the Filchner-Ronne Ice Shelf in agreement with Holland et al., 2007; under the Larsen C Ice Shelf in agreement with Holland et al., 2009; under the Amery Ice Shelf in agreement with Galton-Fenzi et al., 2012).

It is further known that ice-ocean interaction in the Amundsen-Bellingshausen Seas is governed by intrusions of warm CDW that drive strong melting at all depths (Mode 2 melting; see, e.g. Pritchard et al., 2012; Rignot et al., 2013). WAOM resolves this mode of melting for most ice shelves in this region and features bottom layer temperatures comparable to that observed (often warmer than 1 °C; see, e.g. Schmidtko et al., 2014, their Fig. 1A; Pritchard et al., 2012, their Fig. 2).

WAOM also resolves other features in cold water regions that agree with observations. For example, the model indicates enhanced melting in the northwestern part of Ronne Ice Shelf, while predicting refreezing north of Henry Ice Rise and east of Berkner Island. All of these features are also reported by Joughin and Padman (2003) and Rignot et al. (2013), even though the magnitude and extent of marine ice accretion is generally lower in the model. Further, the model predicts elevated melt rates along the deep keel of the Fimbul Ice Shelf and this has also been reported by a well constrained regional simulation by Hattermann et al. (2014).

The final melting mode (Mode 3) describes elevated melt rates close to the ice front and WAOM suggests that this melting is apparent everywhere. Jacobs et al. (1992) hypothesise that intrusions of warm surface waters cause strong melting at the frontal zone of ice shelves (often defined as outermost 50 km) at most places around Antarctica. In situ observations have

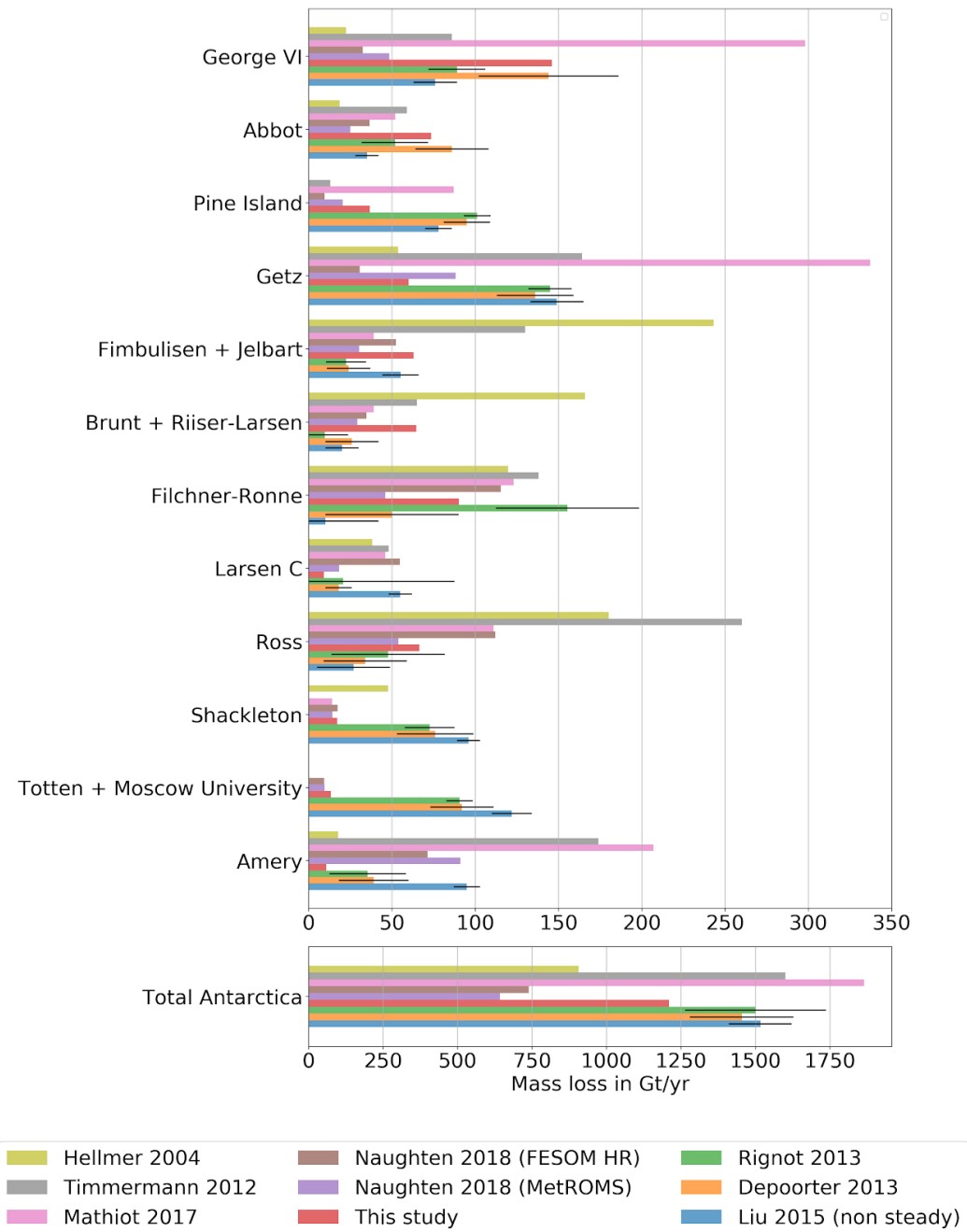

**Figure 6.** Ice shelf basal mass loss from models and satellite observations (equiv. Tab. A1). Estimates of ice shelf basal mass loss for total Antarctica and major ice shelves individually derived from previous ocean-models (Hellmer, 2004; Timmermann et al., 2012; Mathiot et al., 2017; Naughten et al., 2018b), this study and methods combining satellite data with models of surface processes (Rignot et al., 2013; Depoorter et al., 2013; Liu et al., 2015). Among the satellite studies, only Liu et al. (2015) avoids the assumption of a steady state calving front when inferring basal conditions (see Liu et al., 2015, for implications).

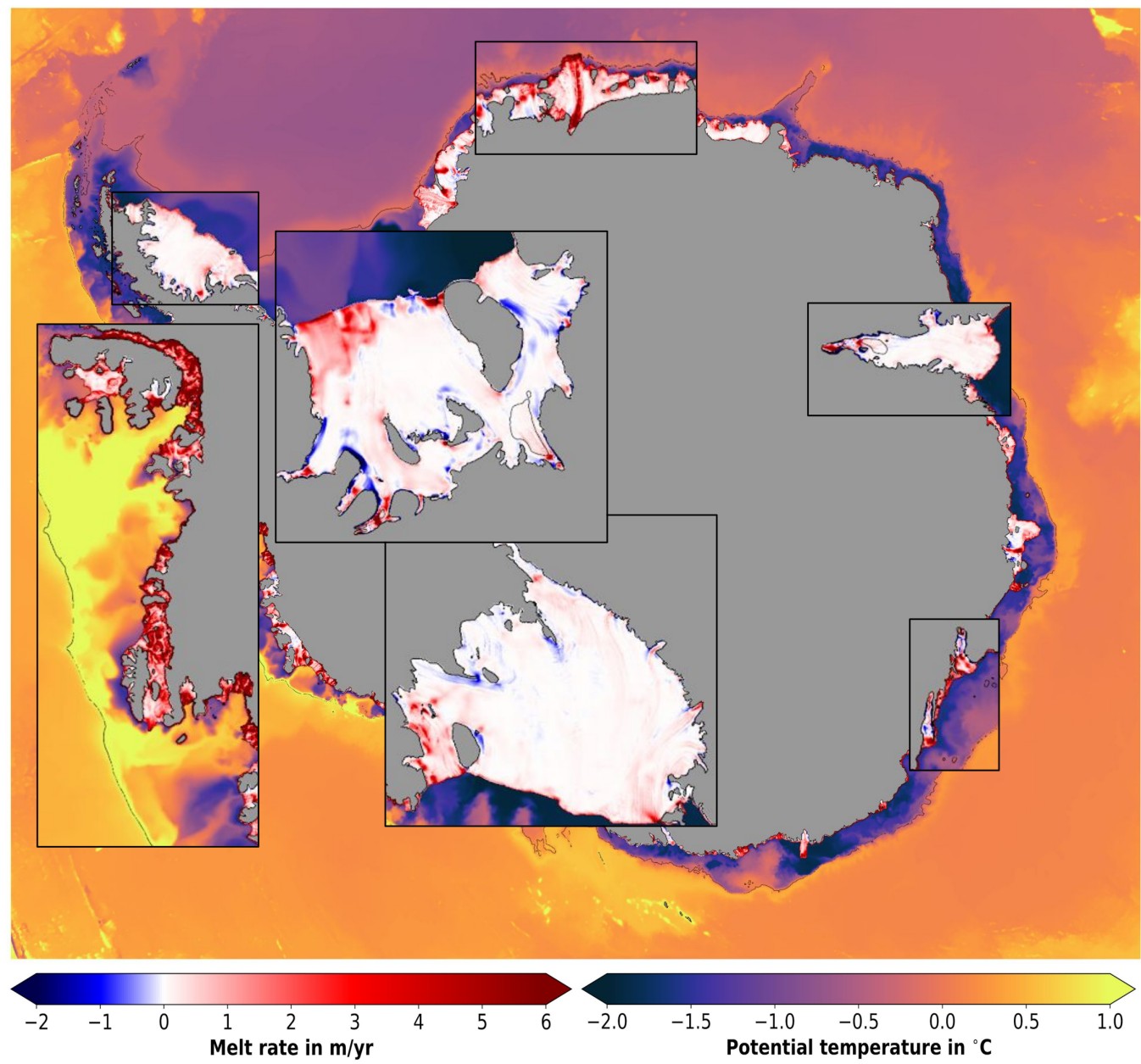

**Figure 7.** Ice shelf melting and bottom layer temperatures. Annual ice shelf basal melt rate is shown where ice shelves are present (negative is refreezing, note the shifted colorbar). Colors seaward of ice shelves show the annual average bottom sigma layer potential temperature. Thin black lines represent the 1500 m isobath.

confirmed Mode 3 melting for parts of the Ross, McMurdow and Fimbul Ice Shelf (e.g. Hattermann et al., 2012; Stern et al., 2013; Stewart et al., 2019) and WAOM suggests elevated melt rates in all these regions, with melt rate magnitudes comparable to the observations: about 3 m/yr at the Ross ice shelf front (see Horgan et al., 2011; Stewart et al., 2019) and about 1 m/yr at Windless Bight (see Stern et al., 2013). The simulation results further suggest ice shelf front melting is not limited to these regions, but rather is a widespread feature.

### 3.4 Ocean hydrography

Figure 8 compares the Temperature-Salinity-Depth distribution of WAOM's on-shelf water masses in summer with the summer climatologies of WOA18 (see Methods, Section 2.6). Most of the subsurface waters (Circumpolar Deep Water, CDW, Modified Circumpolar Deep Water, MCDW, and Low Salinity Shelf Water, LSSW) are well represented in the model. However, High Salinity Shelf Water (HSSW), characterised by temperatures close to freezing and salinities higher than 34.5, is almost entirely missing. In general, HSSW is the densest water mass on the shelf and mixes with other, lighter waters. As a consequence of its absence, all water masses in WAOM are well restricted by the same isopycnal of $1027.8 \ \mathrm{kg \, m^{-3}}$ (also within the cavities, see Fig. D2). We define the near surface ocean as the 15-100 m depth range (the uppermost 2 model layers are excluded due to limitations in the flux-forcing approach, see Section 2.6). At these depths, WOA18 is mostly colder than $0 \ ^{\circ}$C. While WAOM predicts similar upper ocean temperatures in some regions, we also identify waters of up to $1 \ ^{\circ}$C at 15 m depth. Finally, WOA18's water masses feature salinities as fresh as 33.5 (upper ocean and LSSW), but WAOM only reaches 33.75. Together with the lack of the densest waters, this hints towards overly mixed conditions in WAOM. Ice Shelf Water (ISW) in WOA18 remains within $0.25 \ ^{\circ}$C below freezing and is apparent over a wide range of salinities (33.75 to 34.75). In contrast, WAOM features ISW with temperatures of more than $0.5 \ ^{\circ}$C below freezing, but a narrower range of salinities of 34.25 to 34.6.

Figure 9 compares maps of the annual mean bottom layer hydrography of the on-shelf ocean from WAOM against observational estimates by Schmidtko et al. (2014, multidecadal mean from CDW measurements; see Methods, Sect. 2.6). Figure 10 presents sector-wise averages of this comparison. WAOM qualitatively captures the distinction between cold and warm regimes, as the bottom waters of the Amundsen-Bellingshausen Seas are distinctly warmer than in the other sectors (Fig. 10a). However, three main modes of biases are also apparent. First, in the Amundsen-Bellingshausen Seas, predicted bottom waters are too fresh and cold. In particular, the deep waters in the Bellingshausen Seas are on average about $0.75 \ ^{\circ}$C colder in the model compared to the observations (Fig. 10a). The spatial characteristics show that the temperature bias in these regions are often small at the shelf break and increase towards the coast (Fig. 9c). This supports the idea that CDW crosses the shelf break in sufficient amounts, but then is getting mixed with the upper ocean too readily before reaching the ice. Second, a warm and fresh bias is apparent in the Ronne Depression and some parts of East Antarctica, related to the previously identified lack of HSSW formation in the model. Third, in the eastern Ross Sea, a warm bias is combined with accurate salinities. The temperature bias is strongest at the shelf break and diminishes towards the ice, hinting towards intrusions of CDW across the shelf break. Accurate salinities would then be explained by salty CDW offsetting the fresh bias from missing HSSW.

Figures 11 to 13 compare longitudinal transects of temperature and salinity in summer between WAOM and WOA18 in key regions (see Methods, Section 2.6). Mean temperature differences are small (less than $0.4 \ ^{\circ}$C), further supporting that

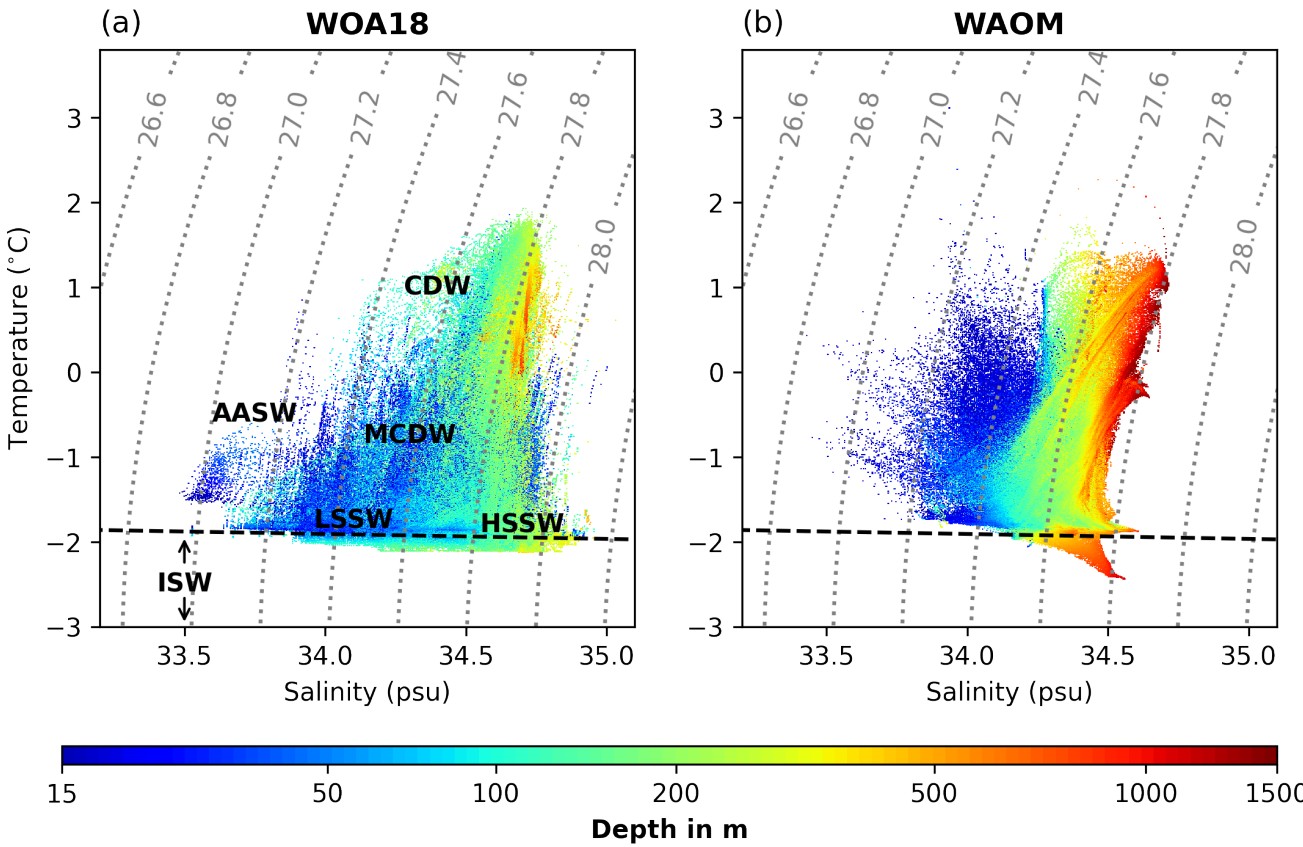

**Figure 8.** On-shelf summer water masses from (a) WAO18, (b) WAOM. Shown are the Potential Temperature-Salinity-Depth distributions of the continental shelf ocean (south of 1500 m isobath and excluding sub-ice shelf cavities) averaged over December, January and February. WOA18 is the seasonal climatology from 2005 to 2017, while WAOM is 2007 only. The uppermost 15 m are excluded for reasons given in the text (see Methods Section 2.6). Each product has been analysed on their original grid. For the analysis, each grid cell has been sorted into 1000x1000 temperature and salinity bins and the depth shown for each bin is the volume-weighted average of all the grid cells in this bin. The dashed black lines show the freezing point at the surface and the dotted grey lines are potential density anomaly contours (in $\mathrm{kg\,m^{-3}}$-1000; referenced to the surface). Labels show different water masses referred to in the text: CDW indicates Circumpolar Deep Water, MCDW indicates Modified Circumpolar Deep Water, LSSW indicates low-salinity shelf water, HSSW indicates high-salinity shelf water, AASW indicates Antarctic Surface Water, and ISW indicates Ice-Shelf Water. WAOM presents a lack of HSSW and bias towards warm waters at shallow and intermediate depths.

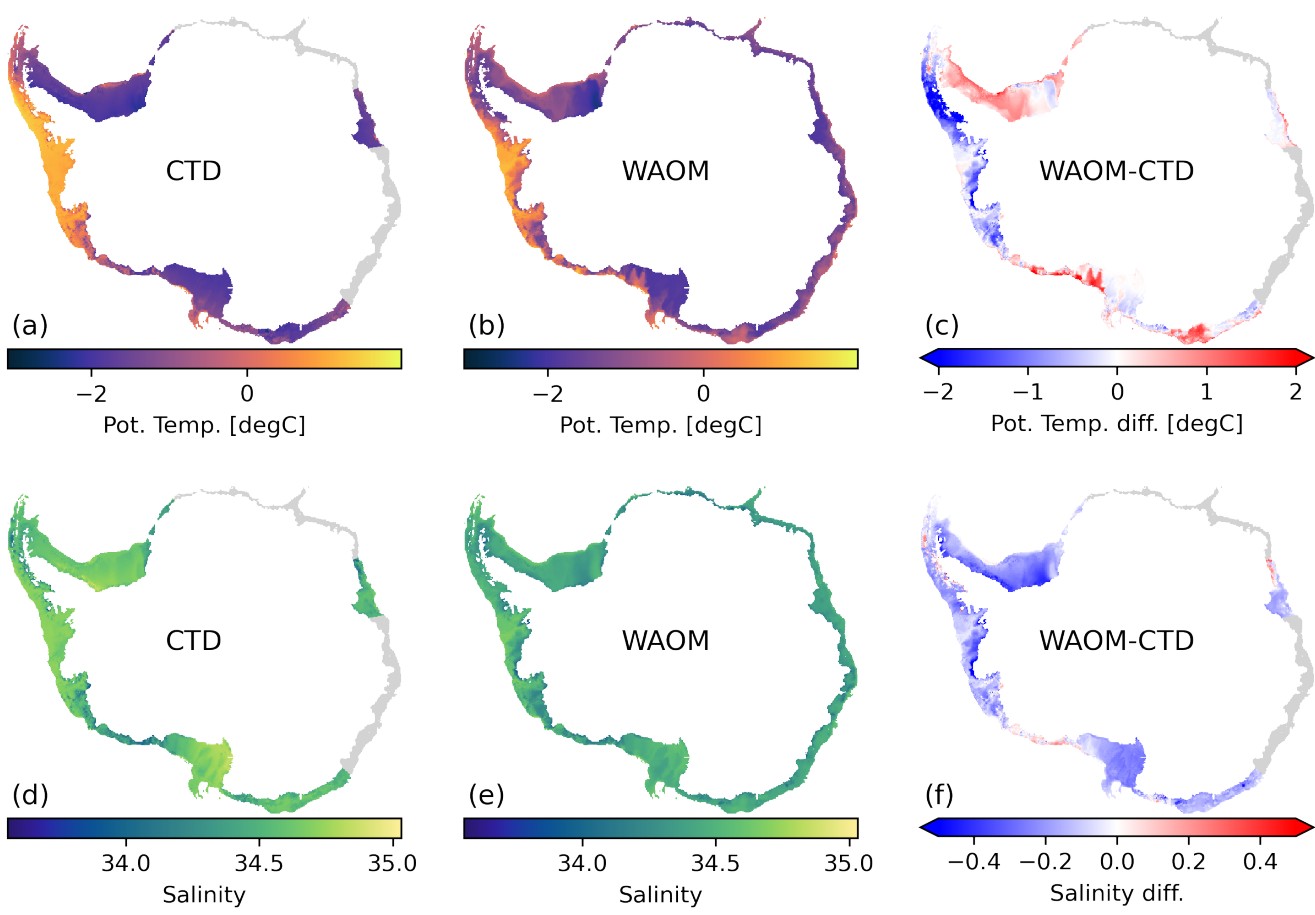

**Figure 9.** Spatial distribution of on-shelf bottom layer hydrography compared against observations. (a) and (d) are multi decadal mean of Potential Temperature and Practical Salinity from Schmidtko et al. (2014). (b) and (e) are 2007 mean of the same quantities as predicted by WAOM and (c) and (f) are the differences between the model and the observations (WAOM - CTD). For the analysis bottom layer CTD measurements have been converted to model quantities (Conservative Temperature to Potential Temperature; Absolute Salinity to Practical Salinity) and interpolated onto the model grid using the nearest neighbour scheme. Model data has been interpolated to the same depth as the observations using the nearest neighbour scheme. Regions with sparse observations have been excluded from the analysis (Western East Antarctica and Sabrina Coast; see Fig. 1 for Sector boundaries)

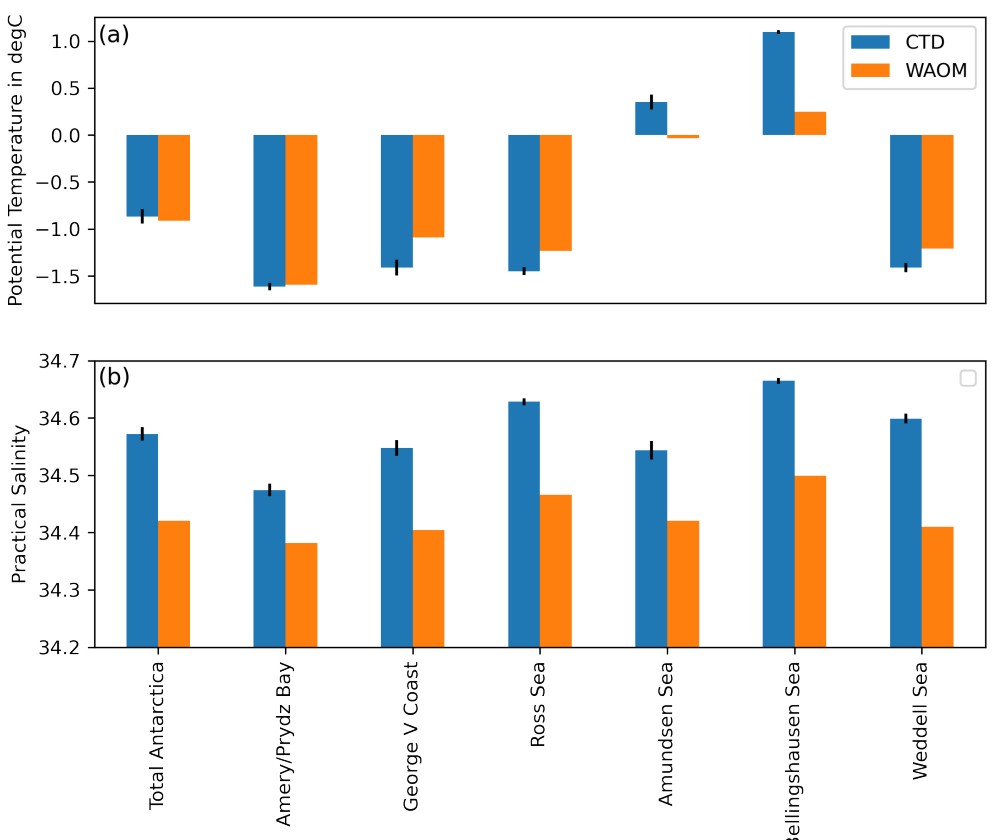

**Figure 10.** Sector-wise mean of on-shelf bottom layer hydrography from WAOM and observations. (a) Potential temperature and (b) Practical Salinity. As Figure 9, but area averaged over individual Antarctic sectors. CTD data also shows the sector mean of the standard deviations provided by Schmidtko et al. (2014). Regions with sparse observations have been excluded from the analysis (Western East Antarctica and Sabrina Coast; see Fig. 1 for Sector boundaries).

WAOM captures the difference between warm and cold regimes correctly. In all transects, however, WAOM is less stratified than WOA18, as salinity differences oppose the observed salinity trends of the region (salinity controls stratification in the Southern Ocean). In agreement with TS-distribution of the entire shelf (Fig. 8), the deep waters in the troughs of the Ross Sea are too fresh compared to WOA18 estimates.

Figure 14 showcases predicted annual mean temperature-salinity transects in key regions on the continental shelf and on the original model grid (2 km resolution). These transects show that WAOM qualitatively captures many of the known regional characteristics of the Antarctic continental hydrography. Examples are given in the following. In the Weddell Sea, ISW resides at the bottom of the Filchner trough, while warmer waters at mid depth resemble characteristics of Modified Weddell Deep Water or Eastern Shelf Water (Fig. 14a; in agreement with, e.g. Nicholls et al., 2009, their Fig. 7). In contrast, deep waters in

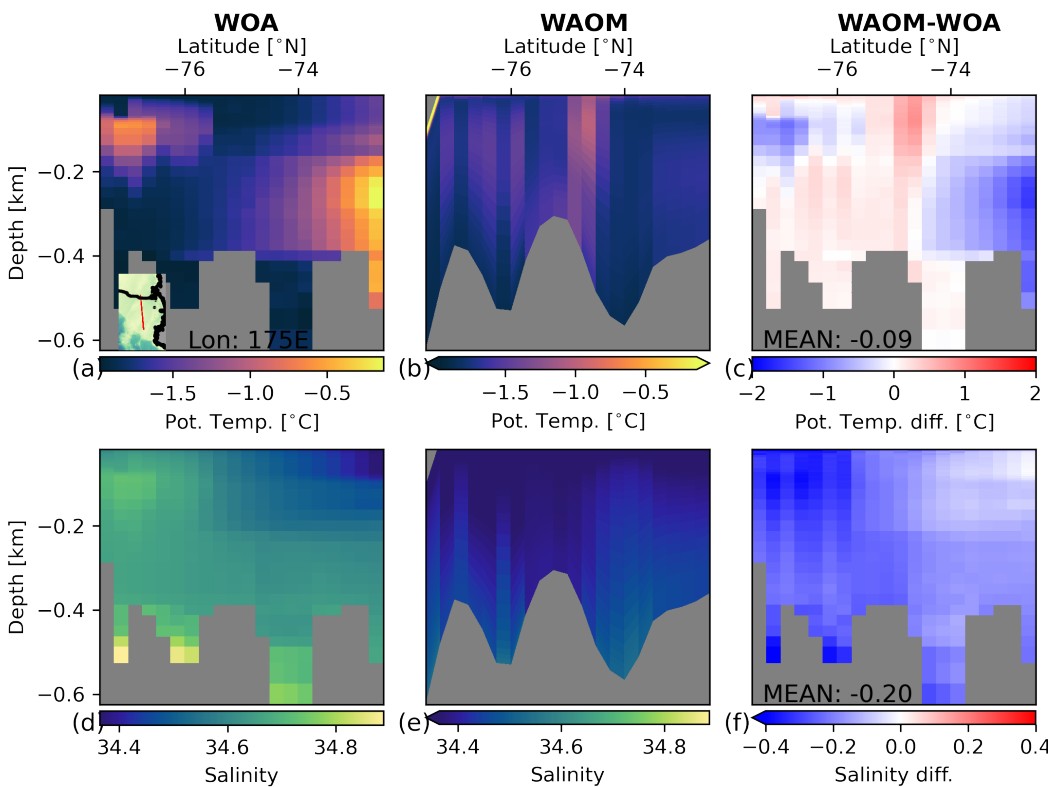

**Figure 11.** Temperature and Salinity transect on the Ross Sea continental shelf (175 °E) compared against observations. (a) and (d) are WOA18 2005-2017 summer mean temperature and salinity, (b) and (e) are 2007 summer mean temperature and salinity from our model (WAOM), and (c) and (f) are the respective differences (WAOM - WOA18). WAOM's data has been interpolated to the WOA18 grid using nearest neighbours (for b and e in the horizontal; for c and f in the horizontal and vertical).

the Amundsen Sea sector feature some of the highest temperatures of the entire Antarctic continental shelf (Fig. 14b). These CDW intrusions are overlaid by colder Winter Water and only held stable by a large gradient in salinity (in agreement with, e.g. Jacobs et al., 2011). Further, inside the Amery Ice Shelf cavity, we detect dense, cold waters at the bottom of the water column (hinting towards HSSW properties, even though they are not salty enough) and ISW at the top of the water column (Fig. 14c; in agreement with, e.g. Galton-Fenzi et al., 2012, their Fig. 9). In this region, CDW is held back from entering the continental

shelf by a sharp front (the Antarctic Slope Front; exaggerated by the choice of color scale; in agreement with, e.g. Guo et al., 2019, their Fig. 2). Along the Sabrina and George V coasts, some MCDW crosses the continental shelf break, e.g. in front of the Totten Ice Shelf (Fig. 14d). Once on the shelf MCDW competes with the lighter WW which occupies most parts of the shelf ocean close to the coast (in agreement with, e.g. Silvano et al., 2017, their Fig. 2 and 3). Finally, we identify advection of warm surface waters into the outer cavities of the Amery ice shelf (Fig. 14c; in agreement with Galton-Fenzi et al., 2012, their

Fig. 9) and the Totten ice shelf (Fig. 14d; in agreement with Silvano et al., 2017, their Fig. 2).

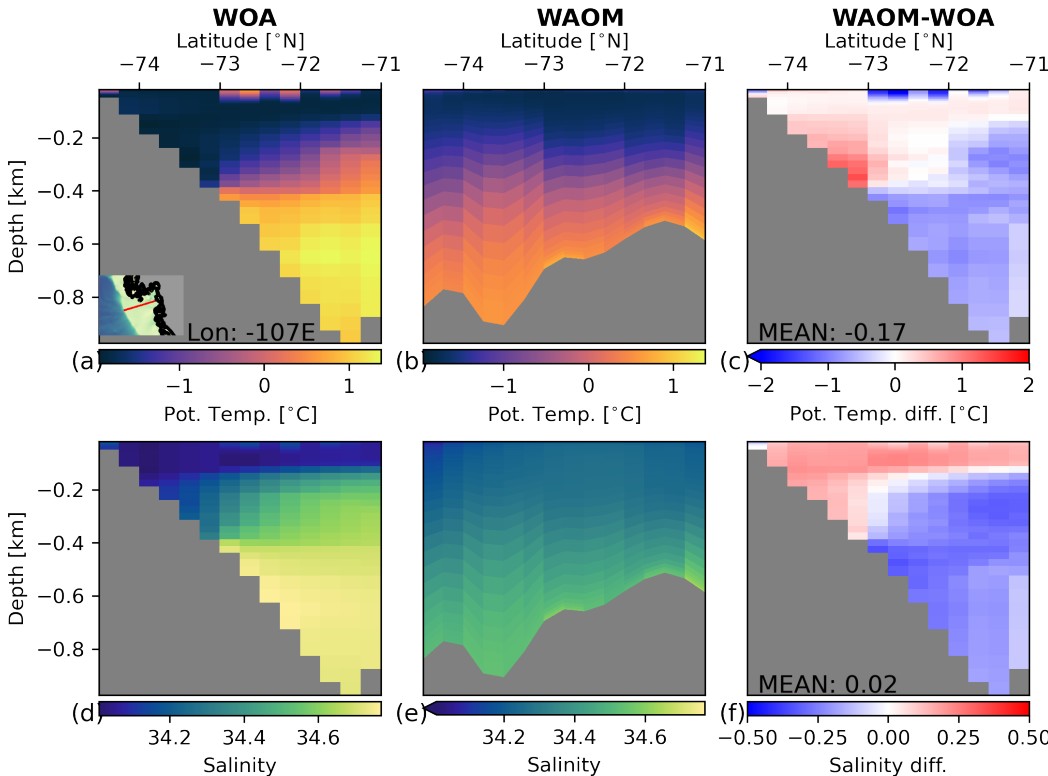

**Figure 12.** As Fig. 11, but for a transect across the Amundsen Seas along 107 °W.

Figure 15 compares the TS distribution of the summer mean climatology from WOA18 against the 2007 summer mean from ECCO2 and WAOM (see Methods, Sect. 2.6). In WAOM, water mass properties at depth mostly resemble the observations, but the warm bias in the upper ocean is even more apparent than on the shelf. Between 15-100 m depth WOA18 is mostly limited to temperatures of less than 0 °C, but WAOM predicts more than 3 °C in some regions. The warm surface bias also affects the properties of adjacent water masses at intermediate depths, effectively warping the overall picture of the TS-distribution away from the freezing point. While ECCO2 also shows shallow waters with temperatures above observed, the bias is less severe and shows less impact on deeper waters. The densest waters in WAOM show only little isopycnal mixing with colder surface waters (also see Fig. D2). In agreement with the earlier identified lack of HSSW formation, this hints towards bottom waters in WAOM, which are mainly sourced by initial and boundary conditions from ECCO2.

Figure 16 compares the annual mean bottom layer hydrography from WOA18 against ECCO2 and WAOM. WAOM shows an overall warm bias by about 0.3 °C, which can now clearly be attributed to the initial and boundary conditions from ECCO2. Bottom layer salinities in both models agree well with WOA18. All biases revealed in this section are discussed later in respect of their sources and consequences for ice shelf-ocean interaction (see Sect. Discussion).

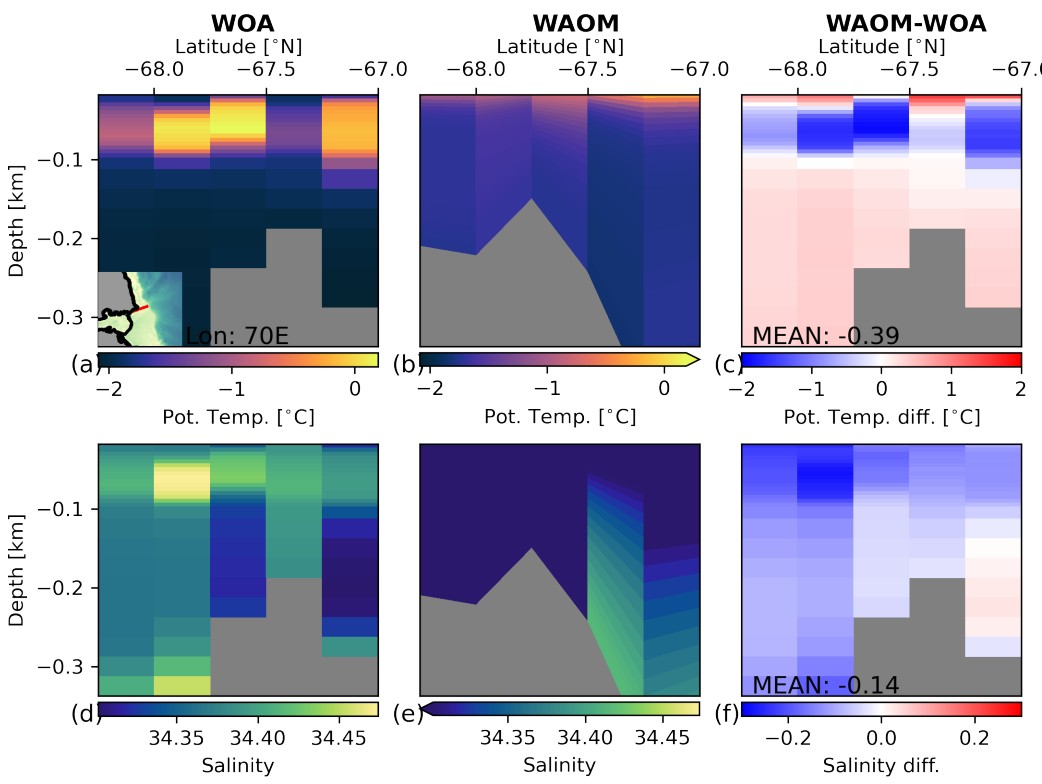

**Figure 13.** As Fig. 11, but for a transect in Prydz Bay (Davis Sea continental shelf) along 70 °E.

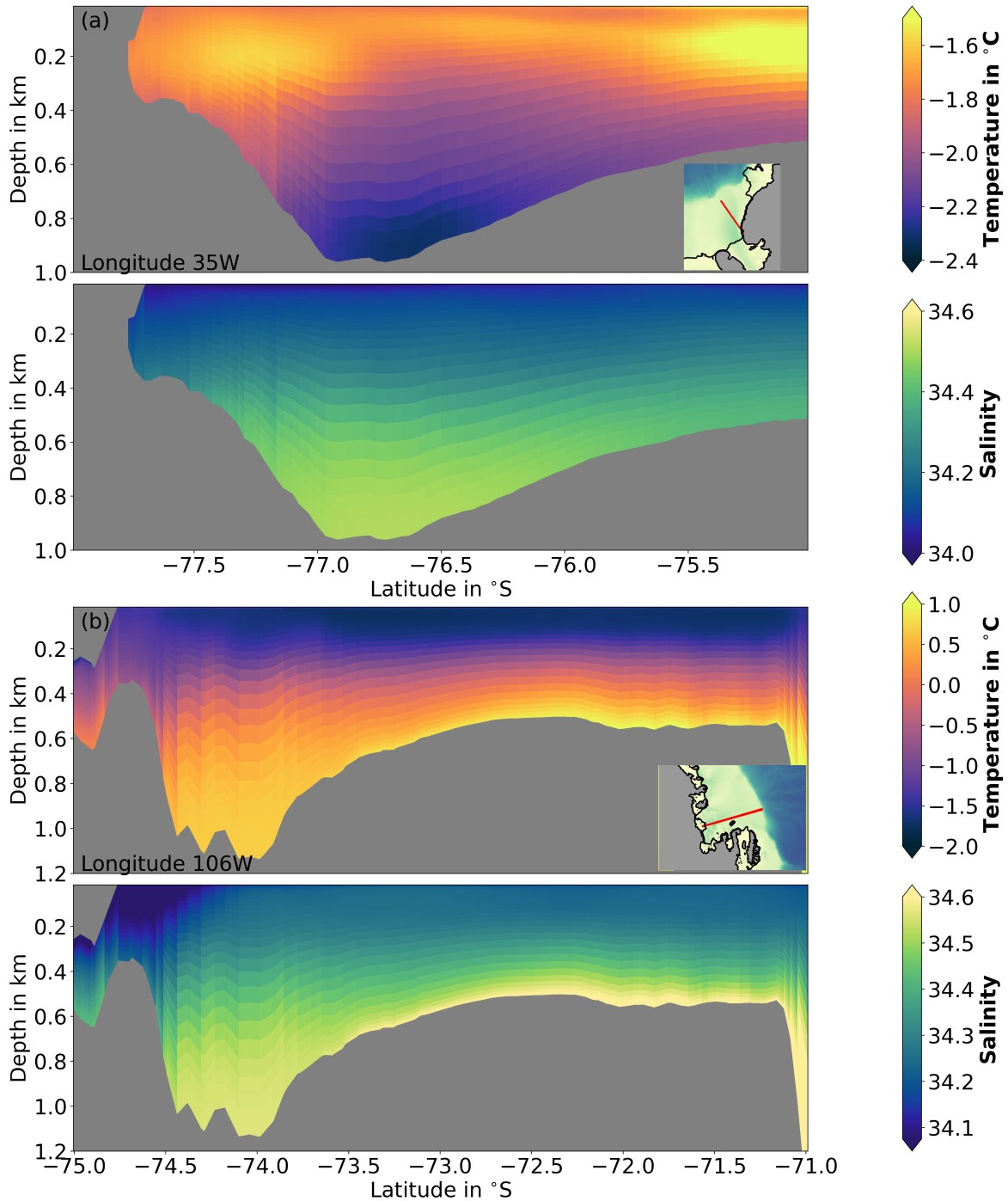

**Figure 14.** Temperature-Salinity distribution on (a) the Weddell Sea continental shelf at 35 °W, (b) the Amundsen Sea at 106 °W, (c) the Prydz Bay at 72 °E and (d) the Sabrina Coast at 120 °E. Inlets show the transect locations.

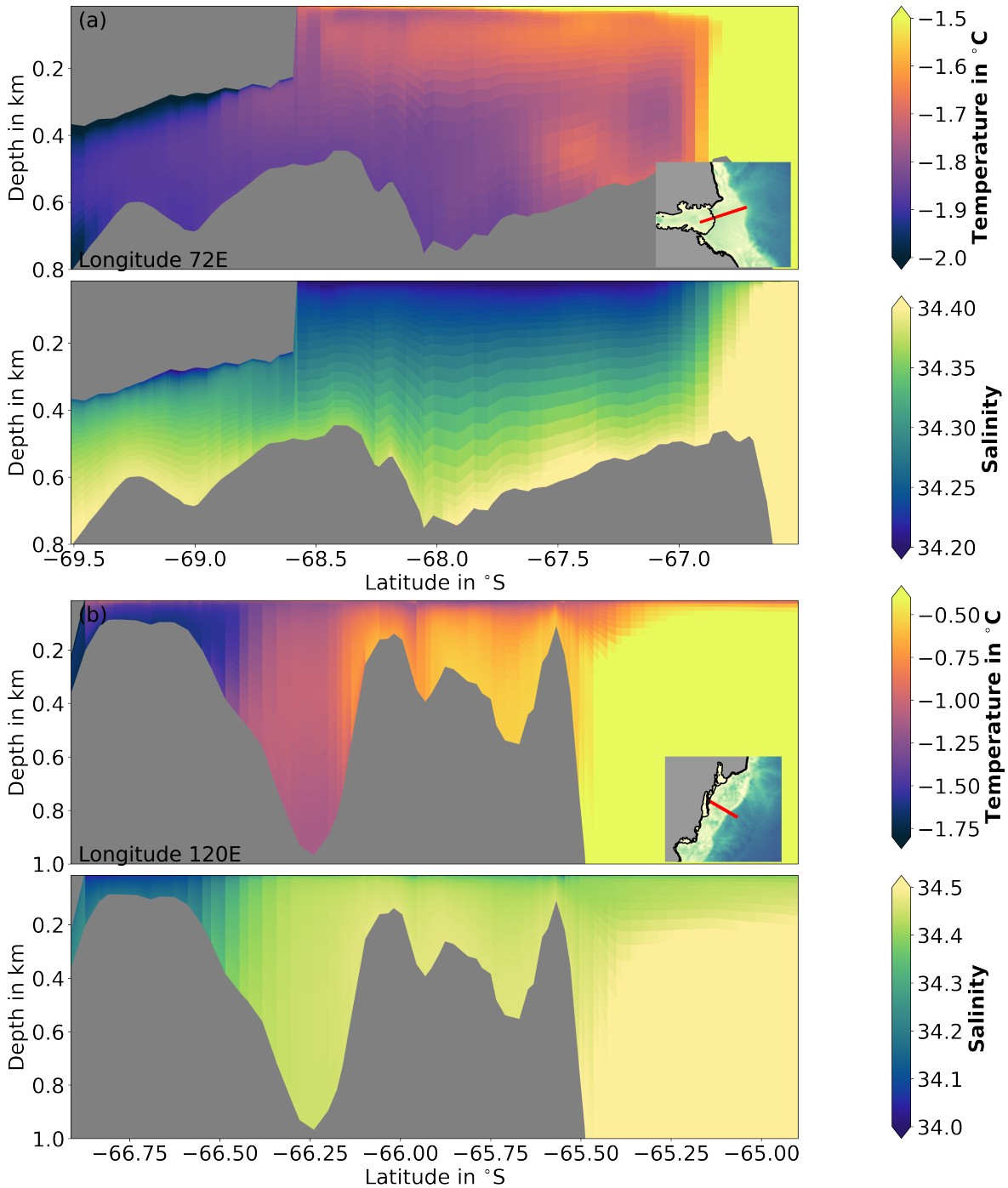

**Figure 14.** Temperature-Salinity distribution on (a) the Weddell Sea continental shelf at 35 °W, (b) the Amundsen Sea at 106 °W, (c) the Prydz Bay at 72 °E and (d) the Sabrina Coast at 120 °E. Inlets show the transect locations. (cont.)

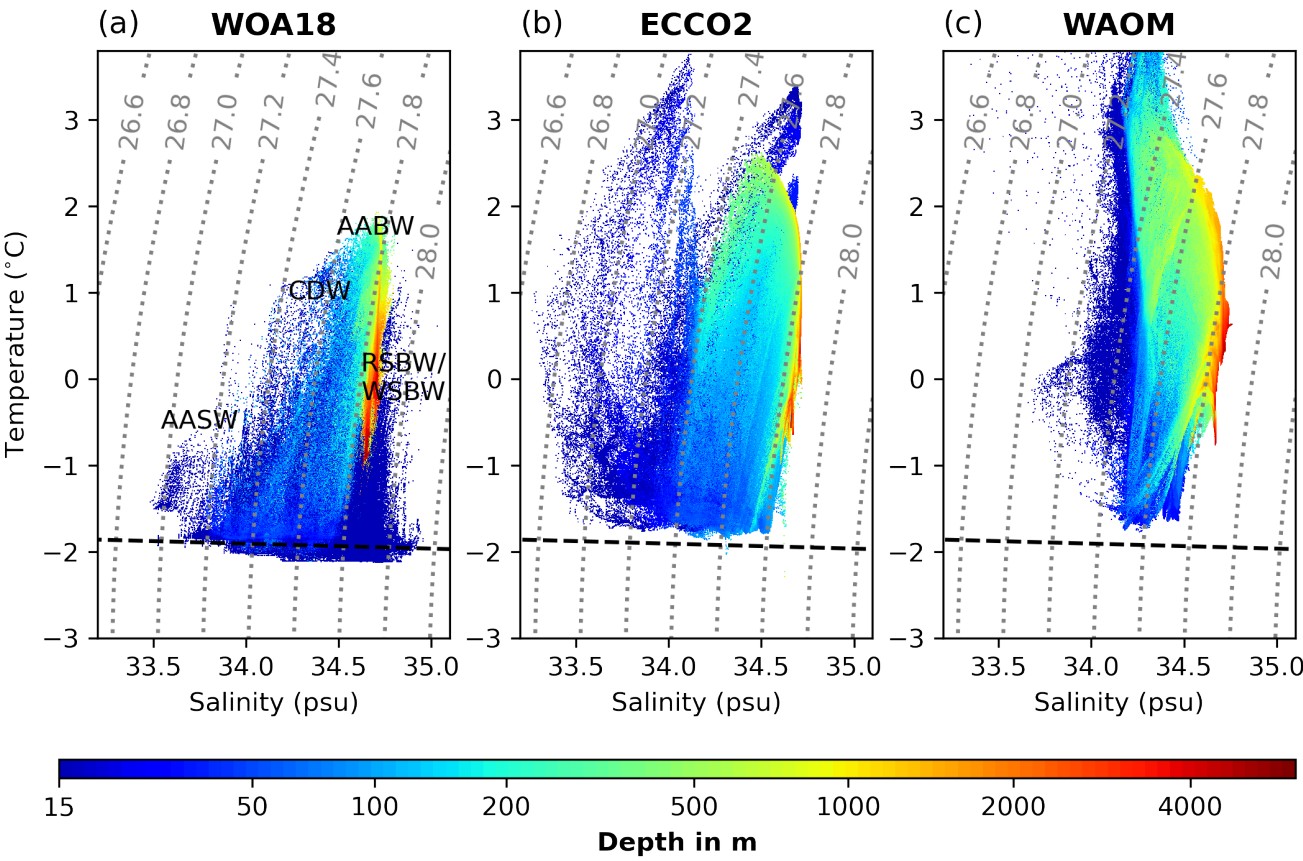

**Figure 15.** Off-shelf summer water masses from (a) WAO18, (b) ECCO2 and (c) WAOM. Shown are the Potential Temperature-Salinity-Depth distributions north of 1500 m isobath and south of 65 °S averaged over December, January and February. WOA18 is the seasonal climatology from 2005 to 2017, while ECCO2 and WAOM is 2007 only. The uppermost 15 m are excluded for reasons given in the text. Each product has been analysed on their original grid. For the analysis, each grid cell has been sorted into 1000x1000 temperature and salinity bins and the depth shown for each bin is the volume-weighted average of all the grid cells in this bin. The dashed black lines show the freezing point at the surface and the dotted grey lines are potential density anomaly contours (in $kg\,m^{-3}$-1000; referenced to the surface). Labels show different water masses referred to in the text: AABW indicates Antarctic Bottom Water, WSBW/RSBW indicates Weddell/Ross Sea Bottom Water, CDW indicates Circumpolar Deep Water and AASW indicates Antarctic Surface Water. WAOM has a fresh and warm bias, which originates from the surface and can not be explained by boundary or initial conditions (ECCO2).

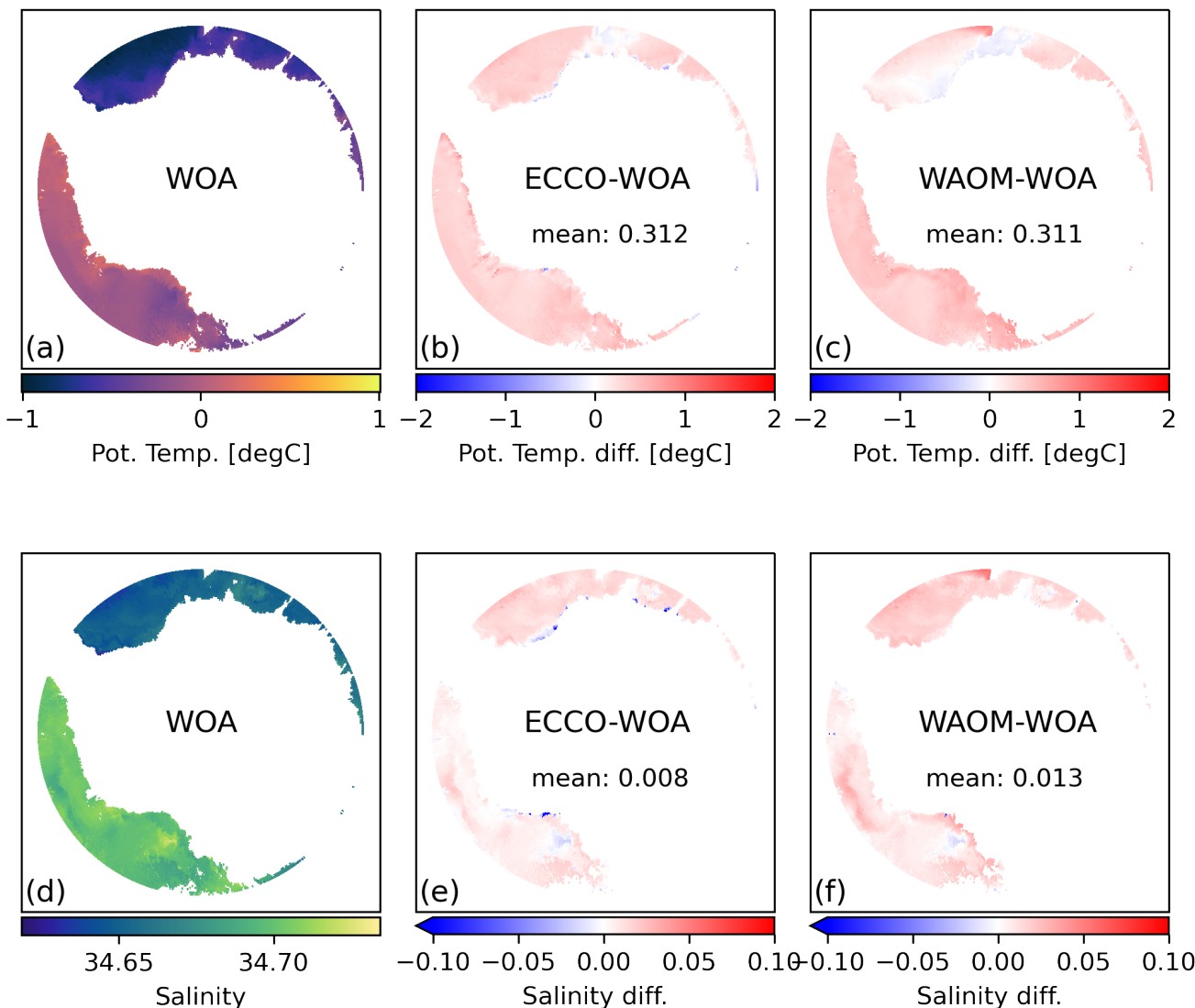

**Figure 16.** Mean bottom water hydrography compared against observations. (a) WOA18 2005 to 2017 climatology mean bottom layer Potential Temperature, (b) difference to ECCO2 2007 mean (ECCO2-WOA18) and (c) difference to WAOM 2007 mean. (d) to (f) are the same for salinity. WAOM and ECCO data has been interpolated to the WOA18 bottom layer using linear interpolation in the vertical and nearest neighbours in the horizontal. Only data for depths below 3000 m and south of 65 °S are shown. WAOM has a salty and warm bias, which can mostly be explained by initial and boundary conditions (ECCO2).

## 4 Discussion

Compared to other models, WAOM includes tides and an eddy resolving resolution, a first for a circum-Antarctic ice-ocean simulation. These features are critical for resolving ocean-ice shelf interactions accurately and, thus, we consider ice shelf melting and the causal oceanic mechanisms at improved resolution as WAOM's most valuable contribution to ice shelf-ocean research. These melt rates are fully independent from satellite based approaches and will provide new, quantitative insights into the driving mechanisms of ice shelf melting in a pan-Antarctic context. Further, idealized studies have started to explore

the average behaviour of the ice shelf cavity system, including its response to a warming ocean (e.g. Holland et al., 2008; Little et al., 2009; Gwyther et al., 2016; Holland, 2017). WAOM provides 176 realistic ice shelf cavities with a single simulation spanning the entire range of present day geometries and ocean conditions. Exploring relations in the average quantities between these systems might help to extrapolate the overall future response of ice shelf melting around Antarctica. Finally, ocean-models are well suited for perturbation experiments and, in the case of WAOM, these can be used to study ocean-ice processes in more

detail, for example, the impact of tides or ice shelf teleconnections.

Against expectations, coarsening the model resolution results in an overall warming of the continental shelf ocean and we attribute this to less accurately resolved tidal processes. Previous studies without tides generally suggest a warming trend of the continental shelf when increasing the resolution from tens of kilometres to kilometres (e.g. Dinniman et al., 2015; Naughten et al., 2018b). This behaviour has been attributed to better resolved bathymetric features, such as troughs, and eddies that act

to increase heat transport onto the shelf (Nakayama et al., 2014; Stewart and Thompson, 2015). The results presented here support the importance of shoreward heat flux by eddies and bathymetry in some regions, e.g., the Amundsen-Bellingshausen Seas. The overall picture, however, is dominated by different processes. Increasing the resolution leads to an overall cooling of the continental shelf. Similar studies without tides only report a warming with increasing resolution (e.g. Nakayama et al., 2014; Dinniman et al., 2016), hinting towards better resolved tidal processes to be the cause. A cooling continental shelf could

either be realized by decreased heat flux onto the shelf, increased heat flux to the atmosphere/sea-ice or increased heat flux into the ice. Stewart et al. (2018) find that tide driven heat flux across the shelf break is mostly balanced by mean flow and, in our simulation, melt rates also decrease with increasing resolution (Fig. 4) and changes in temperature are strongest outside the cavities (Fig. 5). Hence, we hypothesise that increased vertical mixing due to better resolved tidal processes are responsible for the reported continental shelf cooling with increasing resolution. Studies aiming to use WAOM for future predictions should

consider the option of applying it at 10 km or 4 km horizontal resolution for computational efficiency. Such studies will need to evaluate the model (at different resolutions) depending on their research question. Judging on the single scale metric of mean ice shelf melting, the 4 km solution of WAOM is closest to the observations (Fig. 4). For process oriented studies, however, we recommend using the 2 km version, as resolving eddies at a kilometer scale resolution is critical for accurate ice shelf-ocean interaction in some regions (Stewart and Thompson, 2015). Ultimately, we should direct future efforts towards an accurate

eddying model with tides.

Quantifying changes in the heat budget of the continental shelf ocean and determining the exact tidal mechanism responsible for the model behaviour will require future studies. We hypothesise, however, that vertical mixing on the continental shelf due

to internal tide breaking could play an important role. This is based on the following. First, by means of a high resolution circum-Antarctic simulation, Stewart et al. (2018) conclude that tide driven exchanges across the continental shelf break are mostly balanced by mean flow, and, second, the generation of internal tides is sensitive to horizontal model resolution with 4 km being sufficient to resolve the most critical aspects (Robertson, 2006; Padman et al., 2006).

WAOM underestimates melting for some warm water ice shelves and produces too little HSSW, both likely related to overly mixed conditions on the continental shelf. A cold bias in the Amundsen-Bellingshausen Seas is a common issue in large-scale models (e.g. Naughten et al., 2018b) and has been attributed to, either, insufficient transport of CDW onto the continental shelf (e.g. Thoma et al., 2008; Nakayama et al., 2014, discussed earlier), too rapid erosion of heat on the shelf (e.g. Bett et al., 2020), or underestimated conversion efficiency of heat into melting inside the cavity (e.g. Dinniman et al., 2015). The ocean evaluation indicates that the second cause applies in our case. CDW enters the shelf, but gets mixed away too readily before reaching the ice (Fig. 9c). Indeed, WAOM is overly mixed in many regions (incl. the Bellingshausen Seas; see Fig. 12). Spurious currents from pressure gradient force errors may explain part of the discrepancy in the Amundsen-Bellingshausen Seas, but not in other regions (see Sec. 2.3). We note that winds have also been shown to affect shoreward heat transport (e.g. Kimura et al., 2017; Greene et al., 2017) and we do not account for the effect of sea ice on wind stress. However, the sensitivity of ice shelf melting to momentum flux modulations have yet to be explored (as done in Jendersie et al., 2018).

Too much mixing might also be responsible for the reported lack of HSSW formation (Fig. 8). Integrated surface salt input in polynya areas compares well against the original forcing product by Tamura et al. (2011) (not shown), hence our surface salt flux tuning (see Methods) is not the cause for the bias. Instead, waters with salinities higher than 34.5 are indeed present in the uppermost 15 m, but readily mix within this layer before reaching greater depths (Appendix Fig. D2). The reported warm bias at the surface (Fig. 8) could also be linked to reduced HSSW formation. WAOM predicts elevated melt rates right at the ice front in most regions (close to coastal polynyas; Fig. 7) and ISW has been shown to be able to suppress dense water formation (Williams et al., 2016; Silvano et al., 2018). However, we rate this possibility as unlikely, since the warm surface bias is less apparent in winter (not shown), when deep convection events are happening. We note that elevated frontal melting in WAOM is likely favoured by its representation of the ice front. A sloping and smooth representation of the vertical cliff face exposes more ice shelf area to warm surface waters (a geometrical consequence) and eases baroclinic transport (Wåhlin et al., 2020). Ice shelf frontal processes and their representation in models, however, are not well explored. There is evidence, for example, that a smooth representation of the ice front in sigma coordinates actually compensates for an unresolved wedge mechanism that favours intrusions of water surface waters under the ice (Malyarenko et al., 2019). The results presented in this study stress the importance of further research in this area.

We also have reported CDW intrusions onto the continental shelf of the Eastern Ross Sea (see Fig. 9) and this is likely related to boundary effects. Where ACC jets cross the domain's boundary in shallow angles, artificial currents can arise. We have reduced these effects by making the boundary conditions outflow dominant (see Methods), but some artificial currents remain in the Ross Sea (see Fig. D3). We hypothesise that these currents drive CDW onto the shelf by affecting the slope of the isopycnals close to the shelf break.

We consider unresolved sea ice-ocean interactions as the major limitation of WAOM. The ocean connects ice shelf melting and sea ice in a complex manner (Hellmer, 2004; Timmermann and Hellmer, 2013; Padman et al., 2018), having motivated many previous studies to include sea ice models (e.g. Hellmer, 2004; Timmermann et al., 2012; Naughten et al., 2018b). This study, however, follows an approach that prescribes surface fluxes from sea ice observations to accurately capture the position and strength of coastal polynyas. While this is a major component towards accurate ice shelf melt rates, WAOM can not be used to study processes for which sea ice interaction is critical. Future efforts aiming to use WAOM for simulating periods beyond the observational record will need to incorporate a dynamic sea ice model or carefully prescribe surface flux anomalies. Further, the forcing schemes of this first version of WAOM have been designed to study phenomena with hourly to seasonal timescales (e.g. tides and summer surface water advection). To address scientific questions related to inter-annual change, these schemes will need to be extended first. We note that neglecting larger scale variability from interannual change or intrinsic processes (Gwyther et al., 2018) might impact the mean state of the model.

The many wasted land cells in WAOM's domain could also be considered a limitation. WAOM's curvilinear grid using a south polar projection necessitates masking of more than one third of all computational cells, wasting valuable resources with the model integration timestep. This design, however, has been chosen to simplify future efforts that aim to couple WAOM with models of Antarctic ice sheet flow (e.g. Jong et al., 2017), as these coupled models are ultimately needed to improve sea level rise predictions (e.g. Colleoni et al., 2018). Also in regards to coupling, ROMS includes routines to resolve sediment transport and passive tracers (see, e.g. Dinniman et al., 2003; Sherwood et al., 2018, for applications) and activating these options in WAOM will likely be of interest for geological and biological studies.

To further improve the accuracy of WAOM, future development should focus on the following aspects.

- Future studies will need to calibrate and evaluate the model according to their research question. Morrison et al. (2020), for example, uses a pan-Antarctic ocean model to study water mass transport across the shelf break and, hence, evaluates the model using hydrographic profiles in the slope region. Suitable observational datasets for studies focused on the Antarctic seas (in addition to the ones applied in this study) include the Marine Mammals Exploring the Oceans Pole to Pole (MEoP) dataset (Roquet et al., 2014), a review of dense shelf water observations around Antarctica (Amblas, 2018) and a monthly isopycnal/mixed-layer climatology (MIMOC, Schmidtko et al., 2013). Available in-situ observations of ice shelf melting have yet to be compiled (as discussed in the next point).

- Establishing an evaluation matrix for Antarctic ice shelf melting at high resolution would open the path for efficient parameter tuning (similar to Nakayama et al., 2017) and allow the community to compare the performance of different models (see Naughten et al., 2018b) and satellite derived estimates. ApRES seems particularly suitable for large scale model evaluation (e.g. Gwyther et al., 2020) as it comprises a robust and relatively cheap method to observe basal melt rates over longer time periods. As more ApRES measurements are becoming available, their compilation could provide the backbone for such an evaluation matrix, similar to tide gauge measurements for tidal accuracy (King and Padman, 2005). Comparison of a wide array of ApRES data is already underway with the NECKLACE programme[1].

---

[1]NECKLACE programme: http://www.soos.aq/news/current-news/330-necklace-workshop-update.

- To improve WAOM v1.0 (focused on accurate sub-ice shelf melting) future development should focus on reducing mixing to better represent the stratification on the continental shelf. We have scaled horizontal tracer diffusion linearly with resolution, but have not tuned this parameter against observations. Likewise, stratification is sensitive to the chosen mixing (here LMD, which includes KPP) and advection schemes (here 4th-order Akima for the horizontal and vertical), and the effects of different choices have yet to be tested for WAOM. Finally, the sensitivity of stratification to different slope factors (Haney factors) should be explored. Spurious mixing at steep sloping topography (related to pressure gradient force errors in sigma coordinate models; discussed earlier) is sensitive to the degree of smoothing. Our smoothing procedure is similar to regional studies and the smoothing algorithm has been shown to perform well for a realistic, complex case without ice (The Adriatic Sea, see Sikirić et al., 2009). However, spurious currents in our model are significant along the shelf break of the Amundsen-Bellingshausen Seas, possibly reducing the stratification on the adjacent continental shelf. Other pan-Antarctic studies have chosen different routines and algorithms and do not report overly mixed conditions (Naughten et al., 2018b). The Haney factor controls the degree of smoothing within any given scheme and, hence, offers a metric to assess the sensitivity without implementation of new procedures.

- Second priority should be given to the calibration of the surface heat flux, which is likely to reduce the warm surface bias. The warm bias towards the surface can not be explained by initial and boundary conditions, as ECCO2's upper ocean conditions are more realistic (see Fig. 15). Also, 2007 has not been an anomalously warm year (e.g. measured by sea ice extent; see Parkinson, 2019), rendering interannual variability as an unlikely source. Instead, we suspect the applied surface flux schemes to be responsible. A similar scheme is known to overestimate annual heat flux into the ocean by about 50 % (Jendersie et al., 2018). While we aim to account for this by reducing positive heat flux into the ocean by half (see Methods, Sect. 2.6), the approach has not been tested for pan-Antarctic domains.

- In third place, the boundary effects in the Eastern Ross Sea should be addressed. Introducing a sponge layer is difficult, since tides are also forced at the open boundary. Instead we recommend an adjustment to the model boundary locations to avoid intersection with ACC jets at shallow angles.

- Accurate bathymetry on the open continental shelf and inside the sub-ice shelf cavities is essential to resolve warm water intrusions, critical for ice shelf melting and consequent melt water export. Thus, the model bathymetry should be updated according to regional surveys (e.g. Millan et al., 2017; Nash, 2019).

- Studying individual aspects of the model will help gain trust in quantitative results. Schnaase and Timmermann (2019), for example, show that artificially deepening the water column thickness near grounding zones (necessary for numerical stability), does not affect ice shelf average melt rates, and Malyarenko et al. (2019) suggest that the unrealistic ice front representation in sigma-coordinates, could actually account for unresolved small scale processes. Wind stress has been shown to impact ice shelf melting (Davis et al., 2018; Greene et al., 2017; Nakayama et al., 2017; Hattermann, 2018), but how sea ice modulates momentum flux from the atmosphere into the ocean is not well constrained (Lüpkes and Birnbaum, 2005; Nøst et al., 2011; see discussion in Jendersie et al., 2018). Jendersie et al. (2018) provides a first

parameterisation for windstress modulation by sea ice and performs sensitivity experiments using a ROMS configuration of the Ross Sea. The effects on the seasonal variations of the circulation are negligible. WAOM would be well suited to extend these tests for a pan-Antarctic context. If sea ice wind stress modulation is indeed important, ice motion observations could be included for assimilation or calibration.

– The number of wasted land cells in WAOM could be reduced by applying nested grids with coarser resolution in ice sheet areas.

We propose the following experiments to harness the strengths of WAOM.

– Deactivating tides in the model would lead to a first estimate of the impact of tides on Antarctic-wide ice shelf melting and can likely be used to gain further insights into the mechanisms governing tidal melt.

– Experiments that trace individual water masses, such as ISW or AASW, could be used to study the role of ice shelf teleconnections in a pan-Antarctic context or attribute ice shelf mass loss to the individual melting modes.

– Extending the resolution study introduced here would help to attribute the convergence behavior to individual aspects of the model. Future experiments should be designed to isolate effects due to changes in bathymetry, ice draft, tides and sub-grid scale turbulence parameterisation. This way, changes in shore-ward heat flux with increasing model resolution could be more clearly related to better representation of troughs, eddies and internal tides.

– To confirm our hypothesis that tidal mixing governs the reported cooling of the continental shelf ocean with increasing horizontal resolution, future studies should perform additional experiments without tides and apply heat flux analysis across the shelf break, surface and cavity entrance.

– Finally, applying anomalies from future climate projections to the boundary forcing (e.g. from CMIP5; Taylor et al., 2011) could be used to study the response of Antarctic ice shelf melting to warming oceans. This experiment would not just add another estimate that complements other model results by Naughten et al. (2018b), but offers valuable sample points of the average behaviour of the ice shelf cavity system.

# 5 Summary and Conclusion

Here, we present the Whole Antarctic Ocean Model (WAOM v.1.0). WAOM overcomes two major shortcomings of previous circum-Antarctic ocean-ice shelf models by the inclusion of tides and a horizontal resolution which is high enough to resolve critical shoreward heat transport by eddies (e.g. Dinniman et al., 2016). We have simulated present-day conditions by spinning up the model with repeated 2007-forcing. We have brought the model close to equilibrium at 2 km grid spacing.

WAOM qualitatively captures the broad scale difference between warm and cold regimes and many of the known characteristics of regional ice-ocean interaction. Continental shelf ocean temperatures and ice shelf melting converge with increasing model resolution, but a further refinement to 1 km grid spacing or finer is likely needed to reach asymptotic behaviour. The

accuracy of tidal height signals at the coast is comparable to barotropic tide models. The total ice shelf basal mass loss is close to, but 4 % below the lowest estimate derived from satellite observations. The basal mass balance of individual ice shelves agrees with satellite observations in many places, but indicates a cold bias for some warm water ice shelves in the Amundsen-Bellingshausen Seas as well as the Totten and Moscow-University Ice Shelf System. Ice shelf melting and marine ice accretion at high resolution are often in agreement with regional studies, demonstrating that our model captures the known modes of ice shelf-ocean interaction. The on-shelf hydrography resembles many aspects of WOA18 summer climatologies and decadal mean bottom layer temperatures by Schmidtko et al. (2014), but exhibits a lack of HSSW formation, a warm bias at the surface and excessive mixing. We hypothesize that the cold bias in the Amundsen-Bellingshausen Seas and the lack of HSSW is caused by overly mixed conditions on the continental shelf.

Future studies will need to evaluate and calibrate the model according to their specific research question. To improve the model's accuracy regarding ice shelf melting, the biases revealed here should be addressed first. Any further tuning will first require a compilation of available in-situ observations (from ApRES measurements). Such efforts are underway with the NECKLACE programme.

Resolving ice shelf-ocean interaction at high resolution is the main purpose of WAOM. The only available estimate of Antarctic-wide ice shelf basal melting at high resolution has been derived from satellite observations and models of surface processes with unknown uncertainty (Rignot et al., 2013). Thus, new estimates derived from a fully independent method, that also offers an ocean consistent to the melt rates, is likely to result in new insights into the governing processes that drive Antarctic ice shelf melting. Further, WAOM is well suited for giving a first estimate of circum-Antarctic tidal melting and to explore the average behaviour of all ice shelf cavity systems found around the continent. WAOM is not coupled to a dynamic sea ice model and, thus, future simulations will need estimates of sea ice-ocean fluxes from climate models. Alternatively, WAOM could be coupled to a sea ice model, in a manner similar to Naughten et al. (2018b).

To reduce uncertainties in predictions of future sea level rise and climate, models will ultimately need to resolve interaction between the Antarctic ice sheet and the Southern Ocean over glaciological timescales (e.g. Colleoni et al., 2018). Code that communicates the shared properties between ice sheet and ocean models is now available (Jong et al., 2017), and idealized and regional applications with ROMS show promising results (as discussed in Asay-Davis et al., 2017). WAOM has been designed to provide the ocean component of a coupled Antarctic-wide application and this study presented development and evaluation of a present day simulation and is a major step towards this goal.

*Code and data availability.* The model output can be obtained from the authors upon request. The source code and configuration files used for the simulations described here are archived at http://doi.org/10.5281/zenodo.3738985 (Richter, 2020a), while the maintained version is publicly available at https://github.com/kuechenrole/waom. The grid files, atmospheric forcing, initial conditions, and northern boundary conditions can be obtained from the authors upon request. The Python and Matlab scripts used to generate the grid and forcing files and to perform the analysis on the model output are archived at http://doi.org/10.5281/zenodo.3738998 (Richter, 2020b) and the maintained version of these scripts is publicly available at https://github.com/kuechenrole/antarctic_melting.

**Appendix A:  Antarctic Ice Shelf Melting From Observations and Models**

**Table A1.** Antarctic ice shelf basal mass loss from models and satellite observations. Results for this study have been calculated using ice shelf boundaries from the MEaSURES Antarctic boundaries dataset (Mouginot et al., 2016) or Bedmap2 (Fretwell et al., 2013). Other modelling studies are Hellmer (2004); Timmermann et al. (2012); Dinniman et al. (2015); Schodlok et al. (2016); Mathiot et al. (2017); Naughten et al. (2018b) and studies using satellite observations and models of surface processes are Rignot et al. (2013); Depoorter et al. (2013); Liu et al. (2015). Among the satellite studies, only Liu et al. (2015) avoids the assumption of a steady state calving front when inferring basal conditions (referred to as Liu 2015 (non steady); see Liu et al., 2015). Rignot 2013 (hr data) estimates have been calculated by integrating the high resolution solution from Rignot et al. (2013) and using MEaSURES ice shelf boundaries. Naughten et al. (2018b) and Mathiot et al. (2017) area average melt rates for individual ice shelves have been calculated using area definitions from Rignot et al. (2013). Abbreviations are melt rate ($w_b$) and Basal Mass Loss (BML).

| BML (Gt/yr), $w_b$ (m/yr), Area ($10^3$ km$^2$) | This study (MEaSURES) | This study (Bedmap2) | Rignot 2013 (IS flux) | error | Rignot 2013 (hr data) | Liu 2015 (non steady) | error | Liu 2015 (steady state) | error | Depoorter 2013 | error | Hellmer 2004 | Timmermann 2012 | Dinniman 2015 | Schodlok 2016 | Mathiot 2017 | Naughten 2018 (MetROMS) | Naughten 2018 (FESOM HR) |
|---|---|---|---|---|---|---|---|---|---|---|---|---|---|---|---|---|---|---|
| **Total Antarctica** BML | 973.34 | 1209.10 | 1500.00 | 237.00 | 1046.31 | 1516.00 | 106.00 | 1290.00 | 110.00 | 1454.00 | 174.00 | 906.60 | 1600.00 | 664.00 | 1735.00 | 1864.00 | 642.00 | 739.00 |
| $w_b$ | 0.70 | 0.82 | 0.85 | 0.10 | 0.75 | 1.10 | 0.10 | 0.90 | 0.10 | 0.94 | 0.11 | 0.80 | 1.20 | - | 1.16 | - | 0.52 | 0.56 |
| Area | 1523.33 | 1617.62 | 1561.40 | - | 1529.46 | 1542.11 | - | 1542.11 | - | 1555.00 | - | 1233.00 | 1510.00 | - | 1625.00 | - | 1349.00 | 1438.00 |
| **Amery** BML | 10.56 | 10.03 | 35.50 | 23.00 | 40.38 | 95.00 | 8.00 | 63.00 | 9.00 | 39.00 | 21.00 | 17.65 | 174.00 |  |  | 207.00 | 91.00 | 71.40 |
| $w_b$ | 0.19 | 0.18 | 0.58 | 0.40 | 0.77 | 1.70 | 0.10 | 1.10 | 0.20 | 0.65 | 0.35 | 0.35 | 2.90 | 1.10 | 1.25 | 3.73 | 1.64 | 1.29 |
| Area | 59.13 | 60.53 | 60.65 |  | 57.53 | 62.23 |  | 62.23 |  | 60.00 |  | 55.00 | 67.00 |  |  | 60.65 | 60.65 | 60.65 |
| **Totten & Moscow Uni.** BML | 13.26 | 18.39 | 90.60 | 8.00 | 67.50 | 122.00 | 12.00 | 90.00 | 12.00 | 92.00 | 19.00 |  |  |  |  |  | 9.50 | 9.30 |
| $w_b$ | 1.25 | 1.52 | 7.66 | 0.73 | 7.29 | 10.68 | 1.02 | 7.86 | 1.02 | 7.41 | 1.62 |  |  |  | 2.06 |  | 0.88 | 0.86 |
| Area | 11.60 | 13.23 | 11.83 |  | 10.10 | 12.55 |  | 12.55 |  | 12.00 |  |  |  |  |  |  | 11.83 | 11.83 |
| **Shackleton** BML | 17.12 | 23.63 | 72.60 | 15.00 | 55.21 | 96.00 | 7.00 | 60.00 | 7.00 | 76.00 | 23.00 | 47.68 |  |  |  | 14.00 | 14.30 | 17.40 |
| $w_b$ | 0.74 | 0.88 | 2.78 | 0.60 | 2.50 | 3.39 | 0.26 | 2.14 | 0.26 | 2.20 | 0.67 | 1.04 |  |  | 0.52 | 0.59 | 0.60 | 0.73 |
| Area | 25.26 | 29.21 | 26.08 |  | 24.12 | 31.00 |  | 31.04 |  | 35.00 |  | 50.00 |  |  |  | 26.08 | 26.08 | 26.08 |
| **Ross** BML | 66.48 | 69.10 | 47.70 | 34.00 | 27.50 | 27.00 | 22.00 | 71.00 | 25.00 | 34.00 | 25.00 | 180.20 | 260.00 |  |  | 111.00 | 53.80 | 112.00 |
| $w_b$ | 0.15 | 0.15 | 0.10 | 0.10 | 0.06 | 0.10 | 0.00 | 0.20 | 0.10 | 0.07 | 0.05 | 0.49 | 0.60 | 0.14 | 0.36 | 0.24 | 0.12 | 0.24 |
| Area | 490.40 | 493.61 | 500.81 |  | 485.39 | 497.78 |  | 497.81 |  | 477.00 |  | 401.00 | 475.00 |  |  | 500.81 | 500.81 | 500.81 |
| **Larsen C** BML | 9.08 | 13.12 | 20.70 | 67.00 | 5.46 | 55.00 | 7.00 | 6.00 | 8.00 | 18.00 | 8.00 | 38.13 | 48.00 |  |  | 46.00 | 18.20 | 54.70 |
| $w_b$ | 0.22 | 0.26 | 0.45 | 1.00 | 0.13 | 1.00 | 0.10 | 0.10 | 0.20 | 0.30 | 0.14 | 0.63 | 1.00 | 0.35 | 1.47 | 1.08 | 0.43 | 1.29 |
| Area | 45.62 | 55.66 | 46.47 |  | 44.39 | 57.04 |  | 57.04 |  | 60.00 |  | 66.00 | 52.00 |  |  | 46.47 | 46.47 | 46.47 |
| **Filchner-Ronne** BML | 90.25 | 87.09 | 155.40 | 43.00 | 127.66 | 10.00 | 32.00 | 82.00 | 37.00 | 50.00 | 40.00 | 119.70 | 138.00 |  |  | 123.00 | 46.00 | 115.40 |
| $w_b$ | 0.23 | 0.22 | 0.32 | 0.10 | 0.33 | 0.00 | 0.10 | 0.20 | 0.10 | 0.12 | 0.09 | 0.32 | 0.35 | 0.19 | 0.25 | 0.30 | 0.11 | 0.28 |
| Area | 433.32 | 433.95 | 443.14 |  | 426.51 | 426.35 |  | 426.35 |  | 423.00 |  | 408.00 | 438.00 |  |  | 443.14 | 443.14 | 443.14 |
| **Brunt + Riiser-Larsen** BML | 64.53 | 68.45 | 9.70 | 14.00 | 11.05 | 20.00 | 10.00 | 37.10 | 10.00 | 26.00 | 16.00 | 165.90 | 65.00 |  |  | 39.00 | 29.20 | 34.60 |
| $w_b$ | 0.89 | 0.93 | 0.12 | 0.20 | 0.16 | 0.25 | 0.12 | 0.50 | 0.14 | 0.33 | 0.20 | 2.38 | 0.94 | 0.67 | 0.28 | 0.53 | 0.40 | 0.47 |
| Area | 78.86 | 80.65 | 80.34 |  | 76.66 | 81.84 |  | 81.84 |  | 79.00 |  | 76.00 | 77.00 |  |  | 80.34 | 80.34 | 80.34 |
| **Fimbulisen & Jelbart** BML | 62.94 | 67.39 | 22.50 | 12.00 | 14.31 | 55.10 | 11.00 | 46.00 | 11.00 | 24.00 | 13.00 | 243.10 | 130.00 |  |  | 39.00 | 30.30 | 52.40 |
| $w_b$ | 1.36 | 1.41 | 0.43 | 0.22 | 0.32 | 0.75 | 0.12 | 0.62 | 0.15 | 0.52 | 0.27 | 4.91 | 2.80 | 1.51 | 1.16 | 0.82 | 0.64 | 1.11 |
| Area | 50.54 | 52.20 | 51.69 |  | 48.81 | 78.06 |  | 78.06 |  | 46.00 |  | 54.00 | 53.00 |  |  | 51.69 | 51.69 | 51.69 |
| **Getz** BML | 60.12 | 67.13 | 144.90 | 13.00 | 120.91 | 149.00 | 16.00 | 96.00 | 16.00 | 136.00 | 23.00 | 53.64 | 164.00 |  |  | 337.00 | 88.10 | 30.60 |
| $w_b$ | 1.99 | 2.04 | 4.26 | 0.40 | 4.22 | 4.80 | 0.50 | 3.10 | 0.50 | 4.09 | 0.68 | 1.95 | 5.40 | 0.66 | 11.05 | 10.81 | 2.83 | 0.98 |
| Area | 33.05 | 35.86 | 34.02 |  | 31.31 | 33.74 |  | 33.74 |  | 33.00 |  | 30.00 | 35.00 |  |  | 34.02 | 34.02 | 34.02 |
| **Pine Island** BML | 36.65 | 41.18 | 101.20 | 8.00 | 85.96 | 78.00 | 8.00 | 51.00 | 8.00 | 95.00 | 14.00 |  | 13.00 |  |  | 87.00 | 20.50 | 9.50 |
| $w_b$ | 7.03 | 7.28 | 16.20 | 0.10 | 17.75 | 14.00 | 1.40 | 9.10 | 1.40 | 15.96 | 2.38 |  | 3.10 | 1.62 | 14.91 | 15.20 | 3.58 | 1.66 |
| Area | 5.69 | 6.18 | 6.25 |  | 5.29 | 6.09 |  | 6.00 |  | 6.00 |  |  | 5.00 |  |  | 6.25 | 6.25 | 6.25 |
| **Abbot** BML | 73.63 | 80.93 | 51.80 | 20.00 | 38.56 | 35.00 | 7.00 | 46.00 | 7.00 | 86.00 | 22.00 | 18.60 | 59.00 |  |  | 52.00 | 25.00 | 36.60 |
| $w_b$ | 2.60 | 2.65 | 1.75 | 0.60 | 1.47 | 1.20 | 0.20 | 1.60 | 0.20 | 2.72 | 0.70 | 0.55 | 2.10 | 0.34 | 1.23 | 1.91 | 0.92 | 1.35 |
| Area | 30.90 | 33.37 | 29.69 |  | 28.70 | 32.51 |  | 32.51 |  | 32.00 |  | 36.00 | 32.50 |  |  | 29.69 | 29.69 | 29.69 |
| **George VI** BML | 145.99 | 152.31 | 89.00 | 17.00 | 63.20 | 76.00 | 13.00 | 56.00 | 13.00 | 144.00 | 42.00 | 22.48 | 86.00 |  |  | 298.00 | 48.40 | 32.50 |
| $w_b$ | 6.99 | 7.06 | 3.80 | 0.70 | 3.20 | 4.02 | 0.73 | 2.94 | 0.67 | 2.88 | 0.83 | 0.43 | 3.60 | 1.19 | 7.99 | 13.88 | 2.25 | 1.51 |
| Area | 22.79 | 23.50 | 23.43 |  | 21.56 | 20.59 |  | 20.59 |  | 50.00 |  | 57.00 | 27.00 |  |  | 23.43 | 23.43 | 23.43 |

* Liu et al. (2015) also includes Vigrid, Nivil, Lazarev, Borchgrevink.

** Depoorter et al. (2013) also includes Larsen B.

*** Depoorter et al. (2013) and Hellmer (2004) includes Wilkens and Stange.

## Appendix B: Computational Ice Shelf Masks

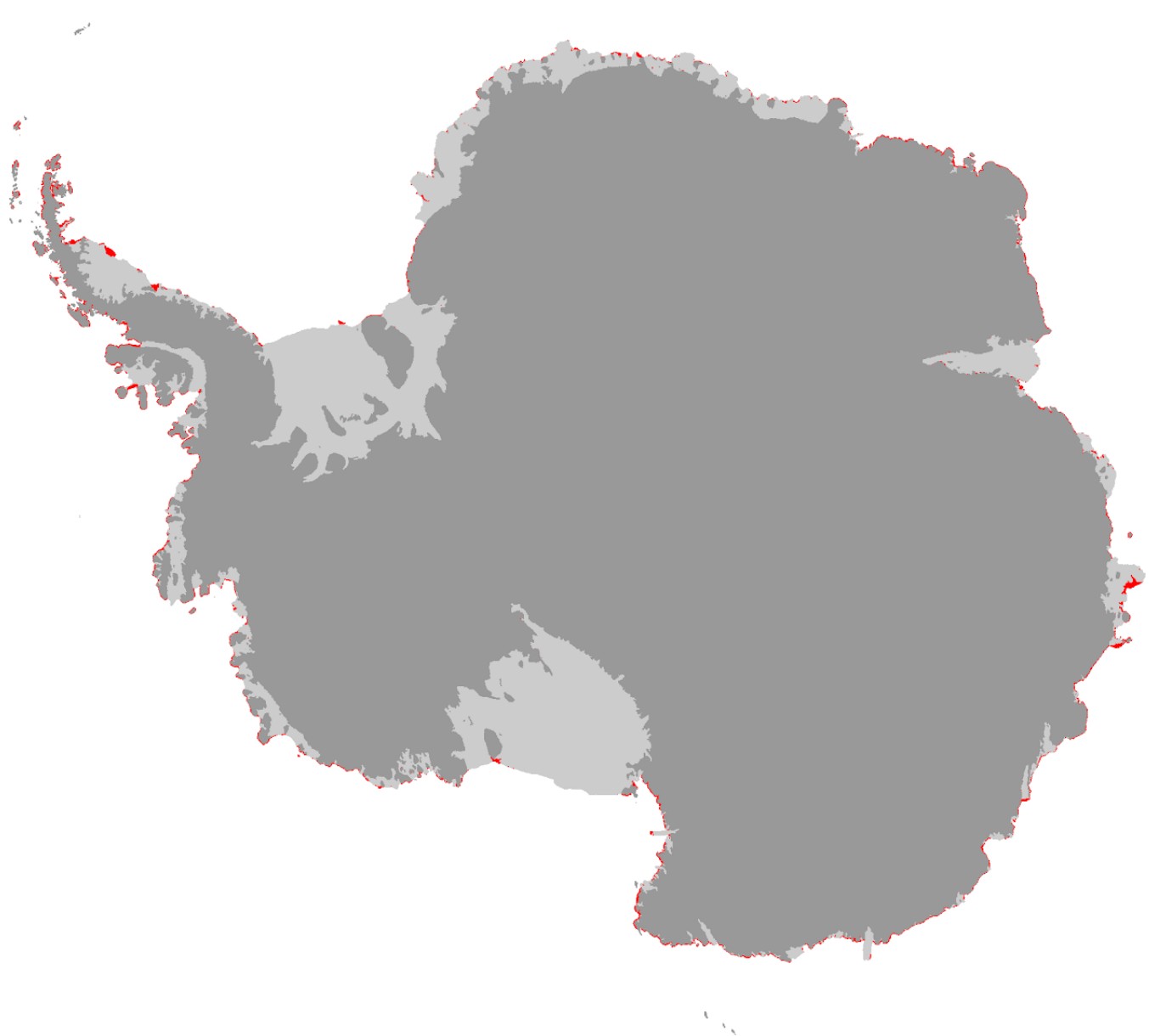

**Figure B1.** Difference in ice area definition. Red areas show ice that is excluded when imposing ice shelf boundaries from the MEaSURES Antarctic Boundaries data set (Mouginot et al., 2016) onto the ice draft from Bedmap2 (Fretwell et al., 2013).

**Appendix C: Model Configuration**

**Table C1.** Activated ROMS options in WAOM.

| Category | CPP option | Description |
| --- | --- | --- |
| Momentum equations | UV_COR | Coriolis term |
| | UV_VIS2 | harmonic horizontal mixing |
| | UV_QDRAG | quadratic bottom friction |
| | UV_ADV | advection terms |
| | MIX_S_UV | mixing along constant S-surfaces |
| | SPLINES_VVISC | splines reconstruction of vertical viscosity |
| pressure gradient | DJ_GRADPS | splines density Jacobian |
| Tracer equations | TS_A4HADVECTION | 4th-order Akima horizontal advection |
| | TS_A4VADVECTION | 4th-order Akima vertical advection |
| | TS_DIF2 | harmonic horizontal mixing |
| | SALINITY | having salinity |
| | MIX_ISO_TS | mixing on epineutral (constant RHO) surfaces |
| | NONLIN_EOS | nonlinear equation of state |
| | QCORRECTION | net heat flux correction |
| | SCORRECTION | freshwater flux correction |
| | SURFACE_OVERFLUX_FIX | corrections for not having a sea ice model |
| Vertical mixing | LMD_MIXING | Large et al. (1994) interior closure |
| | LMD_CONVEC | add convective mixing due to shear instability |
| | RI_SPLINES | splines reconstruction for Richardson Number |
| | LMD_DDMIX | double-diffusive mixing |
| | LMD_RIMIX | add diffusivity due to shear instability |
| | LMD_SKPP | surface boundary layer KPP mixing |
| | LMD_BKPP | bottom boundary layer KPP mixing |
| | LMD_NONLOCAL | nonlocal transport |
| | LMD_SHAPIRO | shapiro filtering boundary layer depth |
| Bottom stress | LIMIT_BSTRESS | limit the magnitude of bottom stress |
| Model configuration | SOLVE3D | 3D primitive equations |
| | CURVGRID | curvilinear coordinates grid |
| | SPHERICAL | spherical grid |
| | AVERAGES | writing out NLM time-averaged data |
| | MASKING | land/sea masking |
| Analytical fields | ANA_BSFLUX | analytical bottom salinity flux |
| | ANA_BTFLUX | analytical bottom temperature flux |
| | ANA_SRFLUX | analytical surface shortwave radiation flux |
| | SPLINES_VDIFF | splines reconstruction of vertical diffusion |
| Ice shelf | ICESHELF | including ice shelf cavities |
| | LIMIT_ICESTRESS | limit the magnitude of ice shelf basal stress |
| | ICESHELF_3EQN_VBC | activate 3-equation ice/ocean thermodynamics |
| Tides | SSH_TIDES | imposing tidal elevation |
| | ADD_FSOBC | add tidal elevation to processed OBC data |
| | UV_TIDES | imposing tidal currents |
| | ADD_M2OBC | add tidal currents to processed OBC data |
| | RAMP_TIDES | ramping (over one day) tidal forcing |
| NetCDF input/output | PERFECT_RESTART | include perfect restart variables |

**Table C2.** Some key model parameters.

| Parameter | value<br>(10/4/2 km resolution) |
|---|:---:|
| Vertical resolution ( # layers) | 31 |
| Vertical coordinate transformation equation # | 2 |
| Vertical coordinate transformation stretching function # | 4 |
| Surface stretching parameter | 7 |
| Bottom stretching parameter | 8 |
| Critical depth (m) | 250 |
| Baroclinic timestep (s) | 900/360/180 |
| Barotropic timestep (s) | 25/10/5 |
| Horizontal diffusivity $(\mathrm{m^2 s^{-1}})$ | 50/20/10 |
| Horizontal viscosity $(\mathrm{m^2 s^{-1}})$ | 500/200/100 |
| Relaxation time scale for tracers at the surface (days) | 365 |
| Relaxation time scale for ocean elevation at the surface (days) | 3 |
| Relaxation time scale for barotropic momentum at the open boundary (days) | 3 |
| Relaxation time scale for baroclinic momentum at the open boundary (days) | 3 |
| Open boundary outflow/inflow nudging factor | 365 |

## Appendix D: Additional Ocean Evaluation

**DATA DISTRIBUTION PLOT:**

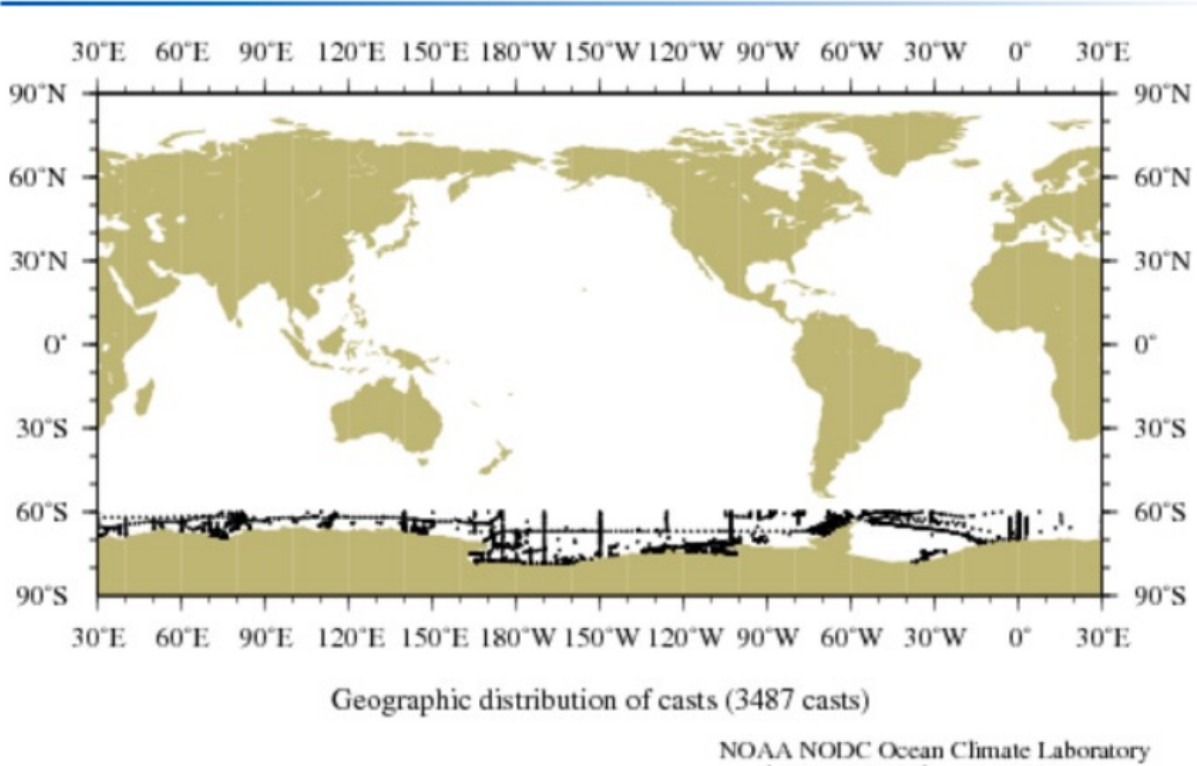

Geographic distribution of casts (3487 casts)

NOAA NODC Ocean Climate Laboratory
http://www.nodc.noaa.gov/OCL/

**COPY OF YOUR SEARCH CRITERIA:**

**DEEPEST MEASUREMENTS:>**       **400**
**OBSERVATION DATES:**           **Year** from 2005 to 2017
**GEOGRAPHIC COORDINATES:**      **Longitude** from -180.0000 to 180.0000; **Latitude** from -60.0000 to -80.0000
**DATASET:**                     CTD,UOR
**MEASURED VARIABLES (extract):** Temperature, Salinity

**Figure D1.** Sampling distribution underlying WOA18 data. Only CTD casks that reached a depth below 400 m and measured both, Temperature and Salinity, are shown. The distribution clearly shows a high sampling density along summer ship tracks (e.g. along longitudes: 170°W, 150°W, 102°W, 40°E, 60°E, 70°E and 175°E) and on the Amundsen Sea continental shelf. The figure has been produced using the World Ocean Database Search Query web application: https://www.nodc.noaa.gov/OC5/SELECT/dbsearch/dbsearch.html.

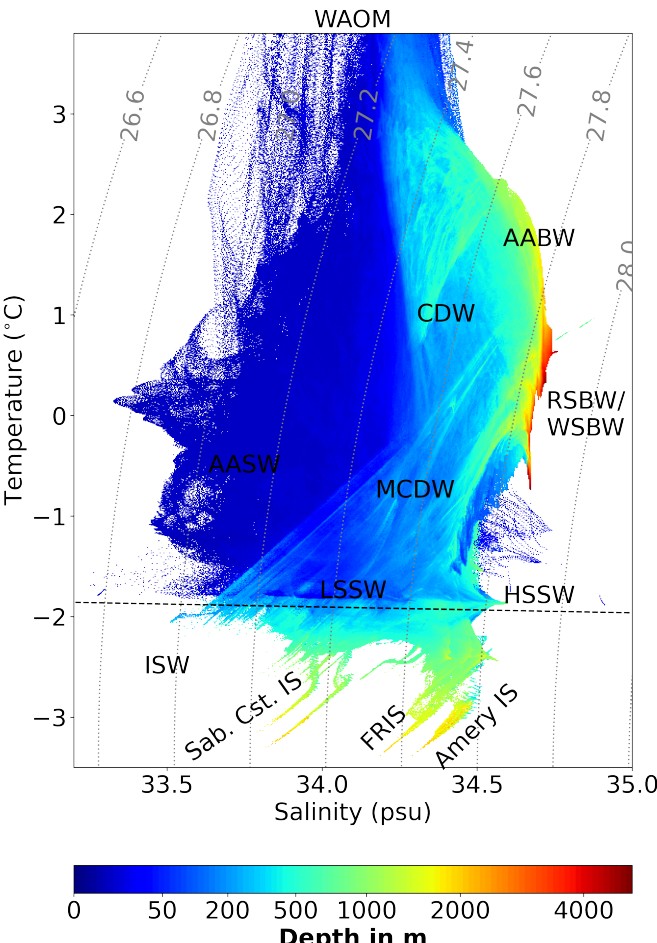

**Figure D2.** WAOM's water masses. Shown are the 2007 mean Potential Temperature-Salinity-Depth distributions south of 65°S (off-shelf, on-shelf and sub-ice shelf cavities). For the analysis, each grid cell has been sorted into 1000x1000 temperature and salinity bins and the depth shown for each bin is the volume-weighted average of all the grid cells in this bin. The dashed black lines show the freezing point at the surface and the dotted grey lines are potential density anomaly contours (in km m -3 -1000; referenced to the surface). Horizontal labels show different water masses: CDW indicates Circumpolar Deep Water, MCDW indicates Modified Circumpolar Deep Water, LSSW indicates low-salinity shelf water, HSSW indicates high-salinity shelf water, AASW indicates Antarctic Surface Water, ISW indicates Ice-Shelf Water, AABW indicates Antarctic Bottom Water, and WSBW/RSBW indicates Weddell/Ross Sea Bottom Water. Rotated labels show source region. Abbreviations are Ice Shelf (IS), Filchner-Ronne (FR) and Sabrina Coast (Sab. Cst.).

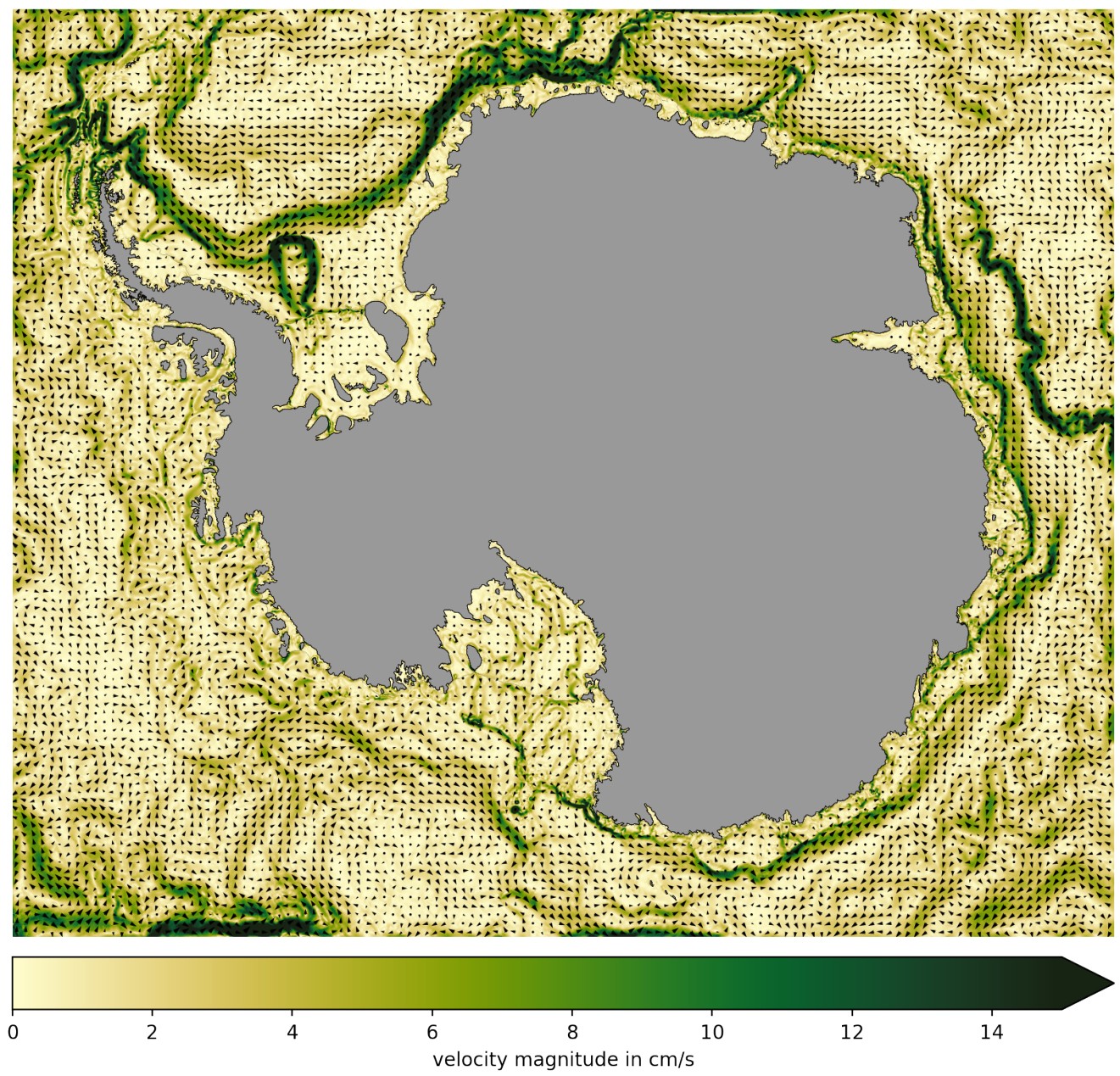

**Figure D3.** Mean barotropic currents in WAOM. Colors denote 2007 mean barotropic current velocity magnitude and arrows indicate direction. WAOM reproduces known features, such as the southern limb of the ACC around the Kerguelen Plateau, the southern limbs of the Ross and Weddell Sea Gyres, the slope current (e.g. around East Antarctica) and coastal currents (apparent in, e.g. Prydz Bay and in front of the Totten Ice Shelf). However, some boundary effects are apparent in the Eastern Ross Sea.

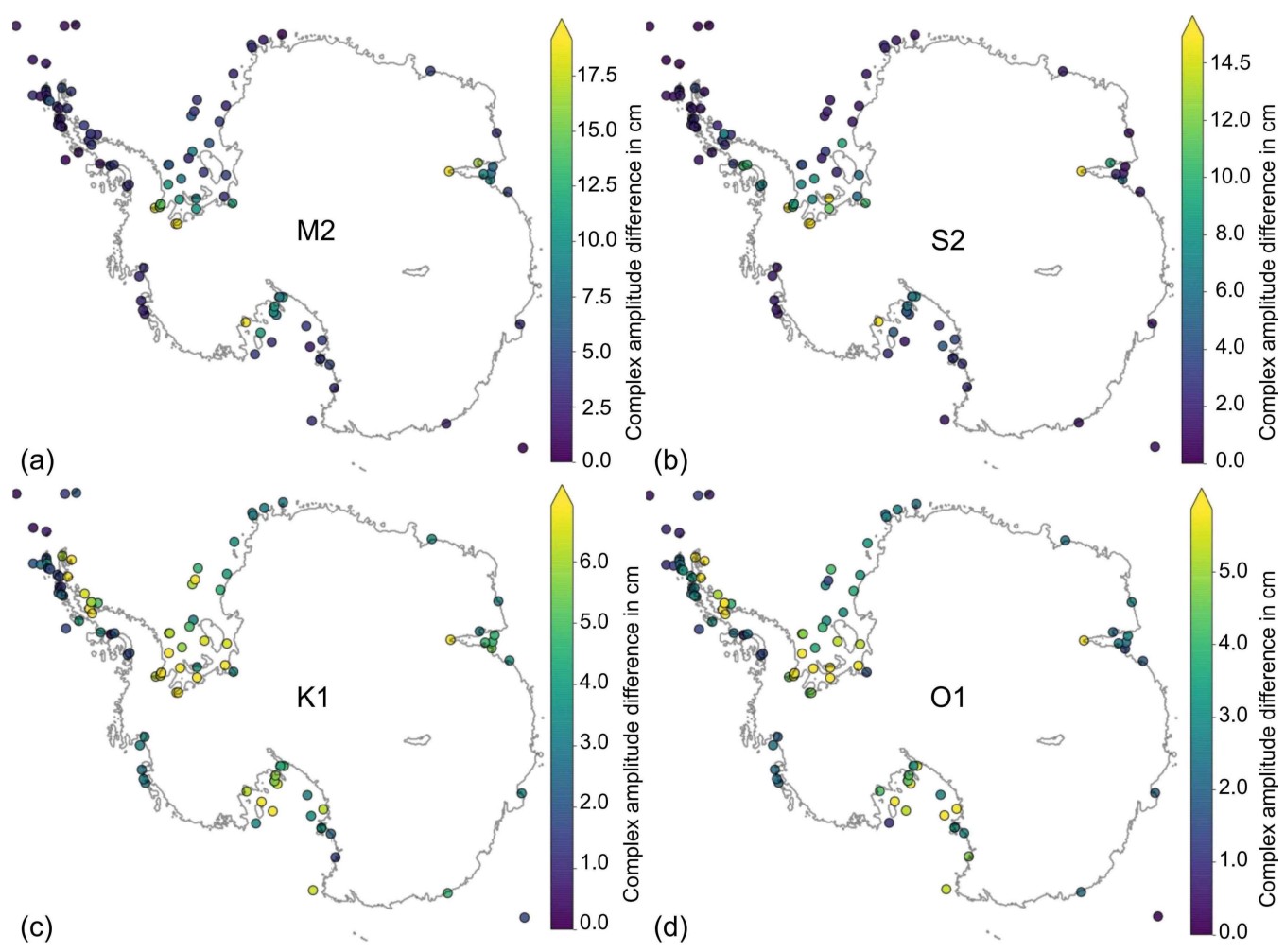

**Figure D4.** Spatial distributions of tidal height accuracy. Complex amplitude differences between the model solution and Antarctic Tide Gauge records are shown for the major tidal constituents (a) M2, (b) S2, (c) K1 and (d) O1. The largest biases occur at the deep grounding lines of the large ice shelves, where the water column thickness is uncertain.

*Author contributions.* Ole Richter and Benjamin K. Galton-Fenzi conceived and designed the experiments. Ole Richter performed the simu-
lations. Ole Richter, Benjamin K. Galton-Fenzi and David E. Gwyther analysed the data, whereby Kaitlin A. Naughten contributed analysis
tools. Ole Richter prepared the manuscript with contributions from all co-authors.

*Competing interests.* The authors declare that they have no conflict of interest.

*Acknowledgements.* This research was supported by scholarships from the Australian Government and the Australian Research Council's
Special Research Initiative for the Antarctic Gateway Partnership SRI40300001. Computational resources were provided by the NCI National
Facility at the Australian National University, through awards under the Merit Allocation Scheme. We would like to thank Eric Rignot and
Jeremie Mouginot for providing us with the satellite derived melt rates. We are also grateful to Richard Coleman for his valuable comments
on the manuscript and Just Berkhout for his excellent IT support.

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
