# Peer review of "The Whole Antarctic Ocean Model (WAOM v1.0): Development and Evaluation"

_Geoscientific Model Development, 2020_

## Referee Comment (RC1) · Anonymous Referee #1 · 23 Jul 2020

In this manuscript the authors describe a new regional model configuration (WAOM) that encompasses the entire Antarctic continental margins, including the cavities beneath Antarctica's floating glaciers. The scientific focus of the model is on simulating ocean-driven melt of Antarctica's ice shelves. The manuscript describes the set-up and integration of the model at three different resolutions (10km, 4km and 2km grid spacing), the highest of which is ostensibly capable of fully resolving tidal and fine-scale processes that contribute to the circulation and stratification on the continental shelf. The authors evaluate WAOM using an ocean state estimate and independent modeled/measured estimates of Antarctica's ice shelf melt rates, focusing on the highest-resolution solution. They concluded that WAOM acceptably reproduces the observed ocean stratification and ice shelf melt rates, and therefore is a suitable tool for addressing scientific questions related to mechanisms of ocean-driven melt. They discuss shortcomings and sources of biases in the model, particularly emphasizing WAOM's lack of an active sea ice component, and discuss future development and scientific goals for the model.

My high-level evaluation is that this is a significant model development that is worthy of publication in GMD. As the authors note, this is the first ocean model with (borderline) tidal- and eddy-resolving resolution that includes all of Antarctica's ice shelves, and therefore offers insights into the role of these processes in modulating Antarctic ice loss at the continental scale. The description of WAOM is appropriate for a model definition manuscript, and configuration and analysis scripts are provided (remotely) to allow for complete reproducibility. The manuscript and figures are very clearly composed.

Below I have provided a series of comments and suggestions on the manuscript for the authors' consideration. My most significant concern pertains to their validation of the ocean state: 1. Offshore, the model is evaluated against the Southern Ocean State Estimate (SOSE), which contains biases of its own, particularly close to the Antarctic margins (see e.g. Dotto et al., 2014, Ocean Sci.). It was unclear to me why the authors chose to SOSE rather than directly against measurements. 2. On the continental shelf the authors perform only a qualitative evaluation of the ocean state, rather than performing direct comparisons. I was surprised by this, because the ocean state on the shelf is critical in determining the circulation and melt rates in the ice shelf cavities. Based on this, I would argue that the hydrography on the continental shelf should be the most closely scrutinized aspect of the model state. Previous studies have compiled measurement from the continental shelf from all around Antarctica in order to compute trends in shelf properties (Schmidtko et al., 2014, Science), characterize different dynamical regimes on the continental shelf (e.g. Amblas and Dowdeswell, 2018, Earth Sci. Rev.) and evaluate models (e.g. Morrison et al., 2020, Sci. Adv.). In my opinion, the manuscript would be strengthened significantly if the authors used one of these datasets to map biases in shelf properties in WAOM.

Comments/questions:

L4-5: How significant is lack of interannual variability? Is it possible that some of the biases in the modeled shelf stratification and melt rates occur because the model excludes anomalous years, e.g. years with particularly strong/weak winds or warm/cool atmospheric temperatures?

L61: "beyond the scope of this study" doesn't really mean anything. If it's not in the study then it's beyond the scope by definition. It would be more helpful to state why an evaluation matrix was not pursued.

Also is model tuning really "rigorous"? A right answer for the wrong reasons is not necessarily better than a slightly wrong answer based on well-grounded physical parameter choices.

L93: Horizontal grid sizes in the form M x N would be more relatable than total number of computational cells.

L98: Are the authors referring to the scheme of Shchepetkin and McWilliams (2003)?

L100, L304-311: How wide does the "vertical cliff" at the ice shelf face become with this smoothing? Does this bias the model toward more Mode 3 ice shelf melt?

L102: Is the algorithm applied to both the ice shelf draft and the bathymetry, or only to the water column thickness?

L105: Could the authors elaborate on "one of the smallest modifications possible"? I don't understand why there would be a hard limit on the ice shelf thickness - only an increasingly severe time step constraint as the thickness approaches zero.

L115-118: I presume that the authors impose heat and salt fluxes, rather than just buoyancy fluxes.

L116, L399-400: Imposing surface wind stresses directly with no accounting for sea ice is a significant caveat. For example, recent Arctic-focused work shows that sea

ice plays a significant role in modulating the stresses felt at the ocean surface, and the ocean surface currents (see Meneghello et al. 2018, GRL and other subsequent papers from John Marshall's group).

L117-118: Please explain why not using a sea ice model is more likely to capture polynyas? This is counter-intuitive.

L119-122: The authors have quite a few rather ad hoc changes to the surface forcing that warrant further explanation. Why do the authors reduce the positive heat flux into the ocean by half? Is this to simulate the sea ice albedo effect? I understand the physical motivation for changing the brine rejection and restoring surface temperatures from below freezing, but it seems inconsistent to do so when all other fluxes are fixed. How do the authors gauge that one month is a "long" time scale for surface relaxation?

L125-126: I am skeptical about the claim that the model state at the boundary is primarily dictated by the interior. Surely this is not a desired outcome, as remotely-formed water masses (especially CDW) need to be supplied by the open boundary conditions.

Fig. 2: It looks like the melt rate drops instantaneously upon re-initialization with a 4km grid spacing, and again with a 2km grid spacing. Is there some geometrical impact of the grid refinement that causes this, or is the adjustment time scale just shorter than the monthly frequency of the model output that was used to create the plot?

Eq. (1): Shouldn't the denominator be 1/N. Also, if $Z_j^m$ and $Z_j^o$ are complex variables (Z is the complex amplitude), then shouldn't the complex magnitude be taken before squaring (or equivalently multiplication by complex conjugate).

Eq. (2): Shouldn't the denominator be 1/(4N)? Also, shouldn't the $Z_j^m$ and $Z_j^o$ be indexed by k as well, to distinguish tidal components.

L175-176: Is a 2005 tide model still considered state-of-the art?

Table 2: I think I understand how the stated quantities (e.g. RMSD phase in deg) are related to equations (1) and (2), but the naming convention and formulation of

equations (1) and (2) make this much less clear than it could be.

L190-191: Should we expect the model state to asymptote under grid refinement? After all, various model parameters (e.g. viscosities/diffusivities) implicitly change with the grid, as does the bathymetry.

Fig. 4: Is 10km->4km grid spacing a 250% increase in resolution? If the resolution is defined as the number of grid points per km, then the resolutions are 0.1, 0.25 and 0.5 km^-1. So 10km->4km->2km grid spacings correspond to resolutions of 0.1->0.25->0.5 km^-1, so the resolutions have increased by 0%, 150% and 400% relative to the 10km grid. Or the resolutions have been multiplied by 100%, 250% and 500% relative to the 10km grid. Please pick a consistent convention!

L215-222: I was initially confused by the authors' explanation of the RSBW and WSBW salinities, which the attribute to the (ECCO2-sourced) open boundary conditions. I would have hoped that water masses formed within the domain would not depend on the boundary conditions. That said, the model boundary cuts right through the Weddell Gyre, so perhaps it is reasonable to have inflow of some bottom waters. Perhaps the authors could clarify this in the text?

I am very confused by the waters with salinities reaching 34.8 in Fig. 6. There is no other water anywhere in the model domain with such high salinities, and these high salinities are only found at great depth, far from the surface. So, what is the source of the very salty bottom waters?

Fig. 6: I suggest adding more lines/arrows to indicate water mass locations more precisely. The CDW label looks to be much too fresh (e.g. compare with Fig. 7), and AABW is labeled at a lower density than CDW!

Also, I presume the portions of T/S space labeled "Weddell Sea" and "Ross Sea" are actually the Filchner-Ronne Ice Shelf cavity and the Ross Ice Shelf cavity, respectively.

L235: Are the transects all annually-averaged? Are there significant deviations in the

agreement between SOSE and WAOM in different seasons?

Fig. 7: Why does the grid spacing in WAOM appear to be so coarse? The model grid spacing is 2km but the data points in this figure appear to be spaced 50-100km apart.

Fig. 8: Given that the authors have just compared a few transects here, why not align the transects with WOCE transects? Then they could include a third column of panels showing the WOCE measurements for additional reference.

Also, often these transects are visually very similar, especially with these colormaps. Difference plots would be more revealing with regard to the model biases.

L421: Has the model equilibrated in the higher-resolution cases? I don't see how the authors can judge this from just a one-year time series in the 2km simulation. I think it would be fairer to say that the model has equilibrated at 10km grid spacing, and has been continued from that equilibrated state at higher resolution.

L433: It's really the "surface" stress that is uncertain: the wind stress is relatively well constrained by reanalyses, whereas the ice-ocean stress is much less well constrained.

Table C2: Please specify which relaxation parameters pertain to the open boundaries vs the ocean surface.

---

## Referee Comment (RC2) · Anonymous Referee #2 · 4 Aug 2020

This is an interesting manuscript that describes some of the more technical aspects of a new circum-Antarctic configuration of ROMS. The novel features are that it includes sub-ice-shelf cavities and tidal forcing, and it is run at high resolution. The subject matter certainly fits the remit of GMD and I think that it could make a valuable contribution to the growing literature on modelling the ocean circulation beneath the Antarctic ice shelves. However, I have a number of concerns with the way the model has been set up and validated, and I think the authors should clarify the reasons for the approach they have taken and provide a more critical evaluation of their results.

Questions: 1) While terrain-following coordinates have a number of advantages, their performance over steep topography can be an issue. The authors apparently deal with this critical point in lines 95-106, but nowhere do they state the extent to which the

topography of the ice front and continental slope have been modified by smoothing, or the magnitude of any residual pressure-gradient errors that they would expect after the smoothing process. This is an important point, because in Figure 7 there are some very obvious artefacts in the model results that appear to be associated with topography. Could similar artefacts be affecting the properties and circulation at the continental shelf edge and at the ice fronts? If so, they could have a major impact on the results that have been presented.

2) The surface forcing is unconventional, in that heat and freshwater fluxes are applied rather than derived from atmospheric variables driving a physical parameterisation of the atmosphere-ice-ocean exchange. This has the advantage of removing the need for a sea ice model that can introduce biases, but has the potential to introduce its own biases. Presumably that is the motivation for the rather ad hoc adjustments made to the fluxes, described in lines 119-124? Those adjustments really should be more carefully described and motivated. Furthermore the use of wind stress without a dynamic sea ice model is questionable. Wouldn't it be more consistent to derive ocean surface stress from ice motion observations? While wind stress may be a reasonable proxy for surface stress where the ice is in free drift, free drift is a poor assumption in many of the regions of interest in this study, particularly the Weddell Sea and almost all near-coastal regions.

3) It is not obvious why the model has been forced with one year of data repeated, especially when that one year is characterised by "a paucity of observational data" (lines 10-11). Why not choose a year with more data? On lines 130-132 the authors state that the strategy of using one year allows them to integrate the model to quasi-equilibrium, but does it really reach that state in only 8 years? I think the authors should show some further diagnostics, such as domain-averaged temperature, salinity, KE, etc, to justify that statement.

4) It is also not clear to me why the authors have used one model (ECCO) to supply initial and open boundary conditions, then compared the results to a second model

(SOSE). Comparing with observations (however limited) would give some kind of indication as to how realistic the results are, while comparing to the same model (ECCO) would inform us about how the differing model architecture and surface boundary forcing impacted the model evolution. Without knowing how ECCO and SOSE differ, it is not clear what conclusions can be drawn from the comparisons that are made between WAOM and SOSE. The problem is highlighted by Figure 6, where WAOM and SOSE look completely different, particularly at depth. Is that because ECCO and SOSE are very different, or are the differences produced by the surface forcing? If it is the latter, how can the deep ocean have been so extensively modified in only a few years? It tends to suggest that the model is much too prone to deep convection, which is quite a common problem in Southern Ocean models. But does that happen everywhere, or is that all a product of those model artefacts that appear over steep topography (see 1 above)?

5) Figure 6 raises some other serious questions about the results. The ISW and RSBW/WSBW mixing lines appear to point to a water mass that apparently doesn't exist (HSSW). That suggests that they were formed from HSSW that was prescribed in the initial conditions, but has since been used up and not renewed. If that is the case, it points towards a model that is not in a quasi-equilibrium state (question 3 above), but is still in the process of evolving from initial conditions to some other state. Similarly the water masses at 1000 m depth are much too warm, suggesting that the original CDW prescribed in the initial conditions has also vanished and has been replaced by something quite different. That suggests some issue with the surface fluxes (question 2 above). It also suggests (again) that a longer integration would see the model continuing to evolve to a different state. If deep waters as warm as 3-4 deg C found their way onto the continental shelf, ice shelf melt rates would be much higher.

6) But perhaps the error is just in the plotting of Figure 6. The deep ocean stratification suggested by the trajectory of points from RSBW/WSBW to the (poorly placed) AABW label appears to be much too strong. I find it hard to believe that the whole domain

can have been so extensively altered in 8 years of integration (assuming the initial conditions taken from ECCO looked something like the SOSE results). Has some error been made in converting temperature to potential temperature, or density to potential density?

7) The problems with the water mass structure apparent in Figure 6 are also seen in Figures 7 & 8. On line 234 the authors state that "The stratification of WAOM agrees well with SOSE for the off-shelf ocean", but I would disagree with that. In most transects the main pycnocline is not well represented, either in strength or depth, and the mid-depth salinity maximum appears to be absent (worryingly consistent with Figure 6). While, this is far from being the only model to get the stratification wrong (it is notoriously hard to get right), I think the results presented here warrant more than the glib statement that they agree well with SOSE. In Figure 8 the absence of a source water mass for all the colder forms of ISW is again apparent, while ISW seems much too prevalent in the Amundsen and Bellingshausen seas, where it is hardly ever observed. Cooling waters too efficiently in the cavities hints at a problem with the balance of heat fluxes into the boundary layer (KPP) and at the ice-ocean interface. But again, some of these issues might arise because of the plotting, which makes the stratification look very odd. The densest waters are shaded blue (Figure 8), implying they are at the surface. How is that possible?

8) Throughout the paper there as in implication that higher resolution is intrinsically-better, but improvement in the model results with increasing resolution is never actually demonstrated. The implication of the discussion around Figure 4 (lines 184-191) is that at higher resolution still, shelf water temperatures and melt rates would drop further. However, at 2 deg resolution the mean melt rate has already dropped below the observed value and the 4 deg resolution simulation is arguably the best according to that single metric. On a related note, I don't understand what the authors mean by "convergence" in that discussion. This term normally refers to the ability of a numerical code to reproduce an analytical solution as the grid size tends to zero. What solution should

the model "converge" to in this case? The authors describe some features, cooling of the Bellingshausen Sea (line 196), for example, that appear to be worse in the higher resolution runs.

9) Again in Figure 10, it is not entirely obvious what has been gained by the addition of tides and the use of high resolution. Results are different from previous studies, but not obviously any better. The spatial distribution of re-freezing is rather poorly captured (Figure 11): almost nothing on Amery and Larsen C ice shelves, and very little in the central Ronne Ice Shelf. Many of the early (admittedly) regional models did much better, despite being run at much lower resolution. That again points to the fact that increasing resolution does not necessarily improve results. I agree that a higher resolution model has the potential to improve the representation of reality, because it can resolve more processes, but no model can resolve every process, and the key to getting things right at any resolution is knowledge of how best to parameterise whatever remains in the sub-grid-scale. Arguably the authors have done a better job at that with the 4 km version of WAOM than with the 2 km.

In summary, while this circum-Antarctic model has the potential to be a useful addition to the growing collection of such tools, I feel the authors should do more to critically evaluate their results and explain the impact of including extra processes and using finer resolution. The main issues at present that make the model results questionable for the applications that the authors have in mind are the problems with water column structure (that may be related to sigma-coordinate problems over steep topography) and the curious water mass properties (Figure 6). If the latter are not due to mis-plotting, then it suggests some serious issues with the model (potentially associated with the surface flux forcing and/or the sigma coordinates). Those really need to be sorted out before the model can be considered fit for purpose.

---

## Referee Comment (RC3) · Ralph Timmermann (Referee) · 21 Aug 2020

Reviewer's comment on

"The Whole Antarctic Ocean Model (WAOM v1.0): Development and Evaluation"

by O. Richter and coworkers

submitted to Geoscientific Model Development (Discussions)

**General comments**

In this paper, the authors report on the development of a new, high-resolution model of the Southern Ocean including its ice shelf cavities. The inclusion of tides represents a major feature of this model and a significant progress in scientific model development. The paper discusses model design und the evaluation of results.

The paper is well written and presents a lot of useful information. Figures are clear and well crafted. I recommend to accept the paper pending revisions guided by the following specific comments. Note that numbers 12 and 19 are a bit more substantial.

**Specific comments**

1. Throughout the text, I felt the urge to add a significant amount of hyphens in composite terms like "eddy-scale circulation", "large-scale models", "present-day conditions", "eddy-resolving horizontal resolution", "nearest-neighbour method", „Spin-up procedure", „depth-averaged temperature" etc. I trust this will be handled by the copyeditor at one of the final stages of publication, but I also encourage the authors to revisit these composits.

2. l. 23: To the list of papers trying to predict future changes, you may want to add Timmermann and Hellmer (2013).

3. l. 27: It may be worth mentioning that several coupled ice sheet--ocean models exist already: Timmermann and Goeller (2017) with global ocean (but regional ice sheet) is one example, your co-author KA Naughten runs another.

4. l. 29: "augmented" (like an add-on) does not quite match the fact that at least some of these models were designed with ice shelves included right from the start. Two of the early, pioneering models of this kind were Beckmann et al. (1999) and Timmermann et al. (2002).

5. l. 47: instead of "usually", I would find "often" or maybe even only "sometimes" more appropriate.

6. l. 60: The statement that the model "includes all the model physics of state-of-the-art regional applications." seems a bit daring to me, given that sea ice (which is commonly regarded as part of the ocean) is only roughly approximated in this model.

7. l. 119: "[polynyas] are critical to resolve accurate ice shelf melting in cold regimes": That's what people say. In fact, it is quite en vogue to stress the importance of coastal or flaw

polynyas. And it is not totally wrong at all. If you do the budgets though (let's say: for the continental shelf of the southern Weddell Sea), it turns out that the leads in the (vast) pack add up more salt flux through sea ice formation than the (comparatively tiny) coastal polynyas. What does make coastal or flaw polynyas important indeed is the fact that they are persistent and stationary. See, e.g., Haid and Timmermann (JGR 2013). So that statement is not totally wrong, but maybe a tad on the simplifying side.

8. l. 149: "while Bedmap2 ice thickness data is mostly based on laser altimetry data from 1994 to 1995" Are you sure this is true? Bedmap2 is much younger than this. Please double-check.

9. l. 166: I believe it should be Z = H (cosG+i sinG) (with brackets)

10. l. 183 etc: In the "Resolution effects" section, I think it would be very useful to not only discuss resolution-caused changes, but also whether these bring modelled hydrography closer to or further away from observations. Maybe this is easier if this section is moved to the end of the chapter? No preference, just an idea. Judging from Fig. 4, I am not convinved that I find the statement "the model solution [....] converges with increasing resolution" fully justified. We are indeed far away still from an asymptotic behaviour. Which is probably true for the vast majority of models in use today, so I am not criticizing the model here.

11. l. 207: You may want to finish the sentence with "and the representation of narrow troughs at the continental shelf break (Nakayama et al., 2014)"

12. l. 220-222: This passage is not fully convincing. Stronger water mass transformation would (in my view) go via more or saltier HSSW - which (according to their statement a few lines above, and consistent with Fig. 6) is not what the authors find. How does the model form WSBW with S>34.8 if no HSSW with at least the same salinity exists? There has to be a source somewhere, and I do not agree that finding this source can be beyond the scope of this study.

13. l. 225-227, particularly with regard to "Which of the models is more accurate close to the surface and what is causing the differences is not clear.": A purely observation-based data product (like the World Ocean Atlas) might help.

14. l. 232/233: "The z-like signature of ISW in the Ross Sea is likely caused by continued mixing of ISW from one ice shelf inside the cavity of another ice shelf downstream and this further supports the presence of ice shelf teleconnections." I think the statement here could/should be more precise. The idea is that these patterns are signatures of meltwater originating from ice shelves upstream from Ross Ice Shelf, right? So, this would be meltwater from the Amundsen / Bellingshausen Seas? I am not sure whether this explains the structure in the Ross ISW, but the idea of a teleconnection between this and those is supported by the findings of Nakayama et al. (2020).

15. l. 239-248: To me it seems as if this whole paragraph calls for a model-to-data comparison instead of (or in addition to) model-to-model.

16. Figure 9: Having the ice shelf on the left of the plot in the section AND in the map would be nice.

17. Figure 9 again: Is the very sharp front in the (c) panels a simulation result or are we too close to the open boundary / sponge layer here?

18. l.283: I think it should be "in agreement or close to FOR others"

19. l.310: The finding that strong ice-shelf basal melting near the ice front in this model is a widespread feature needs some discussion in context with numerics / sigma coordinates. Is there any risk that the particularities of terrain-following coordinates create a certain tendency / bias here? If mixing is not carefully controlled and ideally rotated to density surfaces (instead following lateral coordinate lines), a spurious exchange between the open-ocean surface and the ocean in touch with ice the shelf base near the ice front may be something to keep an eye on, I think.

20. l. 313, "accuarate polynyas": Whether these are better in terms of giving the correct buoyancy flux than a prognostic sea-ice model may still be debatable. So, compared to resolution and tides, this may be a weaker point in the list of strengths of this model. Personally, I would concentrate on the "real strengths", with tides probably being the leader here, followed by resolution, and tune down the enthusiasm about the model's approach to sea ice processes. This may be a matter of scientific taste though and I do not insist that the authors follow my suggestion.

21. l 330: It is SUCH a pity that results from coarser resolution or deactivated tides are not shown!

22. l. 334, "spuriously low conversion rate of heat into ice shelf melting": This point I don't see, because even if you have a spuriously low ocean-to-ice heat flux, the transport of warm water onto the continental shelf is still the same, isn't it?

23. l. 355-357: I will shamelessly advertise RTopo-2 here. That said, it is highly unlikely that everything in RTopo-2 is perfect.

24. l. 359: In the list of studies on interaction between sea ice and ice shelves, you may REALLY want to add Timmermann and Hellmer (2013).

25. l. 360-362, "This study, however, prioritises accurate polynyas by prescribing surface fluxes from sea ice observations. While this is likely to result in more accurate melt rates at the base of the ice shelves" : I am not sure I agree with this. Having the polynyas at the right places is a good step, but the fluxes computed from there are probably much less well constrained.

26. l. 367/368, "This design, however, has been chosen to simplify future efforts that aim to couple WAOM with models of Antarctic ice sheet flow": This has just been said (two sentences back).

27. l. 406: "harness": Sure? Maybe "harvest"?

28. l. 420: Limitations of using just one particular year over and over again as atmospheric forcing need to be discussed. Think of periodic modes of variability and how each of these modes is randomly sampled in one particular phase and then repeated over and over again.

19.08.2020, with apologies for the substantial delay,

Ralph Timmermann

---

## Author Comment (AC1)

**Overall statement**

We thank all three reviewers for their thorough and constructive comments. Overall, we are pleased to see that the reviewers agree with the importance of the model and that the development step it represents is worthy of publication in GMD.

Concerns are mostly related to our evaluation strategy for the ocean. We have substantially strengthened the paper in this respect, closely following the reviewers suggestions. The focus has been shifted to the on-shelf regions and model predictions are now compared against observational estimates from WOA18 and CTD bottom layer measurements (compiled by Schmidtko et al., 2014).

Reviewer 2 also questions if the model is "fit for purpose" at its current stage. We now communicate more clearly that WAOM v1.0 is just the beginning. The model has already been picked up by several research groups focusing on ice sheet-ocean coupling (R. Gladstone 2020, pers. comm., 29 June), dense water formation (U. Petteri 2020, pers. comm., 7 January) and future predictions of ice shelf melting (J. Moore 2019, pers. comm., 11 June). These groups further calibrate and evaluate the model according to their specific research questions. At this stage, as presented in this manuscript, the model is suited to study individual processes involved in ice shelf-ocean interaction, predominantly tides. We now communicate the distinction between the wider scope future of WAOM and the applications of its first version more clearly.

We sincerely hope that our responses are satisfactory for the reviewers. The diverse, ongoing research with WAOM also highlights the need to publish this tool now.

In the following we have addressed all comments. Reviewer comments are back, our response is blue. Changes to the text are printed in *italic*. Modified or added figures are provided at the end of the document.

For cross referencing we have labelled each comment. R1C2, for example, refers to Reviewer 1, Comment 2. Longer responses can include subsections (e.g. R1C2, Discussion: Biases).

**Changes unrelated to the reviews:**

We now have managed to activate parallel I/O for the 2 km version of the model, significantly enhancing the performance of the model. We are also running the model now on the Australian Government National Computing Infrastructure's latest supercomputer gadi.

We have updated Table 1 from:

| Model Resolution | 10 km | 4 km | 2 km |
|---|---|---|---|
| Period simulated | 1 year | 1 year | 1 year |
| CPU hours | 280 h | 6,840 h | 40,030 h |
| Architecture | Sandy Bridge | Sandy Bridge | Broadwell |
| Number of CPUs | 256 | 2304 | 224 |
| Memory | 51 GB | 2.9 TB | 876 GB |
| Walltime | 1 h | 3 h | 142 h |
| Storage for 1 3D field | 40 MB | 250 MB | 1 GB |

**Table 1.** Computational requirements at different resolutions. WAOM has been run on the supercomputer Raijin from the National Computing Infrastructure (NCI) in Australia. Sandy Bridge architecture stands for 2x8 core Intel Xeon E5-2670 2.6GHz with 32 GB RAM per node and Broadwell is 2x14 core Intel Xeon E5-2690v4 2.6GHz with 128 GB RAM per node. We needed to ensure a high RAM per CPU for the 2 km application as input-output was handled in serial.

to:

| Model Resolution | 10 km | 4 km | 2 km |
|---|---|---|---|
| Period simulated | 1 year | 1 year | 1 year |
| CPU hours | 93 h | 1,877 h | 16,983 h |
| Number of CPUs | 288 | 2304 | 5184 |
| Memory | 45 GB | 390 GB | 1.53 TB |
| Walltime | 0.5 h | 1.2 h | 4.4 h |
| Storage for 1 3D field | 40 MB | 250 MB | 1 GB |

**Table 1.** Computational requirements at different resolutions. WAOM has been run on the supercomputer Gadi from the National Computing Infrastructure (NCI) in Australia. The architecture consists of 2x24-core Intel Xeon Platinum 8274 (Cascade Lake) 3.2 GHz CPUs per node with 192 GB RAM per node. Listed times are for time-stepping only, that is without initialization or Input/Output.

And changed the respective paragraph in the model description (p. 4, lines 107-113):

*"Table 1 summarises the computational costs associated with running the model on the Australian National Computing Infrastructure (NCI) supercomputer  Gadi. On the resulting grids with 10 km, 4 km and 2 km resolution the 3-D equations integrate stably with timesteps of, respectively, 900 s, 360 s and 180 s. This leads, for example, to a cost of  1,877 CPU hours for 1 year of simulated period at 4 km resolution.  We note that initialization and Input/Output require additional resources."*

Respective development advice is also redundant now (p. 25, lines 402-405):

"

- "

**Ocean Evaluation**

Comments from all three reviewers challenge our general evaluation strategy for the ocean conditions. We now include a quantitative comparison of the ocean hydrography against observational estimates, closely following the reviewers suggestion. Related changes to the manuscript are extensive, and best presented together and upfront. In the following, we outline these modifications structured by section. Later, within the individual reviewer comments, we refer back to these changes.

**Methods**

We have changed the title of the respective section from *Analysis* to *Model Evaluation* and have extended this section by several paragraphs. Upfront we now outline the scope and strategy of the evaluation presented in this study (p. 7,line 146):

*"2.6 Model evaluation*

*In this study we present a tool for the community that can ultimately be used to address many different questions related to ocean-ice shelf interaction. Future studies intending to apply WAOM will need to tune and evaluate the model to their specific needs. In this manuscript we focus on ice shelf basal melting and, hence, have focused our evaluation strategy on this quantity. Also, melt rates contain the integrated history of the upstream ocean and their evaluation implies insights into the hydrography of sub-ice shelf cavities and the adjacent continental shelf. In addition, we directly compare ocean hydrography against observations to provide a first estimate of the biases. This helps to better explain the predicted melt rates and provides a starting point for future studies with different focus."*

Next, as a consequence of a detailed description of the ocean evaluation, we have also elaborated on our evaluation strategy for ice shelf melting (p. 7,line 148; added parts are underlined):

*"We compare annual mean ice shelf mass loss averaged over individual regions and for total Antarctica against satellite observations from Rignot et al. (2013), Depoorter et al. (2013) and Liu et al. (2015). Uncertainties for satellite derived ice shelf-ocean interaction at high resolution are unknown (discussed earlier). In this regard, we showcase models results and compare predictions against theory, regional studies and satellite estimates in the text. [To calculate basal mass loss … ]"*

For the ocean evaluation, we have removed the paragraph describing SOSE (p. 7, lines 152-156):

*"We use SOSE to evaluate the off-shelf ocean. As mentioned earlier, SOSE assimilates many observations from elephant seals, ships and Argo floats in the Southern Ocean, making it very reliable where such observations exist (Mazloff et al., 2010). On the shelf, however, observations are sparse and the ocean dynamics used to integrate SOSE do not include ice shelf-interaction. Hence, we expect SOSE to have large biases close to the ice and we only use its solution for the off-shelf ocean to evaluate WAOM."*

Instead we now present our choice of observational products and some crucial information about their underlying sampling density. We also use this paragraph to explicitly define the boundaries of the model solution (p.7, lines 152 and following):

*"For the ocean evaluation, we have chosen to use WOA18 climatologies and estimates of on-shelf bottom layer hydrography from Schmidtko et al. (2014). WOA18 is most accurate in summer, when sea ice has its minimum extent and the vast majority of observations are taken. The deep ocean is expected to show little seasonality though. Observations on the shelf are sparse and often concentrated along repeated ship tracks (see Fig. C1). The ocean state on the shelf is critical in determining the circulation and melt rates in the ice shelf cavities. Here, bottom layer hydrography is of particular interest, as it provides information about CDW intrusions and dense water formation. Schmidtko et al. (2014) provides a comprehensive compilation of on-shelf bottom layer hydrography from CTD measurements. The northern extents of the off-shelf ocean should be seen as a sponge layer, likely affected by ECCO2 boundary and initial conditions and not fully spun-up using our procedere. The flux-forced approach at the surface decouples sea ice conditions from the underlying ocean. This is known to create artificial water masses in the uppermost layers of the model. Hence, the top 15 m (equivalent to the uppermost 2 sigma layers in most regions) should be seen as a boundary and are excluded from this analysis."*

Finally, we describe the technical details behind the comparisons regarding the ocean (p.7, lines 156 and following):

*"On the shelf (south of the 1500 m isobath), we compare the summer mean (December, January and February) of the WOA18 climatology from 2005 to 2017 against the summer mean of 2007 as predicted by WAOM. We use Temperature-Salinity (TS) diagrams to assess the water masses and longitudinal transects for the stratification. For the TS-diagrams, both products have been sampled on their original grid. The transects are taken where CTD data underlies the WOA18 product (along ship repeat tracks and on the Amundsen Sea continental shelf, see Figure F.2) and WAOM's estimates have been interpolated to the WOA18 grid (1/4° and up to 102 depth levels) using a nearest neighbour scheme. Further, we compare multidecadal means of bottom layer hydrography from Schmidtko et al. (2014) against the 2007 mean from WAOM. The CTD locations have been interpolated onto the model grid using nearest neighbour interpolation. Then, model data has been interpolated to the depth of the observations using the nearest neighbour scheme. We augment these comparisons by showcasing high resolution transects and regional TS-diagrams that include the cavities from WAOM and compare these results against regional studies in the text.*
*We define the off-shelf ocean as south of 65°N and north of the 1500 m isobath. Here we compare the summer mean of the 2005-2017 climatology from WOA18 against the 2007 summer prediction of WAOM. We also include the prediction of the 2007 summer mean from ECCO2, which provides the initial and boundary conditions for WAOM. Differences in bottom layer hydrography between WOA18, ECCO2 and WAOM are assessed using annual means, as we expect little seasonality at such great depths. All observational estimates have been converted to model quantities (potential temperature and practical salinity)."*

**Results: On-shelf hydrography**

We have restructured the evaluation results to better reflect their importance for this study: 1. Ice shelf melting, 2. on-shelf hydrography and 3. off-shelf hydrography. Results related to ocean hydrography are now presented in one section, resulting in the following sections: *3.3. Ice shelf melting* and *3.4 Ocean hydrography*.

For the on-shelf regions we have added 6 figures comparing the model solution against observational hydrography in TS-space, as bottom layer maps and along selected transects (Figs. 5-10). We have added the following paragraphs to describe these figures (p. 11, lines 210 and following):

*"3.4 Ocean hydrography*

[revised manuscript text omitted]

The focus of the ocean evaluation lays now on the on-shelf regions, leading us to remove results regarding off-shelf stratification from the manuscript (previous Fig. 7 and related text shown below). We note that reviewers have commented on the representation of vertical mixing based on the off-shelf stratification. The same issue is apparent for the on-shelf and is still being discussed (later).

*"The stratification of WAOM agrees well with SOSE for the off-shelf ocean and, as expected, diverges towards the ice shelves. Figures 7a to 7d show longitudinal transects of temperature and salinity of both models. In the open ocean away from the continental shelf break, the solutions agree and this supports realistic boundary constraints and mixing processes in WAOM. Towards the shelf break and on the continental shelf WAOM resolves substantially colder and fresher waters compared to SOSE, which we interpret as the result of melt water from the ice shelf cavities. WAOM also often shows stronger vertical mixing close towards the continental shelf, possibly caused by surface forcing, tides or pressure gradient errors. The ocean close to the continental shelf is often well mixed in WAOM, but remains relatively stratified in SOSE (as, e.g., can be seen in Prydz Bay transect, Fig. 7d) and this could have various reasons. First, brine rejection in sea ice polynyas is known to cause deep mixing of the entire water column (e.g. Silvano et al., 2018). While WAOM and SOSE use the same mixed layer parameterisation (KPP), different surface forcing and melt water in WAOM might change the sensitivity to deep convection. Second, SOSE does not include tides. Tidal currents are known to contribute to ocean mixing and tidal strength amplifies towards shallower waters (e.g. Padman et al., 2009). Finally, spurious currents from pressure gradient errors at steep sloping topography in sigma-coordinate ocean models might also contribute to more mixing in WAOM (Mellor et al., 1994, 1998). This argument is supported by the fact that WAOM produces enhanced mixing also in the vicinity of deep ocean ridges, e.g., in the Ross Sea (Fig. 7b)."*

Finally, four figures presenting auxiliary information related to the ocean evaluation have been added to the supplemental material (Fig. C.1 to C.4).

**Discussion: Biases**
The new findings support that WAOM captures large-scale characteristics of the on-shelf hydrography important for ice-shelf melting. We also have been able to identify three main biases in the ocean: a spuriously warm surface, a lack of HSSW formation and overly mixed conditions on the continental shelf. In the light of ice shelf-ocean interaction, missing HSSW has been rated as most important (in addition to a cold bias in the Amundsen-Bellingshausen Seas, which had already been revealed from the evaluation of ice shelf melting).

In the discussion, we have adapted the paragraph about model biases to communicate these points clearly and hypothesise connections between the biases.

We have changed the original paragraph from (p. 24, lines 342-357):

*"WAOM underestimates melting for some ice shelves and we speculate boundary conditions to be the cause. A cold bias in the Amundsen-Bellingshausen Seas is a common issue in large scale models (e.g. Naughten et al., 2018b) and has been attributed to, either, insufficient transport of deep ocean heat onto the continental shelf (as mentioned earlier), insufficient transport of onshelf heat into the sub-ice shelf cavities or underestimated conversion efficiency of heat into melting inside the cavity (Nakayama et al., 2014; Dinniman et al., 2015). In our simulation, onshelf ocean temperatures in the AmundsenBellingshausen Seas are comparable to observations and where deep warm water intrusions reach the ice shelf cavities melt rates also agree (e.g. for George V and Abbot). Therefore, we expect*

*insufficient transport of onshelf heat into the cavities to be the cause of underestimated melt rates in the model (e.g. for the ice shelves Pine Island, Getz and Totten, compared to Rignot et al., 2013; Depoorter et al., 2013; Liu et al., 2015). There are a multitude of mechanisms that could prevent onshelf heat from entering the cavity and that could vary between regions. For example, at Pine Island Ice Shelf, Davis et al. (2018) shows that local wind forcing modulates thermocline depth, which in turn controls the access of CDW into the cavity on weekly to monthly timescales. We do not account for the effect of sea ice on surface wind stress in the model and, thus, a bias in thermocline depth might cause low melting in this region. In contrast, for Totten and Moscow University Ice Shelves we attribute underestimated heat flux into the cavity mostly to a bathymetry bias. A regional model by Gwyther et al. (2014) resolves similar continental shelf temperatures, but uses a cavity thickness which is 5 times larger along the centreline compared to Bedmap2, and this model resolves melt rates comparable to satellite estimates.*" to:

"*WAOM underestimates melting for some warm water ice shelves and produces too little HSSW, both likely related to overly mixed conditions on the continental shelf. A cold bias in the Amundsen-Bellingshausen Seas is a common issue in large-scale models (e.g. Naughten et al., 2018) and has been attributed to, either, insufficient transport of CDW onto the continental shelf (e.g. Thoma et al., 2008; Nakayama et al., 2014; discussed earlier), too rapid erosion of heat on the shelf (e.g. Bett et al. 2020) or underestimated conversion efficiency of heat into melting inside the cavity (e.g. Dinniman et al., 2015). The ocean evaluation indicates that the second cause applies in our case. CDW enters the shelf, but gets mixed away too readily before reaching the ice (Fig. 6c). Indeed, WAOM is overly mixed in many regions (incl. the Bellingshausen Seas; see Fig. 9). We note that winds have also been shown to affect shoreward heat transport (Kimura et al., 2017; Greene et al., 2017) and we do not account for the effect of sea ice on wind stress. However, the sensitivity of ice shelf melting to momentum flux modulations have yet to be explored (as done in Jendersie, 2018).*

*Too much mixing might also be responsible for the reported lack of HSSW formation (Fig. 5). Integrated surface salt input in polynya areas compares well against the original forcing product by Tamura et al. (2011) (not shown), hence our surface salt flux tuning (see Methods) is not the cause for the bias. Instead, waters with salinities higher than 34.5 g/kg are indeed present in the uppermost 15 m, but readily mix within this layer before reaching greater depths (Appendix Figure C2). The reported warm bias at the surface (Fig. 5) could also be linked to reduced HSSW formation. WAOM predicts elevated melt rates right at the ice front in most regions (close to coastal polynyas; Fig. 11) and ISW has been shown to be able to suppress dense water formation (Williams et al., 2016; Silvano et al., 2018). However, we rate this possibility as unlikely, since the warm surface bias is less apparent in winter (not shown), when deep convection events are happening.*
*We also have reported CDW intrusions onto the continental shelf of the Eastern Ross Sea (see Fig. 6) and this is likely related to boundary effects. Where ACC jets cross the domain's boundary in shallow angles, artificial currents can arise. We have reduced these effects by making the boundary conditions outflow dominant (see Methods), but some artificial currents remain in the Ross Sea (see Fig. C.4). We hypothesise that these currents drive CDW onto the shelf by affecting the slope of the isopycnals close to the shelf break.*"

**Discussion: Future development**

The reviewers comments and related investigations allow us to provide more guidance for future development. For this, we emphasize the limited scope of the evaluation of this study and list observational datasets suitable for extended comparisons. We now lead with the following point (p. 25, lines 374 and following):

"

- *Future studies will need to calibrate and evaluate the model according to their research question. Morrison et al. (2020), for example, uses a pan-Antarctic ocean model to study water mass transport across the shelf break and, hence, evaluates the model using hydrographic profiles in the slope region. Suitable observational datasets for studies focused on the Antarctic seas (in addition to the ones applied in this study) include the Marine Mammals Exploring the Oceans Pole to Pole (MEoP) dataset (Roquet et al. 2014), a review of dense shelf water observations around Antarctica (Amblas and Dowdeswell 2018) and a monthly isopycnal/mixed-layer climatology (MIMOC, Schmidtko, Johnson, and Lyman, 2013). Available in-situ observations of ice shelf melting have yet to be compiled (as discussed in the next point)."*

We also have condensed our point regarding a universal evaluation matrix, now focusing exclusively on ice shelf melting at high resolution (rather than ice shelf-ocean interaction):
"

- ~~*Establishing an evaluation matrix for circum-Antarctic ice shelf-ocean models would open the path for efficient parameter tuning (similar to Nakayama et al., 2017) and allow the community to compare the performance between different models (see Naughten et al., 2018b). Many kinds of observations are useful for this, including ice shelf basal melting from phase-sensitive radar (ApRES), as well as ocean measurements from Conductivity(Salinity)-Temperature-Depth (CTD) sensors, Acoustic Doppler Current Profilers (ADCP) and turbulence measurement packages. These ocean instruments can be mounted on Autonomous Underwater Vehicles (AUVs) with under ice capability, underwater gliders, drifting floats, moorings and Seals. When rating the model performance against such observations, uncertainties of the underlying methods and the spatial and temporal variability of the observed quantities must be carefully considered. ApRES seems particularly suitable for large scale model evaluation as it comprises a robust and cheap method to observe basal melt rates over longer time periods. As more ApRES measurements are becoming available, their compilation could provide the backbone for such an evaluation matrix, similar to tide gauge measurements for tidal accuracy (King and Padman, 2005). Comparison of a wide array of ApRES data is already underway with the NECKLACE programme.*~~

- *Establishing an evaluation matrix for Antarctic ice shelf melting at high resolution would open the path for efficient parameter tuning (similar to Nakayama et al., 2017) and allow the community to compare the performance of different models (see Naughten et al., 2018) and satellite derived estimates. ApRES seems particularly suitable for large scale model evaluation (e.g. Gwyther et al., 2020) as it comprises a robust and relatively cheap method to observe basal melt rates over longer time*

*periods. As more ApRES measurements are becoming available, their compilation could provide the backbone for such an evaluation matrix, similar to tide gauge measurements for tidal accuracy (King and Padman, 2005). Comparison of a wide array of ApRES data is already underway with the NECKLACE programme[1]."*

Further, based on the findings of this study, we have added three points to describe future development steps aiming to improve WAOM's accuracy regarding sub-ice shelf cavity conditions (p. 25, lines 386 and following).
"

- *To improve WAOM v1.0 (focused on accurate sub-ice shelf melting) future development should focus on reducing mixing to better represent the stratification on the continental shelf. We have scaled horizontal tracer diffusion linearly with resolution, but have not tuned this parameter against observations. Likewise, stratification is sensitive to the chosen mixing (here LMD, which includes KPP) and advection schemes (here 4th-order Akima for the horizontal and vertical), and the effects of different choices have yet to be tested for WAOM. Finally, the sensitivity of stratification to different slope factors (Haney factors) should be explored. Spurious mixing at steep sloping topography (related to pressure gradient force errors in sigma coordinate models; discussed earlier) is sensitive to the degree of smoothing. Our smoothing procedure is similar to regional studies and the smoothing algorithm has been shown to perform well for a realistic, complex case without ice (The Adriatic Sea, see Sikirić, Janeković, and Kuzmić 2009). However, other pan-Antarctic studies have chosen different routines and algorithms and do not report overly mixed conditions mixing (Naughten et al., 2018). The Haney factor controls the degree of smoothing within any given scheme and, hence, offers a metric to assess the sensitivity without implementation of new procedures.*
- *Second priority should be given to the calibration of the surface heat flux, which is likely to reduce the warm surface bias. The warm bias towards the surface can not be explained by initial and boundary conditions, as ECCO2's upper ocean conditions are more realistic (see Fig. 10). Also, 2007 has not been an anomalously warm year (e.g. measured by sea ice extent; see Parkinson, 2019), rendering interannual variability as an unlikely source. Instead, we suspect the applied surface flux schemes to be responsible. A similar scheme is known to overestimate annual heat flux into the ocean by about 50% (Jendersie et al., 2018). While we aim to account for this by reducing positive heat flux into the ocean by half (see Methods, Sect. 2.4), the approach has not been tested for pan-Antarctic domains.*
- *In third place, the boundary effects in the Eastern Ross Sea should be addressed. Introducing a sponge layer is difficult, since tides are also forced at the open boundary. Instead we recommend an adjustment to the model boundary locations to avoid intersection with ACC jets at shallow angles.*

"

In the light of the revealed biases, advice for future field campaigns has been removed (p. 25, lines 386)
"

-
* * *
[1] NECKLACE programme: http://www.soos.aq/news/current-news/330-necklace-workshop-update.

*(Pine Island, Getz, combined Brunt and Riiser-Larsen, Shackleton, combined Totten and Moscow University), more ocean measurements, including bathymetry, should be taken near the front of these ice shelves. Also, ApRES measurements are particularly valuable where high resolution satellite estimates have their greatest uncertainties, that is in calving regions and close to grounding lines. Although, crevasses are often present in these regions and can impede the successful interpretation of ApRES results."*

**Summary and conclusion**

We have adapted the concluding paragraph about model performance and future development, being explicit about the revealed biases (p. 26, lines 422-432) from:

"*Model results compare well against available observations. Continental shelf ocean temperatures and ice shelf melting converge with increasing model resolution, but a further refinement to 1 km grid spacing is likely needed to reach asymptotic behaviour. The accuracy of tidal height signals at the coast is comparable to state-of-the-art barotropic tide models and the off-shelf hydrography agrees well with SOSE, which assimilates most of the available observations in the Southern Ocean. On the continental shelf, where observations are sparse, WAOM resolves realistic hydrography, e.g., featuring bottom layer temperatures of 1 °C in the Amundsen-Bellingshausen Seas and WDW in the Weddell Sea. Ice shelf melting and marine ice accretion at high resolution show that WAOM captures the known modes of melting, often in agreement with regional studies. Ice shelf average melt rates agree with satellite observations at many places, but indicate a cold bias for some of the warm water ice shelves in the Amundsen-Bellingshausen Seas as well as the Totten and Moscow-University Ice Shelf System. We attribute these discrepancies to insufficient heat flux from the continental shelf into the sub-ice shelf cavities, likely due to regional uncertainties in bathymetry or wind stress.*" to:

"*WAOM qualitatively captures the broad scale difference between warm and cold regimes and many of the known characteristics of regional ice-ocean interaction. Continental shelf ocean temperatures and ice shelf melting converge with increasing model resolution, but a further refinement to 1 km grid spacing or finer is likely needed to reach asymptotic behaviour. The accuracy of tidal height signals at the coast is comparable to state-of-the-art barotropic tide models. The total ice shelf basal mass loss is close to, but 4% below the lowest estimate derived from satellite observations. The basal mass balance of individual ice shelves agrees with satellite observations in many places, but indicates a cold bias for some warm water ice shelves in the Amundsen-Bellingshausen Seas as well as the Totten and Moscow-University Ice Shelf System. Ice shelf melting and marine ice accretion at high resolution are often in agreement with regional studies, demonstrating that our model captures the known modes of ice shelf-ocean interaction. The on-shelf hydrography resembles many aspects of WOA18 summer climatologies and decadal mean bottom layer temperatures by Schmidtko et al. (2014), but exhibits a lack of HSSW formation, a warm bias at the surface and excessive mixing. We hypothesize that the cold bias in the Amundsen-Bellingshausen Seas and the lack of HSSW is caused by overly mixed conditions on the continental shelf.*

*Future studies will need to evaluate and calibrate the model according to their specific research question. To improve the model's accuracy regarding ice shelf melting, the biases revealed here should be addressed first. Any further tuning will first require a compilation of available in-situ observations (from ApRES measurements). Such efforts are underway with the NECKLACE programme."*

**Introduction**

Where we introduce model evaluation, we have adapted the information about ocean observations (p. 2, lines 54-57) from:

"" to:

"*Further, compilations of ocean observations, such as the World Ocean Atlas 2018 (WOA18), include most of the available data from ships, Argo Floats, gliders and elephant-seals, but the interpolated temperature and salinity fields are only available as climatologies with up to decadal resolution and observations in sea-ice covered regions are sparse, implying large uncertainties on the Antarctic continental shelf.*"

In the light of the biases, now revealed for the on-shelf ocean, we have relaxed the tone of the paper, emphasizing that WAOM v1.0 should be seen as the first step (p. 3, lines 58-63) from:

[revised manuscript text omitted]

**Response to Review #1**

In this manuscript the authors describe a new regional model configuration (WAOM) that encompasses the entire Antarctic continental margins, including the cavities beneath Antarctica's floating glaciers. The scientific focus of the model is on simulating ocean-driven melt of Antarctica's ice shelves. The manuscript describes the set-up and integration of the model at three different resolutions (10km, 4km and 2km grid spacing), the highest of which is ostensibly capable of fully resolving tidal and fine-scale processes that contribute to the circulation and stratification on the continental shelf. The authors evaluate WAOM using an ocean state estimate and independent modeled/measured estimates of Antarctica's ice shelf melt rates, focusing on the highest resolution solution. They concluded that WAOM acceptably reproduces the observed ocean stratification and ice shelf melt rates, and therefore is a suitable tool for addressing scientific questions related to mechanisms of ocean-driven melt. They discuss shortcomings and sources of biases in the model, particularly emphasizing WAOM's lack of an active sea ice component, and discuss future development and scientific goals for the model.

**R1C1** My high-level evaluation is that this is a significant model development that is worthy of publication in GMD. As the authors note, this is the first ocean model with (borderline) tidal- and eddy-resolving resolution that includes all of Antarctica's ice shelves, and therefore offers insights into the role of these processes in modulating Antarctic ice loss at the continental scale. The description of WAOM is appropriate for a model definition manuscript, and configuration and analysis scripts are provided (remotely) to allow for complete reproducibility. The manuscript and figures are very clearly composed.

We thank the reviewer for this positive feedback. We would like to highlight that the reviewer rates the development step that WAOM v1.0 represents as significant enough for publication.

**R1C2** Below I have provided a series of comments and suggestions on the manuscript for the authors' consideration. My most significant concern pertains to their validation of the ocean state: 1. Offshore, the model is evaluated against the Southern Ocean State Estimate (SOSE), which contains biases of its own, particularly close to the Antarctic margins (see e.g. Dotto et al., 2014, Ocean Sci.). It was unclear to me why the authors chose to SOSE rather than directly against measurements. 2. On the continental shelf the authors perform only a qualitative evaluation of the ocean state, rather than performing direct comparisons. I was surprised by this, because the ocean state on the shelf is critical in determining the circulation and melt rates in the ice shelf cavities. Based on this, I would argue that the hydrography on the continental shelf should be the most closely scrutinized aspect of the model state. Previous studies have compiled measurement from the continental shelf from all around Antarctica in order to compute trends in shelf properties (Schmidtko et al., 2014, Science), characterize different dynamical regimes on the continental shelf (e.g. Amblas and Dowdeswell, 2018, Earth Sci. Rev.) and evaluate models (e.g. Morrison et al., 2020, Sci.

Adv.). In my opinion, the manuscript would be strengthened significantly if the authors used one of these datasets to map biases in shelf properties in WAOM.

This comment challenges our general evaluation strategy. We now include a quantitative comparison of the ocean hydrography against observational estimates, closely following the reviewers suggestion (see Sect. Ocean Evaluation). We agree with the reviewer that this has strengthened the manuscript considerably.
Regarding 1.: We had chosen SOSE as it provides estimates for the year 2007, omitting some questions related to interannual variability. However, we acknowledge that SOSE is not the same as observations and in fact is just another model, although it is constrained by data where it is available. Instead, we now use WOA18 climatologies as our main product to evaluate the off-shelf hydrography.
Regarding 2.: We have now added a quantitative comparison of the predicted on-shelf hydrography against observational estimates from WOA18 climatologies and Schmidtko et al. (2014). This first assessment of the on-shelf conditions has helped to explain biases in ice shelf-ocean interaction and provides a sound starting point for future development. We would like to note, however, that WAOM can be used to address many different research questions and future studies intending to use this tool to study processes other than ice shelf melting and beyond hourly to seasonal time scales, will need to tune and evaluate the model to their needs.
Related changes to the manuscript have been outlined earlier (Sect. Ocean Evaluation).

**R1C3** L4-5: How significant is lack of interannual variability? Is it possible that some of the biases in the modeled shelf stratification and melt rates occur because the model excludes anomalous years, e.g. years with particularly strong/weak winds or warm/cool atmospheric temperatures?

This is a great question. Also low-frequency intrinsic ocean processes might contribute to interannual variability in the model (Gwyther et al., 2018). Our application of the model using repeated year forcing allows us to study processes with hourly to seasonal time scales. However, we do acknowledge that limitations related to longer scale variability might impact the mean state. With the performed experiments, we can not answer to which degree this is the case. We have added a note to the limitations paragraph, including a statement about the timescales WAOM v1.0 has been designed for (p. 24, line 364):

*"Further, the forcing schemes of this first version of WAOM have been designed to study phenomena with hourly to seasonal timescales (e.g. tides and summer surface water advection). To address scientific questions related to inter-annual change, these schemes will need to be extended first. We note that neglecting larger scale variability from interannual change or intrinsic processes (Gwyther et al., 2018) might impact the mean state of the model."*

**R1C4** L61: "beyond the scope of this study" doesn't really mean anything. If it's not in the study then it's beyond the scope by definition. It would be more helpful to state why an evaluation matrix was not pursued. Also is model tuning really "rigorous"? A right answer for the wrong reasons is not necessarily better than a slightly wrong answer based on well-grounded physical parameter choices.

Development of an evaluation matrix was not pursued because we did not have the time resources to do both (compilation of available observations and model development). The difficulty behind evaluation of this particular kind of model has been discussed above (L46-49) and just been picked up here again to define the scope of this study. We agree with the reviewer that "rigorous" is not the right word here. We have removed the whole statement as it seems to cause conflict rather than clarification. We have changed the sentence in question from (p. 25, line 374):

"" to:

"We aim to convince the reader that WAOM is capable of [...] ".

**R1C5** L93: Horizontal grid sizes in the form M x N would be more relatable than total number of computational cells.

We agree and have changed the respective numbers from (p. 4, line 93):

"." to:

"[...], which results in 530x630, 1325x1575 and 2650x3150 horizontal cells, respectively."

**R1C6** L98: Are the authors referring to the scheme of Shchepetkin and McWilliams (2003)?

Yes. We have added the citation at the end of the sentence (p. 4, line 98):

"ROMS is designed to minimise this issue by applying the splines density Jacobian method for the calculation of the pressure gradient force (Shchepetkin and McWilliams 2003)."

**R1C7** L100, L304-311: How wide does the "vertical cliff" at the ice shelf face become with this smoothing? Does this bias the model toward more Mode 3 ice shelf melt?

How well WAOM represents processes at the ice shelf front is a great question. In WAOM we have used standard smoothing routines, established in regional studies (Galton-Fenzi et al., 2012; Cougnon et al., 2013; D. E. Gwyther et al., 2014). Assessment of the impact of these routines on the model solution warrants its own studies better done using idealized or regional experiments (similar to Schnaase and Timmermann, 2019). Further, the realism of frontal processes in different coordinate ocean models is ongoing research. New mechanisms for allowing surface waters to access ice shelf cavities continue to be discovered and there is evidence that the unrealistic ice front representation in sigma-coordinates actually compensate for unresolved processes (Malyarenko et al., 2019). WAOM predicts enhanced front melting in many regions, highlighting the importance to answer these questions. The original manuscript already highlights this issue as future work (L 395-397). We now have added a note to the model description (p. 4, line 103):

"[ … we smooth the bathymetry and ice draft iteratively until a maximum slope factor $r = (h_i − h_{i+1})/(h_i + h_{i+1}) \leq 0.3$ is satisfied … .] While this approach has been developed in regional

*studies (Galton‑Fenzi et al. 2012; Cougnon et al. 2013; D. E. Gwyther et al. 2014), the impact of these manipulations on the ice front representation and related processes (e.g. Mode 3 melting) is unknown."*

**R1C8** L102: Is the algorithm applied to both the ice shelf draft and the bathymetry, or only to the water column thickness?

We are now more explicit about the smoothing approach. The respective paragraph has changed from (p. 4, lines 101-103):

"" to:

*"We apply the Mellor-Ezer-Oey algorithm (Mellor et al., 1994), which is well established for bathymetry smoothing (Sikirić, Janeković, and Kuzmić 2009). We smooth the bathymetry and water column thickness directly until a maximum slope factor r = (hi − hi+1)/(hi + hi+1) ≤ 0.3 is satisfied for both. The ice draft is then redefined as the superposition of bed and water column thickness."*

**R1C9** L105: Could the authors elaborate on "one of the smallest modifications possible"? I don't understand why there would be a hard limit on the ice shelf thickness - only an increasingly severe time step constraint as the thickness approaches zero.

We agree with the reviewer. 20 m is only a practical limit to stay within a reasonable time step range. We have changed the respective statement (p. 4, line 105):

*"20 m is considered  small within reasonable stability constraints (Schnaase and Timmermann, 2019)."*

**R1C10** L115-118: I presume that the authors impose heat and salt fluxes, rather than just buoyancy fluxes.

Yes, we are more precise now (p. 4, line 115):

"At the surface, we apply daily  heat and salt fluxes [...]".

**R1C11** L116, L399-400: Imposing surface wind stresses directly with no accounting for sea ice is a significant caveat. For example, recent Arctic-focused work shows that sea ice plays a significant role in modulating the stresses felt at the ocean surface, and the ocean surface currents (see Meneghello et al. 2018, GRL and other subsequent papers from John Marshall's group).

We acknowledge that wind stress modulation by sea ice plays a significant role for the seasonal variations of the larger scale circulation and hydrography in the Arctic (Meneghello, Marshall, Timmermans, et al. 2018; Meneghello, Marshall, Campin, et al. 2018). How well this conclusion can be transferred to the Antarctic, however, is not clear - and thus the significance of this caveat is likewise unclear. The importance of wind variability for Antarctic

coastal hydrography and ice shelf melting has been shown in several regions and for various timescales: weekly to monthly variations of thermocline depth in Pine Island Bay (Davis et al. 2018); seasonal variations of shoreward heat transport in the Amundsen Sea (Kimura et al. 2017); interannual variability of Totten ice shelf melting (Greene et al. 2017); seasonal suppression of the thermocline in the Weddell Sea (Hattermann 2018). However, how sensitive these phenomena are to stress modulations by sea ice has yet to be quantified. In fact, the only Antarctic sensitivity study that we are aware of concludes that sea ice wind stress modulations are irrelevant for seasonal variations of the Ross Sea circulation (Jendersie et al. 2018). WAOM would be well suited to extend this sensitivity study to all Antarctic regions. If indeed coastal hydrography in other Antarctic regions is sensitive to sea ice modulation by wind stress, WAOM should be equipped with the parameterisations developed by (Jendersie et al. 2018), or use drift ice observations for surface stress instead. All of these ideas would be worthwhile pursuing in future work.

We have added a note to the model description (p. 6, line 123):
*"[We do not account for the effect of sea ice on wind stress or frazil ice formation (as in, e.g. Galton-Fenzi et al., 2012).] While wind stress modulation by sea ice has been shown to play an important role for the circulation and hydrography in the Arctic (Meneghello, Marshall, Timmermans, et al. 2018; Meneghello, Marshall, Campin, et al. 2018), its importance for Antarctic ocean-ice shelf interaction has yet to be constrained (as discussed in (Jendersie et al. 2018)."*

We also have elaborated on our proposal for future work, changing the respective paragraph from (p. 25, 398-400):
"" to:

*"Wind stress has been shown to impact ice shelf melting (Davis et al., 2018; Greene et al., 2017, Kimura et al. 2017, Hattermann 2018), but how sea ice modulates momentum flux from the atmosphere into the ocean is not well constrained (Lüpkes and Birnbaum, 2005; Nøst et al., 2011; see discussion in Jendersie et al. 2018). Jendersie (2018) provides a first parameterisation for windstress modulation by sea ice and performs sensitivity experiments using a ROMS configuration of the Ross Sea. The effects on the seasonal variations of the circulation are negligible. WAOM would be well suited to extend these tests for a pan-Antarctic context."*

**R1C12** L117-118: Please explain why not using a sea ice model is more likely to capture polynyas? This is counter-intuitive.

We now have elaborated on this statement. Following comment R3C20 we have also relaxed our statement from *prescribed surface fluxes accurately capture polynyas* to *prescribed surface fluxes accurately capture polynya position and strength (surface salt flux)*.

The sentence has changed from (p. 4, lines 116-118):

"*Prescribing surface buoyancy fluxes, rather than including a sea ice model, is likely to more accurately capture polynyas that form in the lee of fast ice and icebergs, and are critical to resolve accurate ice shelf melting in cold regimes.*" to:

"*Accurate coastal polynyas that form in the lee of fast ice and icebergs are critical to resolve accurate ice shelf melting in cold regimes (see Mode 2 melting described in Jacobs et al., 1992). Small scale katabatic winds and grounded icebergs play an important role for these polynyas* (Kusahara, Hasumi, and Tamura 2010; Mathiot et al. 2010)*, but both (small-scale winds and ice bergs) are not well represented in current generation sea-ice models. Hence, prescribed surface buoyancy fluxes rather than including a sea ice model, is  more likely to capture the position and strength of coastal polynyas.*"

**R1C13** L119-122: The authors have quite a few rather ad hoc changes to the surface forcing that warrant further explanation. Why do the authors reduce the positive heat flux into the ocean by half? Is this to simulate the sea ice albedo effect? I understand the physical motivation for changing the brine rejection and restoring surface temperatures from below freezing, but it seems inconsistent to do so when all other fluxes are fixed. How do the authors gauge that one month is a "long" time scale for surface relaxation?

Decoupling sea ice-ocean fluxes from the ocean state is known to create artificial water masses. The applied tuning aims to compensate for these effects, but compromises consistency in the conservation of heat and salt. This is an inevitable downside of our approach. The tuning has been developed in regional studies and not yet calibrated for the pan-Antarctic domain. We believe that 0.5 is a good starting point for the heat flux tuning parameter, as a similar approach is known to overestimate heat flux into the ocean by up to 51% (Jendersie et al. 2018). The overestimation of summer heat flux into the ocean arises from the blended products (NCAR/NCEP or ERA-Interim) that are used where no sea ice is present. These products do not account for latent heat of sea ice melting.

In the methods, we now communicate the limitations of the flux-forced approach more clearly and provide more background for the tuning. We have added the following note (p. 6 , lines 119-122):

"*We tune the surface forcing by reducing positive heat flux into the ocean to half its original value, omit brine injection when the ocean is warmer than the freezing point and relax surface temperatures towards freezing when they are being forced below freezing.*"

"*Decoupling sea ice-ocean fluxes from the ocean state is known to create artificial water masses. A similar approach, for example, is known to overestimate heat flux into the ocean by up to 51% (Jendersie et al. 2018). To reduce such effects, we tune the surface forcing by reducing positive heat flux into the ocean to half its original value, omit brine injection when the ocean is warmer than the freezing point and relax surface temperatures towards freezing when they are being forced below freezing.*"

We now communicate explicitly that few resources have been dedicated for model calibration (changes repeated from R1C3, Discussion, Future development):

*"A similar scheme is known to overestimate annual heat flux into the ocean by up to 51% (Jendersie et al. 2018). While we aim to account for this by reducing positive heat flux into the ocean by half (see Methods, Sect. 2.4), the approach has not been tested for pan-Antarctic domains."*

The surface salt flux relaxation with timescales of one month is "long" in comparison to the daily atmosphere-sea ice-ocean flux forcing. The reviewer is right that we can not extend this statement easily to heat flux tuning. We have changed the statement to be purely factual (p. 6, line 121):

*"Further, to avoid model drift, the surface ocean is relaxed  to the solution from SOSE [...]."*

**R1C14** L125-126: I am skeptical about the claim that the model state at the boundary is primarily dictated by the interior. Surely this is not a desired outcome, as remotely-formed water masses (especially CDW) need to be supplied by the open boundary conditions.

We have introduced stronger nudging on inflow than outflow to reduce boundary effects. The reviewer is right, we do not actually know the consequences of this tuning for the model state at the boundary.

We change the statement to be purely factual (p.6, lines 125-126):
*" we nudge inflow and outflow with timescales of 1 day and 1 year, respectively."*

**R1C15** Fig. 2: It looks like the melt rate drops instantaneously upon re-initialization with a 4km grid spacing, and again with a 2km grid spacing. Is there some geometrical impact of the grid refinement that causes this, or is the adjustment time scale just shorter than the monthly frequency of the model output that was used to create the plot?

It is a geometrical impact. Coarser resolution runs have greater ice shelf area than higher resolution versions (e.g. 4 km has 11% less area than 10 km). Most of the additional ice appears in frontal regions where melt rates are elevated.

We have added a note about this (p. 7, line 146):
*"The instantaneous drops in melting upon re-initialization is caused by geometrical effects of the grid refinement. The ice shelf area reduces with increasing resolution (e.g. 11% between 10 km and 4 km), predominantly in ice shelf frontal regions, where melt rates are elevated."*

**R1C16** (a) Eq. (1): Shouldn't the denominator be 1/N. (b) Also, if $Z\_j\hat{\ }m$ and $Z\_j\hat{\ }o$ are complex variables (Z is the complex amplitude), then shouldn't the complex magnitude be taken before squaring (or equivalently multiplication by complex conjugate).

Regarding (a): In Eq. (1), the factor 2 in the denominator comes from the definition of the deviation variance for each harmonic X by Padman and Fricker (2005):

*"At each location (xi, yi) where we have tidal harmonics from in situ data, we define the deviation variance for harmonic X, with observed (modeled) amplitude and phase ai,Xobs (ai,Xmod) and pi,Xobs (pi,Xmod) as:*

$$d_{i,X}^2 = \frac{1}{2}\left(a_{i,X}^{obs}\cos\left(p_{i,X}^{obs}\right) - a_{i,X}^{mod}\cos\left(p_{i,X}^{mod}\right)\right)^2$$

$$+ \left(a_{i,X}^{obs}\sin\left(p_{i,X}^{obs}\right) - a_{i,X}^{mod}\sin\left(p_{i,X}^{mod}\right)\right)^2.$$

*The ensemble-averaged rms deviation for a set of N tidal stations is given by*

$$E_X = \left(\frac{1}{N}\sum_i d_{i,X}^2\right)^{1/2},$$
*"*

In Eq. (1) shown in our manuscript, we have taken the factor 0.5 out of the sum (here the deviation variance is denoted as $\sigma$):

$$\sigma_x = \sqrt{\frac{1}{2N}\sum_{j=1}^{N}\left[Z_j^m - Z_j^o\right]^2},$$

The variance definition of a complex number involves the factor 2. This arises from the idea that a complex number can be described by two real and uncorrelated vectors (e.g. well explained in the prepress from Robert G. Gallager[2]).

Regarding (b): The reviewer is right. We have changed the formulas accordingly ("| |" instead of "[ ]"):

$$\sigma_x = \sqrt{\frac{1}{2N}\sum_{j=1}^{N}\left|Z_j^m - Z_j^o\right|^2},$$

and

$$\sigma_{comb} = \sqrt{\frac{1}{2N}\sum_{k=1}^{4}\sum_{j=1}^{N}\left|Z_{k,j}^m - Z_{k,j}^o\right|^2},$$

**R1C17** Eq. (2): Shouldn't the denominator be 1/(4N)? Also, shouldn't the Z_j^m and Z_j^o be indexed by k as well, to distinguish tidal components.
* * *
[2] https://www.rle.mit.edu/rgallager/documents/CircSymGauss.pdf

We do not take the mean of the deviation of all constituents, but the sum. Hence, factor 2 in the denominator comes from the discussion above and no further factor is required. We have corrected the indexing for Equation 2 to include k:

$$\sigma_{comb} = \sqrt{\frac{1}{2N} \sum_{k=1}^{4} \sum_{j=1}^{N} \left| Z_{k,j}^{m} - Z_{k,j}^{o} \right|^2} \, ,$$

**R1C18** L175-176: Is a 2005 tide model still considered state-of-the art?

We are not aware of significant improvements (or a study that shows these). However, we do acknowledge that referring to a 2005 study with *state-of-the-art* causes confusion. We have removed *state-of-the-art* from the statement. We have also updated our tide assessment study resulting in a slightly better performance.

Overall, the performance statement changes from (p. 8, lines 175-176):
*"The model has a combined RMS error of 27 cm, 17 % higher compared to an RMS error of 23 cm for state-of-the-art 2D Antarctic tide models assessed in King and Padman (2005)."*
to:

*"The model has a combined RMS error of 20 cm, which is within the accuracy of 2D Antarctic tide models (assessed by King and Padman, 2005)."*

We also have added a description of the update in the methodology (p. 8, line 168):
*"[... fails to converge.] We also disregard 3 stations, which are noted as partially grounded and show non-sinusoidal and complex behaviour (70 Amery IS, 43 Rutford ISTR, 106 Evans ISTR)."*.

**R1C19** Table 2: I think I understand how the stated quantities (e.g. RMSD phase in deg) are related to equations (1) and (2), but the naming convention and formulation of equations (1) and (2) make this much less clear than it could be.

The reviewer makes a fair point here. We have not presented the equations for the second and the third row of Table 2 (RMSD amplitude and RMSD phase). It is not custom to present this much detail (see e.g. Padman and Fricker, 2005; King and Padman, 2005) and we do not discuss these quantities in the text. Discussion of the RMSD of the complex amplitude is sufficient. We clarified the Table by removing the two rows in question and printing the mathematical symbols for the RMSD complex amplitude and combined complex amplitude (as in the equations above).

Table 2 changed from:

|                              | M2    | S2    | O1   | K1   |
|------------------------------|-------|-------|------|------|
| Number of ATG stations       | 101   | 94    | 87   | 79   |
| RMSD amp in m                | 0.23  | 0.18  | 0.07 | 0.09 |
| RMSD phase in deg            | 27.14 | 22.65 | 8.92 | 8.30 |
| RMSD complex amp in m        | 0.20  | 0.15  | 0.06 | 0.07 |
| Combined complex RMSD in m   |       | 0.27  |      |      |

to:

|                         | M2   | S2   | O1   | K1   |
|-------------------------|------|------|------|------|
| Number of ATG stations  | 98   | 91   | 87   | 79   |
| $\sigma_x$ [m]          | 0.14 | 0.11 | 0.06 | 0.08 |
| $\sigma_{comb}$ [m]     |      | 0.20 |      |      |

.

**R1C20** L190-191: Should we expect the model state to asymptote under grid refinement? After all, various model parameters (e.g. viscosities/diffusivities) implicitly change with the grid, as does the bathymetry.

The reviewer's concerns are justified. We do not perform a CFD-like resolution study, as we change several aspects of the model at the same time (sub-gridscale turbulence description, eddies, bathymetry, ice draft, tidal effects, etc). We use the term "model" in a very wide sense, including all these aspects and our experience with them. As stated in the text, convergence of this wide-sense model shows "that we start resolving what is most critical to our problem". We would hope that as we resolve these critical aspects better (and tune the other parameters according to our experience), the solution converges further. We acknowledge that this discussion is somewhat philosophical and we have changed the text to reflect this from (p. 10, lines 189-191):

"" to:

"*We note that several aspects related to model resolution have been changed simultaneously (bathymetry, ice draft topography, horizontal viscosity, horizontal diffusion, the model's ability to resolve physical processes such as internal tides and eddies). Thus, we use the term "model" in its widest possible sense here, referring to all these aspects together. From 10 km to 1 km, we expect the model solution to be less dependent on resolution, as we start resolving the processes most critical to our problem. Demonstrating convergence of WAOM as a whole is an important first step, proving consistency between our understanding and the models behaviour. Attribution of change to the individual resolution dependent aspects is also important, but out of the scope of this study, as it would require several additional series of experiments (discussed later)."*

We also describe future experiments in this regards in more detail (p. 26, lines 412-413):

*"Repeating the resolution experiment introduced in this chapter, but, successively deactivating tides and keeping the bathymetry resolution constant, would unravel the impact of grid spacing on shoreward heat flux from tides, eddies and bathymetry."*

*"Extending the resolution study introduced here would help to attribute the convergence behavior to individual aspects of the model. Future experiments should be designed to isolate effects due to changes in bathymetry, ice draft, tides and sub-grid scale turbulence parameterisation. This way, changes in shore-ward heat flux with increasing model resolution could be more clearly related to better representation of troughs, eddies and internal tides."*

**R1C21** Fig. 4: Is 10km->4km grid spacing a 250% increase in resolution? If the resolution is defined as the number of grid points per km, then the resolutions are 0.1, 0.25 and 0.5 km^-1. So 10km->4km->2km grid spacings correspond to resolutions of 0.1->0.25->0.5 km^-1, so the resolutions have increased by 0%, 150% and 400% relative to the 10km grid. Or the resolutions have been multiplied by 100%, 250% and 500% relative to the 10km grid. Please pick a consistent convention!

We corrected this mistake and changed the values for the increase in resolution to 150% and 400% in the text and Figure 4:

[Figure]

This mistake does not affect our discussion.

**R1C22** L215-222: I was initially confused by the authors' explanation of the RSBW and WSBW salinities, which the attribute to the (ECCO2-sourced) open boundary conditions. I would have hoped that water masses formed within the domain would not depend on the boundary conditions. That said, the model boundary cuts right through the Weddell Gyre, so perhaps it is reasonable to have inflow of some bottom waters. Perhaps the authors could clarify this in the text?

The reviewer makes a good point here. RSBW and WSBW are not entirely formed within the domain. They are the combination of dense shelf water (expected to form within the domain) mixing with other deep water masses like CDW (inherited from the boundary conditions) as they are exported down the continental slope. The model has been designed to resolve

accurate ocean conditions on the continental shelf. The off-shelf ocean should be seen as a sponge layer, likely affected by boundary and initial conditions. We have now included a comparison of bottom layer hydrography between WOA18, ECCO2 and WAOM (see Fig. 13, also see Sect. Ocean Evaluation). This comparison shows indeed that most of WAOM's deep ocean biases originate from ECCO2 boundary and initial conditions. We now define the boundaries of the model solution upfront (repeated from Sect. Ocean Evaluation, Methods):

*"The off-shelf ocean should be seen as a sponge layer, likely affected by ECCO2 boundary and initial conditions and not fully spun-up using our procedere."*

And state the source of the bottom layer bias in the results (repeated from Sect. Ocean Evaluation, Results: Off-shelf hydrography):

*"WAOM shows an overall warm bias by about 0.3 °C, which can clearly be attributed to the initial and boundary conditions from ECCO2. Bottom layer salinities in both models agree well with WOA18."*

**R1C23** I am very confused by the waters with salinities reaching 34.8 in Fig. 6. There is no other water anywhere in the model domain with such high salinities, and these high salinities are only found at great depth, far from the surface. So, what is the source of the very salty bottom waters?

We have made a mistake in the plotting routine, mixing up latitude and longitude. Instead of showing pan-Antarctic water masses south of 65 degS latitude, we originally plotted all water in the domain (incl. its boundaries) from 65-180 deg W. We have corrected the figure (left old; right new):

[Figure]

In the corrected figure salinities in the deep do not exceed 34.7. However, a source for these very salty bottom waters is still not apparent. The plot shows surface waters with comparable densities at about -1 degC, but these are mixed with lighter waters close to the surface, rather than descending to greater depths. A possible explanation is that the salty bottom waters are sourced by initial and/or boundary conditions from ECCO2 (also suggested by the reviewer under R1C22). WAOM does not generate sufficient HSSW to fuel these bottom waters. The lack of HSSW formation has been identified and discussed earlier (Sect. Ocean Evaluation). We now also have included a comment about the source of the salty deep waters (repeated from Sect. Ocean Evaluation, Results: Off-shelf hydrography):

*"The densest waters in WAOM show only little isopycnal mixing with colder surface waters (also see Fig. C.2). In agreement with the earlier identified lack of HSSW formation, this hints towards bottom waters in WAOM, which are mainly sourced by initial and boundary conditions from ECCO2."*

The corrected figure is shown as supplemental material now (Fig. C.2; as a consequence of the changes under Sect. Ocean Evaluation). We note that other TS-diagrams are not affected by this mistake.

**R1C24** Fig. 6: I suggest adding more lines/arrows to indicate water mass locations more precisely. The CDW label looks to be much too fresh (e.g. compare with Fig. 7), and AABW is labeled at a lower density than CDW!

We have placed the labels more carefully in the new reference figures (Fig. 5 and 12). As the salinity and temperature ranges for the individual water masses are not precisely constrained, we decided against lines and arrows to reflect this.

**R1C25** Also, I presume the portions of T/S space labeled "Weddell Sea" and "Ross Sea" are actually the Filchner-Ronne Ice Shelf cavity and the Ross Ice Shelf cavity, respectively.

The reviewer is correct. Water masses in the sub-ice shelf cavities are now only presented in the appendix (Fig. C.2; as a consequence of the new evaluation strategy). We now have labeled the source region of ISW in this figure precisely. The new presentation of on-shelf water masses displayed in the manuscript (Fig. 5b) does not include the cavities and attribution of remaining ISW does not add value to the discussion.

**R1C26** L235: Are the transects all annually-averaged? Are there significant deviations in the agreement between SOSE and WAOM in different seasons?

Yes, these are annual averages. We are now more explicit about this in the text (e.g. see Sect. Ocean Evaluation, Results) and the captions of all relevant figures (e.g. Fig. 5 and 6). The focus of the ocean evaluation lays now on the shelf, leading us to remove off-shelf stratification from the manuscript (including previous Fig. 7; see discussion under Sect. Ocean Evaluation, Results: Off-shelf hydrography).

Seasonality of model prediction is a great question. However, we only aim to provide a first bias estimate of the ocean conditions for WAOM. Future studies aiming to investigate seasonal processes will need to tune and evaluate the model to their needs. We

communicate this more clearly now and provide suitable datasets for future evaluations (see first point under Sect. Ocean Evaluation, Discussion: Future development).

**R1C27** Fig. 7: Why does the grid spacing in WAOM appear to be so coarse? The model grid spacing is 2km but the data points in this figure appear to be spaced 50-100km apart.

Fig. 7 has been removed from the manuscript (see discussion under R1C26 and Sect. Ocean Evaluation, Results). WAOM's computational grid is not aligned with geographical coordinates, hence the data has to be interpolated to longitude transects. We had chosen to use a target resolution identical to SOSE (⅙ deg) which is sufficient for the comparison and computational efficient. High resolution transects are showcased for the on-shelf (Fig. 11).

**R1C28** Fig. 8: Given that the authors have just compared a few transects here, why not align the transects with WOCE transects? Then they could include a third column of panels showing the WOCE measurements for additional reference.

The focus of the ocean evaluation lays now on the on-shelf, leading us to remove off-shelf stratification from the manuscript (including previous Fig. 7; see discussion under Sect. Ocean Evaluation, Results). Transects on the shelf have been aligned with ship repeat tracks (see Sect. Ocean Evaluation, Methods; Fig. C.1) as the reviewer suggests.

**R1C29** Also, often these transects are visually very similar, especially with these colormaps. Difference plots would be more revealing with regard to the model biases.

We now compare WOA18 and WAOM along transects using difference plots (Figs. 8-10).

**R1C30** L421: Has the model equilibrated in the higher-resolution cases? I don't see how the authors can judge this from just a one-year time series in the 2km simulation. I think it would be fairer to say that the model has equilibrated at 10km grid spacing, and has been continued from that equilibrated state at higher resolution.

We agree with the reviewer and have relaxed the respective statement (p. 26, lines 421-422) from:

"" to:

*"We have simulated present-day conditions by spinning up the model with repeated 2007-forcing. The model has equilibrated at 10 km grid spacing, and has been continued from that equilibrated state at higher resolutions (up to 2 km)."*

**R1C31** L433: It's really the "surface" stress that is uncertain: the wind stress is relatively well constrained by reanalyses, whereas the ice-ocean stress is much less well constrained.

We agree with the reviewer. The respective statement, however, has been removed as a consequence of new findings related to the on-shelf evaluation (see Sect. Ocean Evaluation, Summary and Conclusion).

**R1C32** Table C2: Please specify which relaxation parameters pertain to the open boundaries vs the ocean surface.

We now have specified the boundary for each relaxation time scale in Table C2:

**Table C2.** Some key model parameters.

| Parameter | value (10/4/2 km resolution) |
|---|---|
| Vertical resolution ( # layers) | 31 |
| Vertical coordinate transformation equation # | 2 |
| Vertical coordinate transformation stretching function # | 4 |
| Surface stretching parameter | 7 |
| Bottom stretching parameter | 8 |
| Critical depth (m) | 250 |
| Baroclinic timestep (s) | 900/360/180 |
| Barotropic timestep (s) | 25/10/5 |
| Horizontal diffusivity ($m^2s^{-1}$) | 50/20/10 |
| Horizontal viscosity ($m^2s^{-1}$) | 500/200/100 |
| **Relaxation time scale for tracers at the surface** (days) | 365 |
| **Relaxation time scale for ocean elevation at the surface** (days) | 3 |
| **Relaxation time scale for barotropic momentum at the open boundary** (days) | 3 |
| **Relaxation time scale for baroclinic momentum at the open boundary** (days) | 3 |
| Open boundary outflow/inflow nudging factor | 365 |

**Response to Review #2**

**R2C0** This is an interesting manuscript that describes some of the more technical aspects of a new circum-Antarctic configuration of ROMS. The novel features are that it includes sub-ice-shelf cavities and tidal forcing, and it is run at high resolution. The subject matter certainly fits the remit of GMD and I think that it could make a valuable contribution to the growing literature on modelling the ocean circulation beneath the Antarctic ice shelves. However, I have a number of concerns with the way the model has been set up and validated, and I think the authors should clarify the reasons for the approach they have taken and provide a more critical evaluation of their results.

We thank the reviewer for this positive feedback. We would like to highlight that they too rate the development step that WAOM v1.0 represents as worthy of publication in GMD. We have elaborated on the design choices where questions have been raised and substantially strengthened the evaluation by comparing model results directly against ocean observations (see Sect. Ocean Evaluation). We are explicit about the revealed biases and communicate the development state of the model more clearly (see Sect. Ocean Evaluation, Discussion).

**R2C1** Questions: 1) While terrain-following coordinates have a number of advantages, their performance over steep topography can be an issue. The authors apparently deal with this critical point in lines 95-106, but nowhere do they state the extent to which the topography of the ice front and continental slope have been modified by smoothing, or the magnitude of any residual pressure-gradient errors that they would expect after the smoothing process.

This is an important point, because in Figure 7 there are some very obvious artefacts in the model results that appear to be associated with topography. Could similar artefacts be affecting the properties and circulation at the continental shelf edge and at the ice fronts? If so, they could have a major impact on the results that have been presented.

This is a fair, but far reaching question. As the reviewer already points out, one of the challenges of using sigma coordinate is to strike a delicate balance between realistic topography (ice and bed) and little spurious mixing due to Pressure Gradient Force Errors (PGFE, well discussed in Naughten et al., 2018). It is customary to present the smoothness of the final topography using the slope parameter (r factor). Quantification of remaining artificial currents is not straight forward (Mellor, Ezer, and Oey 1994) and has not been done before for ice shelf-ocean models, where the concurrent change of bedrock and ice draft poses additional complexity. The effect of ice draft smoothing on ice shelf frontal processes is an active research question (Malyarenko et al. 2019; Wåhlin et al. 2020), best addressed in idealized or regional models first. We agree with the reviewer, that the off-shelf artefacts (previous Fig. 7) are likely related to PGFE. However, judged by the high resolution transects shown in Figure 11, we rate the level of localized artificial mixing on the continental shelf as acceptable. However, the entire on-shelf ocean seems overly mixed (see Fig. 8-10), possibly related to PGFE.
We are now more explicit about our smoothing procedure (see R1C8) and discuss the possibility of PGFE as a source for overly mixed conditions on the shelf (see Sect. Ocean Evaluation, Discussion: Future development).

**R2C2** 2) The surface forcing is unconventional, in that heat and freshwater fluxes are applied rather than derived from atmospheric variables driving a physical parameterisation of the atmosphere-ice-ocean exchange. This has the advantage of removing the need for a sea ice model that can introduce biases, but has the potential to introduce its own biases. Presumably that is the motivation for the rather ad hoc adjustments made to the fluxes, described in lines 119-124? Those adjustments really should be more carefully described and motivated. Furthermore the use of wind stress without a dynamic sea ice model is questionable. Wouldn't it be more consistent to derive ocean surface stress from ice motion observations? While wind stress may be a reasonable proxy for surface stress where the ice is in free drift, free drift is a poor assumption in many of the regions of interest in this study, particularly the Weddell Sea and almost all near-coastal regions.

Yes, the modifications aim to compensate for the absent coupling between sea ice-ocean fluxes and the ocean state ocean. This has already been discussed under R1C13. We have described the motivation and approaches in more detail now (changes to the text also presented under R1C13).
The issue of missing wind stress modulation by sea ice has already been discussed under R1C11 (including changes to elaborate on this issue in the manuscript). Deriving surface stress from ice motion is an interesting proposition and we have added it as suggestion to future work (p. 25, line 400; in addition to changes outlined under R1C11):

*"If sea ice wind stress modulation is indeed important, ice motion observations could be included for assimilation or calibration."*

**R2C3** 3) It is not obvious why the model has been forced with one year of data repeated, especially when that one year is characterised by "a paucity of observational data" (lines 10-11). Why not choose a year with more data? On lines 130-132 the authors state that the strategy of using one year allows them to integrate the model to quasi equilibrium, but does it really reach that state in only 8 years? I think the authors should show some further diagnostics, such as domain-averaged temperature, salinity, KE, etc, to justify that statement.

At the time of development, 2007 has been one of the few years for which all input data has been readily available (2005-2010; SOSE, ECCO2, surface fluxes from sea ice observations, Era-Interim). 2007 has been selected from these years, because surface heat, salt and momentum fluxes did not deviate qualitatively from the 1992-2011 mean.
We acknowledge that our point about the paucity of observations is poorly formulated. It is not particularly 2007, but rather observations for any individual year. As a consequence of our new ocean evaluation, this statement has already been removed from the manuscript (see Sect. Ocean Evaluation, Abstract).
The statement about the model equilibrium has been bold. We agree that it is unlikely that the entire domain (incl. the deep ocean) equilibrates in just 8 years. We now emphasise that the off-shelf part of the domain should be seen as a "sponge layer" rather than part of the model's solution (see Sect. Ocean Evaluation, Methods). Hence, we only focus on the equilibration of the on-shelf ocean. For this, mean ice shelf melting is a very powerful quantity, as the model has been designed to derive accurate sub-ice shelf melt rates and they include the integrated information of the shelf ocean upstream. We have clarified the importance of ice shelf melting for our evaluation (Sect. Ocean Evaluation, Methods). Further, as a consequence to R1C30, we have already relaxed our statement about the quasi equilibrium to the 10 km version only. After these changes, we believe it is no longer necessary to present the temporal evolution of additional quantities.

**R2C4** 4) It is also not clear to me why the authors have used one model (ECCO) to supply initial and open boundary conditions, then compared the results to a second model (SOSE). Comparing with observations (however limited) would give some kind of indication as to how realistic the results are, while comparing to the same model (ECCO) would inform us about how the differing model architecture and surface boundary forcing impacted the model evolution. Without knowing how ECCO and SOSE differ, it is not clear what conclusions can be drawn from the comparisons that are made between WAOM and SOSE. The problem is highlighted by Figure 6, where WAOM and SOSE look completely different, particularly at depth. Is that because ECCO and SOSE are very different, or are the differences produced by the surface forcing? If it is the latter, how can the deep ocean have been so extensively modified in only a few years? It tends to suggest that the model is much too prone to deep convection, which is quite a common problem in Southern Ocean models. But does that happen everywhere, or is that all a product of those model artefacts that appear over steep topography (see 1 above)?

This has been a very valuable comment. We have changed our strategy for the ocean evaluation (see Sect. Ocean Evaluation). We now compare model results against WOA18 climatologies and have included ECCO2 in these comparisons (Fig. 12, 13). We have removed comparisons to SOSE (incl. Fig. 6). We did indeed find that the off-shelf ocean is

impacted to large degrees by the boundary and initial conditions, explaining most of the biases for the off-shelf (see Sect. Ocean Evaluation, Results: Off-shelf hydrography).

**R2C5** 5) Figure 6 raises some other serious questions about the results. The ISW and RSBW/WSBW mixing lines appear to point to a water mass that apparently doesn't exist (HSSW). That suggests that they were formed from HSSW that was prescribed in the initial conditions, but has since been used up and not renewed. If that is the case, it points towards a model that is not in a quasi-equilibrium state (question 3 above), but is still in the process of evolving from initial conditions to some other state. Similarly the water masses at 1000 m depth are much too warm, suggesting that the original CDW prescribed in the initial conditions has also vanished and has been replaced by something quite different. That suggests some issue with the surface fluxes (question 2 above). It also suggests (again) that a longer integration would see the model continuing to evolve to a different state. If deep waters as warm as 3-4 deg C found their way onto the continental shelf, ice shelf melt rates would be much higher.

We have made a mistake in plotting of Figure 6 (discussed under R1C23). The problem of mixing towards non-existent HSSW remains. We now have identified a lack of HSSW in the model and communicate this clearly (see Sect. Ocean Evaluation, Discussion: Biases). We also communicate missing isopycnal mixing of deep waters with the surface (see changes under R1C23). Bottom waters are fueled by boundary and initial conditions to large degrees. How far the off-shelf ocean is from a quasi equilibrium, however, is not relevant for this study. The off-shelf ocean should be seen as a "sponge layer" (now communicated explicitly under Sect. Ocean Evaluation, Methods).
The corrected TS-diagram (Fig. 6) now shows cooler temperatures at 1000 m depth, more in line with CDW. Still, we have now identified a warm bias at the surface, impacting deeper water masses (mostly in the off-shelf regions, see Sect. Ocean Evaluation, Results: Off-shelf hydrography). This has been communicated clearly including a recommendation to tune surface fluxes in future studies (see Sect. Ocean Evaluation, Discussion: Future development).

**R2C6** 6) But perhaps the error is just in the plotting of Figure 6. The deep ocean stratification suggested by the trajectory of points from RSBW/WSBW to the (poorly placed) AABW label appears to be much too strong. I find it hard to believe that the whole domain can have been so extensively altered in 8 years of integration (assuming the initial conditions taken from ECCO looked something like the SOSE results). Has some error been made in converting temperature to potential temperature, or density to potential density?

We have made a plotting mistake (related to the region plotted; discussed under R1C23). The new comparison (Fig. 12) looks much closer to the observations, rendering the comment about deep ocean stratification redundant. All labels have been placed more carefully (including the AABW one). All TS-diagrams have been plotted using the displayed quantities.

**R2C7** 7) (a) The problems with the water mass structure apparent in Figure 6 are also seen in Figures 7 & 8. On line 234 the authors state that "The stratification of WAOM agrees well with SOSE for the off-shelf ocean", but I would disagree with that. In most transects the main pycnocline is not well represented, either in strength or depth, and the mid-depth salinity

maximum appears to be absent (worryingly consistent with Figure 6). While this is far from being the only model to get the stratification wrong (it is notoriously hard to get right), I think the results presented here warrant more than the glib statement that they agree well with SOSE. (b) In Figure 8 the absence of a source water mass for all the colder forms of ISW is again apparent, while ISW seems much too prevalent in the Amundsen and Bellingshausen seas, where it is hardly ever observed. Cooling waters too efficiently in the cavities hints at a problem with the balance of heat fluxes into the boundary layer (KPP) and at the ice-ocean interface. But again, some of these issues might arise because of the plotting, which makes the stratification look very odd. (c) The densest waters are shaded blue (Figure 8), implying they are at the surface. How is that possible?

Regarding (a), we now have focused our evaluation on the on-shelf regions (incl. their stratification) and results regarding the off-shelf stratification are no longer central to the manuscript (see discussion under Sect. Ocean Evaluation, Methods). We still respond to the reviewer's concern here. We fully agree about their rating of the realism of WOAM's off-shelf stratification. WAOM generates too much mixing (vertical for the off-shelf; vertical or horizontal for the onshelf). We do not know the source of the spurious vertical mixing, but suspect PGFE or the mixing scheme to be responsible (now discussed as future development, Sect. Ocean Evaluation, Discussion: Future development). Therefore, we agree with the reviewer that the original statement was too glib and it would be more correct to say that WAOM's stratification in the off-shelf regions shows overly mixed conditions and the magnitude of the bias is comparable to other forward models. We now show and discuss spurious mixing for the on-shelf regions (Fig. 8-10; Sect. Ocean Evaluation, Results, Discussion), including suggestions for future development (see Sect. Ocean Evaluation, Discussion).

Regarding (b), the lack of HSSW production and artificially dense surface waters have been discussed earlier (see Sect. Ocean Evaluation, Discussion). In summary, we now have identified a lack of HSSW formation as a main bias and communicate explicitly that the uppermost 15 m of the model are not part of the model solution (also Sect. Ocean Evaluation, Methods). In addition to these points the reviewer is concerned about the realism of ISW. We agree with the reviewer that missing HSSW is apparent, as the Gade lines do not extend up to the surface freezing point (as, e.g. in Naughten et al. 2018), but rather are constrained to individual isopycnals at the salty end (e.g. 1027.8 kg m-3, in the densest case). We would like to note that this is not necessarily pointing towards non-equilibrated conditions on the shelf (i.e. remnants of HSSW from initial conditions), as the displayed behaviour can also be explained with a combination of isopycnal mixing and mixing along Gade lines. In this scenario, the second component for "Gade line-mixing" (the first is glacial melt water) has already been supercooled due to isopycnal mixing with melt water from upstream regions. We now have added a comment to the evidence of missing HSSW presented in Sect. Ocean Evaluation, Results (additions in bold):

"*In general, HSSW comprises the densest water mass on the shelf and mixes with other, lighter waters. As a consequence of its absence, all water masses in WAOM are well restricted by the same isopycnal of 1027.8 kg m-3* **(also within the cavities, see Fig. C2)."**

The lack of HSSW and the possibility of spurious vertical mixing are now discussed under future work (see Sect. Ocean Evaluation, Discussion: Future development).

ISW in the Amundsen-Bellingshausen Seas are only apparent inside the cavities (e.g. compare previous Fig. 8 with Fig. 5, where no ISW fresher than 34.0 g/kg is apparent). However, we are not aware of observations of ISW in this region, even inside the cavities. Potential artificial ISW in the AB-Seas is likely linked to the cold bias. This bias has already been identified using ice shelf melting and on-shelf hydrography (Ocean Evaluation, Results: On-shelf hydrography), and is now discussed in more detail (see Ocean Evaluation, Discussion: Biases). Figure 8 has been removed from the manuscript as a consequence of the new strategy for the Ocean Evaluation (see Ocean Evaluation, Methods). Therefore presentation of potential artificial ISW does not add much information to the discussion and we have not modified the manuscript in this respect.

Regarding (c), the fluxed forced approach is known to generate artificial water masses in the uppermost layers of the model. We have now excluded the top 15 m from our analysis and explicitly state that these regions do not belong to the model's solution (see Ocean Evaluation, Methods). Nevertheless, we mention this feature as part of the discussion around the lack of HSSW (repeated from Ocean Evaluation, Discussion: Biases):

*"[...]. Instead, waters with salinities higher than 34.5 g/kg are indeed present in the uppermost 15 m, but readily mix within this layer before reaching greater depths (Appendix Figure C2)."*

**R2C8** 8) (a) Throughout the paper there is an implication that higher resolution is intrinsically better, but improvement in the model results with increasing resolution is never actually demonstrated. The implication of the discussion around Figure 4 (lines 184-191) is that at higher resolution still, shelf water temperatures and melt rates would drop further. However, at 2 deg resolution the mean melt rate has already dropped below the observed value and the 4 deg resolution simulation is arguably the best according to that single metric. (b) On a related note, I don't understand what the authors mean by "convergence" in that discussion. This term normally refers to the ability of a numerical code to reproduce an analytical solution as the grid size tends to zero. What solution should the model "converge" to in this case? The authors describe some features, cooling of the Bellingshausen Sea (line 196), for example, that appear to be worse in the higher resolution runs.

Regarding (a), the reviewer makes a fair point here. Yes, we implicitly convey that higher resolution is more accurate, as the model relies less on uncertain parameterisations. We are convinced that this is true for well calibrated ice shelf-ocean models and in the range of 10 km to 1 km. For WAOM v1.0 only few resources have been invested into calibration. Indeed, the 4 km solution of mean ice shelf melting is closest to observations and it could be argued that, if no further calibration is done, this solution should be used for scientific questions in this regard. However, depending on the research question, resolving relevant processes can be more important than model agreement with observations (as also stated by Reviewer 1 under R1C4). It is established in the field that eddies and troughs are important processes for ice shelf melting and need a kilometer scale resolution in models like ROMS (Dinniman et al. 2016). Ultimately, we should direct future efforts to get an eddying model (at least 2km) with realistic tides. WAOM is a major step towards this goal. We now have added a discussion around this point (p. 24, line 335):

*"[These findings stress the importance of resolving tides at 4 km horizontal resolution or finer in large-scale models.] Studies aiming to use WAOM for future predictions should consider the option of applying it at 10 km or 4 km horizontal resolution for computational efficiency. Such studies will need to evaluate the model (at different resolutions) depending on their research question. Judging on the single scale metric of mean ice shelf melting, the 4 km solution of WAOM is closest to the observations (Fig. 4). For process oriented studies, however, we recommend using the 2 km version, as resolving eddies at a kilometer scale resolution is critical for accurate ice shelf-ocean interaction in some regions (Stewart and Thompson 2015). Ultimately, we should direct future efforts towards an accurate eddying model with tides."*

Regarding (b), we use *convergence* for lack of a better term. "Model" includes our understanding of the processes and tuning (discussed under R1C20). If this "wide sense" model would diverge with increasing resolution, we would have a serious problem with our understanding of the models behaviour and the importance of the processes included. All we want to show is that this is not the case. We acknowledge that *convergence* often implies convergence towards a known solution in the field of model evaluation and that the term might be misleading. We now have defined what we mean by *convergence* (repeated from R1C20):

*"From 10 km to 1 km, we expect the model solution to be less dependent on resolution, as we start resolving the processes most critical to our problem. Demonstrating convergence of WAOM as a whole [...]"*

and clarified that we only expect the model to converge towards **a** solution, not necessarily reality (at the end of the same paragraph):

*"We note that we do not necessarily expect the model to converge towards the observations, without further calibration."*

We would like to highlight that most studies do not include any kind of grid resolution analysis.

**R2C9** 9) Again in Figure 10, it is not entirely obvious what has been gained by the addition of tides and the use of high resolution. Results are different from previous studies, but not obviously any better. The spatial distribution of re-freezing is rather poorly captured (Figure 11): almost nothing on Amery and Larsen C ice shelves, and very little in the central Ronne Ice Shelf. Many of the early (admittedly) regional models did much better, despite being run at much lower resolution. That again points to the fact that increasing resolution does not necessarily improve results. I agree that a higher resolution model has the potential to improve the representation of reality, because it can resolve more processes, but no model can resolve every process, and the key to getting things right at any resolution is knowledge of how best to parameterise whatever remains in the sub-grid-scale. Arguably the authors have done a better job at that with the 4 km version of WAOM than with the 2 km.

It is well known that eddies, troughs and tides play an important role for ice shelf melting. Our model, for the first time, allows us to explore the impact of these processes on, and their interaction with, ice shelf melting in a circum-Antarctic sense. This is the key strength of our

approach over previous large scale models. Beyond that we also represent processes (tides, eddies permitted by the higher resolution) which were completely excluded from many previous regional studies. These studies might have captured ice shelf melting at high resolution more accurately. However, that does not exclude the possibility that regional models might have been getting the right answer for the wrong reasons - as they missed important processes.

**R2C10** In summary, while this circum-Antarctic model has the potential to be a useful addition to the growing collection of such tools, I feel the authors should do more to critically evaluate their results and explain the impact of including extra processes and using finer resolution. The main issues at present that make the model results questionable for the applications that the authors have in mind are the problems with water column structure (that may be related to sigma-coordinate problems over steep topography) and the curious water mass properties (Figure 6). If the latter are not due to misplotting, then it suggests some serious issues with the model (potentially associated with the surface flux forcing and/or the sigma coordinates). Those really need to be sorted out before the model can be considered fit for purpose.

There has been a plotting mistake in Figure 6 and the corrected figure looks much closer to observations (see R1C23). We substantially strengthened the model evaluation by providing a first bias estimate of the on-shelf ocean (see Sect. Ocean Evaluation). We have elaborated on the purpose of the resolution study (see R1C20, R2C8). We rate attribution of change to individual processes as out of the scope of this study, but propose experiments in more detail to address these questions (see R1C20). WAOM v1.0 indeed has biases (e.g. related to ocean stratification). We communicate these clearly (see Sect. Ocean Evaluation, Discussion: Biases) and propose future development to address them (see Sect. Ocean Evaluation, Discussion: Future Development). WAOM v1.0 is already being used for ice sheet coupling (pers. com. Rupert Gladstone), quantifying the impact of future climate scenarios (pers. com. John Moore) and studying dense water production (pers. Com. Petteri Uotila). Further calibration of WAOM is ongoing within these studies.

**Response to Review #3**

**R3C0** In this paper, the authors report on the development of a new, high-resolution model of the Southern Ocean including its ice shelf cavities. The inclusion of tides represents a major feature of this model and a significant progress in scientific model development. The paper discusses model design and the evaluation of results.
The paper is well written and presents a lot of useful information. Figures are clear and well crafted. I recommend to accept the paper pending revisions guided by the following specific comments. Note that numbers 12 and 19 are a bit more substantial.

We would like to thank the reviewer for this positive feedback. They too rate the development step that WAOM v1.0 represents as worthy for publication.

**R3C1**. Throughout the text, I felt the urge to add a significant amount of hyphens in composite terms like "eddy-scale circulation", "large-scale models", "present-day conditions",

"eddyresolving horizontal resolution", "nearest-neighbour method", „Spin-up procedure",
„depth-averaged temperature" etc. I trust this will be handled by the copy editor at one of the
final stages of publication, but I also encourage the authors to revisit these composits.

We have corrected the listed composites.

**R3C2**. l. 23: To the list of papers trying to predict future changes, you may want to add
Timmermann and Hellmer (2013).
We have added the reference.

**R3C3**. l. 27: It may be worth mentioning that several coupled ice sheet--ocean models exist
already: Timmermann and Goeller (2017) with global ocean (but regional ice sheet) is one
example, your co-author KA Naughten runs another.

We agree and have added (p. 2, line 27):
"[... and coupled ice sheet-ocean models for climate predictions will ultimately need
Antarctic-wide domains (Asay-Davis et al., 2017).] The first realistic, coupled models are
now becoming available (Ralph Timmermann and Goeller 2017; Naughten et al. 2021)."

**R3C4**. l. 29: "augmented" (like an add-on) does not quite match the fact that at least some of
these models were designed with ice shelves included right from the start. Two of the early,
pioneering models of this kind were Beckmann et al. (1999) and Timmermann et al. (2002).

We thank the reviewer for this information. We have adapted the phasing from (p. 2, line 29):
"

" to:

"Many ocean models with pan-Antarctic coverage have either been designed with cavities
from the beginning (Beckmann, Hellmer, and Timmermann 1999; R. Timmermann,
Beckmann, and Hellmer 2002; Hellmer 2004) or augmented by an ice shelf component at a
later stage (e.g. Timmermann et al., 2012; Kusahara and Hasumi, 2013; Dinniman et al.,
2015; Schodlok et al., 2016; Mathiot et al., 2017; Naughten et al., 2018; for review see
Dinniman et al., 2016; Asay-Davis et al., 2017)."

**R3C5**. l. 47: instead of "usually", I would find "often" or maybe even only "sometimes" more
appropriate.

We agree that "often" is more appropriate and have changed the term.

**R3C6**. l. 60: The statement that the model "includes all the model physics of state-of-the-art
regional applications." seems a bit daring to me, given that sea ice (which is commonly
regarded as part of the ocean) is only roughly approximated in this model.

We fully agree with the reviewer and have changed the wording from (p. 3, lines 59-60):

*"The Whole Antarctic Ocean Model (WAOM v1.0) includes tides and an eddy-resolving* *horizontal resolution of 2 km and, thus, includes all the model physics of state-of-the-art* *regional applications."* to:

*"The Whole Antarctic Ocean Model (WAOM v1.0) includes tides and an eddy-resolving* *horizontal resolution of 2 km, both known to be critical to resolve accurate ice shelf melt* *rates from state-of-the-art regional applications."* (already shown under Sect. Ocean Evaluation, Introduction).

**R3C7**. l. 119: "[polynyas] are critical to resolve accurate ice shelf melting in cold regimes": That's what people say. In fact, it is quite en vogue to stress the importance of coastal or flaw polynyas. And it is not totally wrong at all. If you do the budgets though (let's say: for the continental shelf of the southern Weddell Sea), it turns out that the leads in the (vast) pack add up more salt flux through sea ice formation than the (comparatively tiny) coastal polynyas. What does make coastal or flaw polynyas important indeed is the fact that they are persistent and stationary. See, e.g., Haid and Timmermann (JGR 2013). So that statement is not totally wrong, but maybe a tad on the simplifying side.

We thank the reviewer for this clarification. We agree that the statement is to simple and have changed added the following note (p. 6, line 119):

**"***While flaw leads in the vast pack ice are likely to add more salt into the ocean in total (shown for the Weddell See region**, see Haid and Timmermann, 2013), coastal polynyas play a more critical role for regional ice shelf interaction due to their stationary character.***"**

**R3C8.** l. 149: "while Bedmap2 ice thickness data is mostly based on laser altimetry data from 1994 to 1995" Are you sure this is true? Bedmap2 is much younger than this. Please double-check.

We have double checked this. The original statement is true.

Fretwell et al. (2013) states: *"A single gridded dataset of ice thickness derived from satellite altimetry (Griggs and Bamber, 2011) provided full coverage and uniform consistency of all the significant floating ice shelves around Antarctica. This was adopted as the primary ice-thickness data source for these regions."*

Griggs and Bamber (2011) states: *"We present a satellite retrieval of the ice thickness for all Antarctic ice shelves using satellite radar altimeter data from the geodetic phases of the European Remote-sensing Satellite (ERS-1) during 1994–95 supplemented by ICESat data for regions south of the ERS-1 latitudinal limit."*

**R3C9**. l. 166: I believe it should be Z = H (cosG+i sinG) (with brackets)

We have corrected this mistake.

**R3C10**. l. 183 etc: (a) In the "Resolution effects" section, I think it would be very useful to not only discuss resolution-caused changes, but also whether these bring modelled hydrography closer to or further away from observations. Maybe this is easier if this section is moved to the end of the chapter? No preference, just an idea. (b) Judging from Fig. 4, I am not convinved that I find the statement "the model solution [....] converges with increasing resolution" fully justified. We are indeed far away still from an asymptotic behaviour. Which is probably true for the vast majority of models in use today, so I am not criticizing the model here.

Regarding (a): Evaluation and tuning depends on the question you want to answer. Here we focus on developing a tool for process oriented studies. Such studies should aim to get the right answers for the right reasons, i.e. use an eddying model. Future studies aiming to use WAOM as a tool for future prediction, however, should consider the option of using it at lower resolution. Such studies will need to evaluate the model (at different resolutions) depending on their research question. There are many different measures to assess the model. We have focused on melt rates, and indeed, for the single measure total mass loss it looks best for 4 km.

We now have communicated the scope of this study more clearly:

(repeated from Sect. Ocean Evaluation, Introduction): "*This way, we aim to convince the reader that this first version of WAOM is realistic enough to be applied to specific, process oriented studies and to justify further development of our approach.*"

We have also added a discussion around the use and evaluation of WAOM at lower resolutions (repeated from R2C8; p. 24, line 335):

"*[These findings stress the importance of resolving tides at 4 km horizontal resolution or finer in large-scale models.] Studies aiming to use WAOM for future predictions should consider the option of applying it at 10 km or 4 km horizontal resolution. Such studies will need to evaluate the model (at different resolutions) depending on their research question. Judging on the single scale metric of mean ice shelf melting, the 4 km solution of WAOM is closest to the observations (Fig. 4). For process oriented studies, however, we recommend using the 2 km version, as resolving eddies at a kilometer scale resolution is critical for accurate ice shelf-ocean interaction in some regions (Stewart and Thompson 2015). Ultimately, we should direct future efforts towards an accurate eddying model with tides.*"

We acknowledge that the term "convergence" is somewhat misleading (see discussion under R2C8). We now have defined what we mean by "convergence" (repeated from R1C20; p. 10, lines 189-191):

"*From 10 km to 1 km, we expect the model solution to be less dependent on resolution, as we start resolving the processes most critical to our problem. Demonstrating convergence of WAOM as a whole is [...]*"

**R3C11**. l. 207: You may want to finish the sentence with "and the representation of narrow troughs at the continental shelf break (Nakayama et al., 2014)"

We gratefully apply this recommendation (p. 11, line 207):

*"As mentioned earlier, this phenomenon is often associated with shoreward heat transport by eddies that need a grid spacing on the order of 1 km to be resolved by ocean models (Dinniman et al., 2016; Mack et al., 2019) and the representation of narrow troughs at the continental shelf break (Nakayama et al., 2014)."*

**R3C12**. l. 220-222: This passage is not fully convincing. Stronger water mass transformation would (in my view) go via more or saltier HSSW - which (according to their statement a few lines above, and consistent with Fig. 6) is not what the authors find. How does the model form WSBW with S>34.8 if no HSSW with at least the same salinity exists? There has to be a source somewhere, and I do not agree that finding this source can be beyond the scope of this study.

There has been a plotting mistake in Figure 6 (see R1C23). Dense waters shown by the corrected plot can be well explained by boundary and initial conditions (already discussed under R1C23, incl. changes to the text).

**R3C13**. l. 225-227, particularly with regard to "Which of the models is more accurate close to the surface and what is causing the differences is not clear.": A purely observation-based data product (like the World Ocean Atlas) might help.

We are now comparing model predictions against WOA18 climatologies (see Sect. Ocean Evaluation, Results: On-shelf hydrography, Results: Off-shelf hydrography). We have identified a warm bias at the surface and discussed this bias in the light of ice shelf basal melting (taken from Sect. Ocean Evaluation, Discussion: Biases):

*"The reported warm bias at the surface (Fig. 5) could also be linked to reduced HSSW formation. WAOM predicts elevated melt rates right at the ice front in most regions (close to coastal polynyas; Fig. 11) and ISW has been shown to be able to suppress dense water formation (Williams et al. 2016; Silvano et al. 2018)."*

and proposed steps for future development (taken from Sect. Ocean Evaluation, Discussion: Future development):

*"Second priority should be given to the calibration of the surface heat flux, which is likely to reduce the warm surface bias. The warm bias towards the surface can not be explained by initial and boundary conditions, as ECCO2's upper ocean conditions are more realistic (see Fig. 10). Also, 2007 has not been an anomalously warm year (e.g. measured by sea ice extent; see Parkinson, 2019), rendering interannual variability as an unlikely source. Instead, we suspect the applied surface flux schemes to be responsible. A similar scheme is known to overestimate annual heat flux into the ocean by about 50% (Jendersie et al. 2018). While we aim to account for this by reducing positive heat flux into the ocean by half (see Methods, Sect. 2.4), the approach has not been tested for pan-Antarctic domains."*

**R3C14**. l. 232/233: "The z-like signature of ISW in the Ross Sea is likely caused by continued mixing of ISW from one ice shelf inside the cavity of another ice shelf downstream and this further supports the presence of ice shelf teleconnections." I think the statement

here could/should be more precise. The idea is that these patterns are signatures of meltwater originating from ice shelves upstream from Ross Ice Shelf, right? So, this would be meltwater from the Amundsen / Bellingshausen Seas? I am not sure whether this explains the structure in the Ross ISW, but the idea of a teleconnection between this and those is supported by the findings of Nakayama et al. (2020).

This part of our results is a very small detail and we agree that the evidence for this hypothesis is very thin. We have removed this hypothesis from the manuscript (no longer present in the new ocean evaluation presented under Sect. Ocean Evaluation, Results: On-shelf hydrography, Results: Off-shelf hydrography).

**R3C15**. l. 239-248: To me it seems as if this whole paragraph calls for a model-to-data comparison instead of (or in addition to) model-to-model.

We now have compared the model to observations, instead of SOSE. We have identified overly mixed conditions in WAOM (see Sect. Ocean Evaluation, Results: On-shelf hydrography), discussed the consequences of this bias for predicted ice shelf melt rates and provided suggestions for future development (see Sect. Ocean Evaluation, Discussion: Biases, Discussion: Future development).

**R3C16**. Figure 9: Having the ice shelf on the left of the plot in the section AND in the map would be nice.

We agree and have adapted the insets accordingly (see new Fig. 11).

**R3C17**. Figure 9 again: Is the very sharp front in the (c) panels a simulation result or are we too close to the open boundary / sponge layer here?

This is a model result. The choice of colors scale (only up to -1.5) exaggerates the front, but is necessary to display the stratification inside the cavity. We have added a note to the text (taken from Sect. Ocean Evaluation, Results: On-shelf hydrography; additions in bold):

"*In this region, CDW is held back from entering the continental shelf by a sharp front (the Antarctic Slope Front; **exaggerated by the choice of color scale**; in agreement with, e.g. Guo et al., 2019, their Fig. 2).*"

**R3C18**. l.283: I think it should be "in agreement or close to FOR others"

We have corrected this mistake.

**R3C19**. l.310: The finding that strong ice-shelf basal melting near the ice front in this model is a widespread feature needs some discussion in context with numerics / sigma coordinates. Is there any risk that the particularities of terrain-following coordinates create a certain tendency / bias here? If mixing is not carefully controlled and ideally rotated to density surfaces (instead following lateral coordinate lines), a spurious exchange between the openocean surface and the ocean in touch with ice the shelf base near the ice front may be something to keep an eye on, I think.

We understand the reviewers' concern. A smooth and sloping representation of the ice front in sigma coordinates favours elevated melt rates (also see discussion under R2C1). More ice shelf area is exposed to warm surface waters (geometrical consequence) and baroclinic transport is eased (dynamic consequence, see Wåhlin et al., 2020). However, the realism of frontal processes in models with different coordinates is an active research question. Malyarenko et al.( 2019), for example, suggest that a smooth ice front representation in sigma coordinates actually compensates for unresolved processes that enhance surface water intrusion (the right outcome for the wrong reason). WAOM's results stress the timeliness of research in this area.

We already explicitly note this point under future studies (p. 25, line 395):

*"Studying individual aspects of the model will help gain trust in quantitative results. Schnaase and Timmermann (2019), for example, show that artificially deepening the water column thickness near grounding zones (necessary for numerical stability), does not affect ice shelf average melt rates, and Malyarenko et al. (2019) suggest that the unrealistic ice front representation in sigma-coordinates, could actually account for unresolved small scale processes."*

And now have added a discussion around this point (in addition to the biases presented under Sect. Ocean Evaluation, Discussion: Biases; p. 24, lines 342-357):

*"We note that elevated frontal melting in WAOM is likely favoured by its representation of the ice front. A sloping and smooth representation of the vertical cliff face exposes more ice shelf area to warm surface waters (a geometrical consequence) and eases baroclinic transport (Wåhlin et al. 2020). Ice shelf frontal processes and their representation in models, however, are not well explored. There is evidence, for example, that a smooth representation of the ice front in sigma coordinates actually compensates for an unresolved wedge mechanism that favours intrusions of water surface waters under the ice (Malyarenko et al. 2019). The results presented in this study stress the importance of further research in this area."*

**R3C20**. l. 313, "accurate polynyas": Whether these are better in terms of giving the correct buoyancy flux than a prognostic sea-ice model may still be debatable. So, compared to resolution and tides, this may be a weaker point in the list of strengths of this model. Personally, I would concentrate on the "real strengths", with tides probably being the leader here, followed by resolution, and tune down the enthusiasm about the model's approach to sea ice processes. This may be a matter of scientific taste though and I do not insist that the authors follow my suggestion.

We agree with the reviewer about the list of strengths and their order. Accurate amount and position of the surface fluxes resulting from polynya activity does not necessarily translate into accurate polynya-driven effects on ice shelf-ocean interaction. The lack of HSSW in our model is evidence for this point. We now have removed accurate polynya activity from the strengths of the model and have put tides upfront (p.23, line 313):

*"Compared to other models, WAOM includes tides and an eddy resolving resolution, tides and accurate polynyas, a first for a circumAntarctic ice-ocean simulation."*

We already have specified which part of the polynyas are accurate with our approach (repeated from R1C12; p. 4, lines 116-118):

*"Prescribed surface buoyancy fluxes, rather than including a sea ice model, is, hence, more likely to capture the position and strength of coastal polynyas."*

**R3C21**. l 330: It is SUCH a pity that results from coarser resolution or deactivated tides are not shown!

We do show the impact of resolution as difference plots in Figure 5. Tidal modulations are the main results of a separate paper (Richter et al. in review) and can not be included here. We acknowledge that referring to these results without showing them is unsatisfactory for the reader. We have now relaxed the related statement (continental shelf cooling due to better resolved tidal effects) to a hypothesis and base this hypothesis purley on results shown in this study (resolution effects) and results from previous studies.

For this we have removed some results (p. 10, lines 197-203):

*"When increasing the grid resolution from 10 km to 4 km, the shelf ocean cools at many places, most likely due to better resolved tidal processes. We find that resolution-induced changes in depth averaged temperature are governed by changes in the bottom sigma layer (not shown). Figure 5 shows how bottom sigma layer temperatures change with increasing resolution. The ocean cools at many places when refining the horizontal grid spacing from 10 km to 4 km (Fig. 5a). Differences exceed 1 ◦C in the eastern Bellingshausen Sea and in the eastern Ross Sea, and are on the order of 0.25 ◦C in the Amundsen Sea and around the East Antarctic coastline. We attribute most of these changes to better resolved tidal processes, based on additional sensitivity experiments that remove the tides (not shown). For example, activating tides in the model at 4 km resolution also leads to warm water intrusions that extend under the north-western part of the Ronne Ice Shelf and ocean temperature changes 200 that resemble a dipole pattern in the eastern Ross Sea. Also, in both cases, effects are well constrained by the continental shelf break, where tides start to weaken with increasing water column thickness. Finally, the overall reduction in continental shelf temperature has a similar magnitude in both experiments."*

And rewritten the discussion around this (p.23, line 330):
*"The overall picture, however, is dominated by different processes. Compared to our most complex simulation (2 km resolution and with tides), coarsening the horizontal resolution or deactivating tides (not shown) leads to a warmer continental shelf with similar regional changes. Increasing the resolution leads to an overall cooling of the continental shelf. Similar studies without tides only report a warming with increasing resolution (Nakayama et al. 2014; Dinniman et al. 2016), hinting towards better resolved tidal processes to be the cause."*

**R3C22**. l. 334, "spuriously low conversion rate of heat into ice shelf melting": This point I don't see, because even if you have a spuriously low ocean-to-ice heat flux, the transport of warm water onto the continental shelf is still the same, isn't it?

This is a misunderstanding. The reviewer is referring to heat transport across the shelf break, while we mean across the cavity entrance. Enhanced melt rates would cool the continental shelf ocean (given that heat flux across the surface and shelf break are constant). We have rewritten this discussion for clarification (p. 23, lines 332-335).

""

*"A cooling continental shelf could either be realized by decreased heat flux onto the shelf, increased heat flux to the atmosphere/sea-ice or increased heat flux into the ice. (Stewart, Klocker, and Menemenlis 2018) find that tide driven heat flux across the shelf break is mostly balanced by mean flow and, in our simulation, melt rates also decrease with increasing resolution (Fig. 4) and changes in temperature are strongest outside the cavities (Fig. 5). Hence, we hypothesise that increased vertical mixing due to better resolved tidal processes are responsible for the reported continental shelf cooling with increasing resolution."*

And leave the confirmation of this hypothesis to future studies (p. 26, line 414):

*"To confirm our hypothesis that tidal mixing governs the reported cooling of the continental shelf ocean with increasing horizontal resolution, future studies should perform additional experiments without tides and apply heat flux analysis across the shelf break, surface and cavity entrance."*

**R3C23**. l. 355-357: I will shamelessly advertise RTopo-2 here. That said, it is highly unlikely that everything in RTopo-2 is perfect.

We thank the reviewer for the advice. We do not have a preference. Both are state-of-the-art products. Now there is also a BedMachine (Morlighem et al. 2020). In the text, the discussion has moved away from biases in bathymetry (see Sect. Ocean Evaluation, Discussion: Biases).

**R3C24**. l. 359: In the list of studies on interaction between sea ice and ice shelves, you may REALLY want to add Timmermann and Hellmer (2013).

We gratefully follow this advice (p. 25, line 359; additions in bold):
*"[...] having motivated many previous studies to include sea ice models (e.g. Hellmer, 2004; Timmermann et al., 2012; **Timmermann and Hellmer, 2013**; Naughten et al., 2018)."*

**R3C25**. l. 360-362, "This study, however, prioritises accurate polynyas by prescribing surface fluxes from sea ice observations. While this is likely to result in more accurate melt rates at the base of the ice shelves" : I am not sure I agree with this. Having the polynyas at

the right places is a good step, but the fluxes computed from there are probably much less well constrained.

We fully agree (see discussions under R1C12, R3C20). We now also have changed this part of the discussion to reflect this (p. 24, lines 360-361) from:

*" "* to:

*"This study, however, follows an approach that prescribes surface fluxes from sea ice observations to accurately capture the position and strength of coastal polynyas. While this is a major component towards accurate ice shelf melt rates, [...]"*

**R3C26**. l. 367/368, "This design, however, has been chosen to simplify future efforts that aim to couple WAOM with models of Antarctic ice sheet flow": This has just been said (two sentences back).

We agree and have removed the repetition from the beginning of the paragraph (p. 24, line 365):

*"The many wasted land cells in WAOM's domain could also be considered a limitation."*

**R3C27**. l. 406: "harness": Sure? Maybe "harvest"?

We believe the use of "harness" is more appropriate here.

**R3C28**. l. 420: Limitations of using just one particular year over and over again as atmospheric forcing need to be discussed. Think of periodic modes of variability and how each of these modes is randomly sampled in one particular phase and then repeated over and over again.

The consequences of missing interannual variability in the model have already been discussed under R1C3 (incl. changes to the text). This discussion includes atmospheric forcing.

**Figures**

[revised manuscript text omitted]

Asay-Davis, Xylar S., Nicolas C. Jourdain, and Yoshihiro Nakayama. 2017. "Developments in Simulating and Parameterizing Interactions Between the Southern Ocean and the Antarctic Ice Sheet." *Current Climate Change Reports* 3 (4): 316–29. https://doi.org/10.1007/s40641-017-0071-0.

Beckmann, A., Hartmut Hellmer, and Ralph Timmermann. 1999. "A Numerical Model of the Weddell Sea: Large Scale Circulation and Water Mass Distribution." *Journal of Geophysical ResearchC10)* 104: 23375–91.

Bett, David T., Paul R. Holland, Alberto C. Naveira Garabato, Adrian Jenkins, Pierre Dutrieux, Satoshi Kimura, and Andrew Fleming. 2020. "The Impact of the Amundsen Sea Freshwater Balance on Ocean Melting of the West Antarctic Ice Sheet." *Journal of Geophysical Research: Oceans* 125 (9): e2020JC016305. https://doi.org/10.1029/2020JC016305.

Cougnon, E. A., B. K. Galton-Fenzi, A. J. S. Meijers, and B. Legrésy. 2013. "Modeling Interannual Dense Shelf Water Export in the Region of the Mertz Glacier Tongue (1992–2007)." *Journal of Geophysical Research: Oceans* 118 (10): 5858–72. https://doi.org/10.1002/2013JC008790.

Davis, Peter E. D., Adrian Jenkins, Keith W. Nicholls, Paul V. Brennan, E. Povl Abrahamsen, Karen J. Heywood, Pierre Dutrieux, Kyoung-Ho Cho, and Tae-Wan Kim. 2018. "Variability in Basal Melting Beneath Pine Island Ice Shelf on Weekly to Monthly Timescales." *Journal of Geophysical Research: Oceans* 123 (11): 8655–69. https://doi.org/10.1029/2018JC014464.

Depoorter, M. A., J. L. Bamber, J. A. Griggs, J. T. M. Lenaerts, S. R. M. Ligtenberg, M. R. van den Broeke, and G. Moholdt. 2013. "Calving Fluxes and Basal Melt Rates of Antarctic Ice Shelves." *Nature* 502 (7469): 89–92. https://doi.org/10.1038/nature12567.

Dinniman, Michael S., Xylar Asay-Davis, Benjamin Galton-Fenzi, Paul Holland, Adrian Jenkins, and Ralph Timmermann. 2016. "Modeling Ice Shelf/Ocean Interaction in Antarctica: A Review." *Oceanography* 29 (4): 144–53. https://doi.org/10.5670/oceanog.2016.106.

Dinniman, Michael S., John M. Klinck, Le-Sheng Bai, David H. Bromwich, Keith M. Hines, and David M. Holland. 2015. "The Effect of Atmospheric Forcing Resolution on Delivery of Ocean Heat to the Antarctic Floating Ice Shelves." *Journal of Climate* 28 (15): 6067–85. https://doi.org/10.1175/JCLI-D-14-00374.1.

Fretwell, P., H. D. Pritchard, D. G. Vaughan, J. L. Bamber, N. E. Barrand, R. Bell, C. Bianchi, et al. 2013. "Bedmap2: Improved Ice Bed, Surface and Thickness Datasets for Antarctica." *The Cryosphere* 7 (1): 375–93. https://doi.org/10.5194/tc-7-375-2013.

Galton-Fenzi, B. K., J. R. Hunter, R. Coleman, S. J. Marsland, and R. C. Warner. 2012. "Modeling the Basal Melting and Marine Ice Accretion of the Amery Ice Shelf." *Journal of Geophysical Research: Oceans* 117 (C9). https://doi.org/10.1029/2012JC008214.

Greene, Chad A., Donald D. Blankenship, David E. Gwyther, Alessandro Silvano, and Esmee van Wijk. 2017. "Wind Causes Totten Ice Shelf Melt and Acceleration." *Science Advances* 3 (11). https://doi.org/10.1126/sciadv.1701681.

Griggs, J.A., and J.L. Bamber. 2011. "Antarctic Ice-Shelf Thickness from Satellite Radar Altimetry." *Journal of Glaciology* 57 (203): 485–98. https://doi.org/10.3189/002214311796905659.

Guo, Guijun, Jiuxin Shi, Libao Gao, Takeshi Tamura, and Guy D. Williams. 2019. "Reduced Sea Ice Production Due to Upwelled Oceanic Heat Flux in Prydz Bay, East Antarctica." *Geophysical Research Letters* 46 (9): 4782–89.

https://doi.org/10.1029/2018GL081463.

Gwyther, D. E., B. K. Galton-Fenzi, J. R. Hunter, and J. L. Roberts. 2014. "Simulated Melt Rates for the Totten and Dalton Ice Shelves." *Ocean Science* 10 (3): 267–79. https://doi.org/10.5194/os-10-267-2014.

Gwyther, David E., Terence J. O'Kane, Benjamin K. Galton-Fenzi, Didier P. Monselesan, and Jamin S. Greenbaum. 2018. "Intrinsic Processes Drive Variability in Basal Melting of the Totten Glacier Ice Shelf." *Nature Communications* 9 (1): 3141. https://doi.org/10.1038/s41467-018-05618-2.

Gwyther, David E., Erica A. Spain, Peter King, Damien Guihen, Guy D. Williams, Eleri Evans, Sue Cook, Ole Richter, Benjamin K. Galton-Fenzi, and Richard Coleman. 2020. "Cold Ocean Cavity and Weak Basal Melting of the Sørsdal Ice Shelf Revealed by Surveys Using Autonomous Platforms." *Journal of Geophysical Research: Oceans* 125 (6): e2019JC015882. https://doi.org/10.1029/2019JC015882.

Haid, V., and R. Timmermann. 2013. "Simulated Heat Flux and Sea Ice Production at Coastal Polynyas in the Southwestern Weddell Sea." *Journal of Geophysical Research: Oceans* 118 (5): 2640–52. https://doi.org/10.1002/jgrc.20133.

Hattermann, Tore. 2018. "Antarctic Thermocline Dynamics along a Narrow Shelf with Easterly Winds." *Journal of Physical Oceanography* 48 (10): 2419–43. https://doi.org/10.1175/JPO-D-18-0064.1.

Hellmer, H. H. 2004. "Impact of Antarctic Ice Shelf Basal Melting on Sea Ice and Deep Ocean Properties." *Geophysical Research Letters* 31 (10). https://doi.org/10.1029/2004GL019506.

Jacobs, Stanley S., H. H. Helmer, C. S. M. Doake, A. Jenkins, and R. M. Frolich. 1992. "Melting of Ice Shelves and the Mass Balance of Antarctica." *Journal of Glaciology* 38 (130): 375–87. https://doi.org/10.3189/S0022143000002252.

Jacobs, Stanley S., Adrian Jenkins, Claudia F. Giulivi, and Pierre Dutrieux. 2011. "Stronger Ocean Circulation and Increased Melting under Pine Island Glacier Ice Shelf." *Nature Geoscience* 4 (8): 519. https://doi.org/10.1038/ngeo1188.

Jendersie, Stefan, Michael J. M. Williams, Pat J. Langhorne, and Robin Robertson. 2018. "The Density-Driven Winter Intensification of the Ross Sea Circulation." *Journal of Geophysical Research: Oceans* 123 (11): 7702–24. https://doi.org/10.1029/2018JC013965.

Kimura, Satoshi, Adrian Jenkins, Heather Regan, Paul R. Holland, Karen M. Assmann, Daniel B. Whitt, Melchoir Van Wessem, Willem Jan van de Berg, Carleen H. Reijmer, and Pierre Dutrieux. 2017. "Oceanographic Controls on the Variability of Ice-Shelf Basal Melting and Circulation of Glacial Meltwater in the Amundsen Sea Embayment, Antarctica." *Journal of Geophysical Research: Oceans* 122 (12): 10131–55. https://doi.org/10.1002/2017JC012926.

King, Matt A., and Laurie Padman. 2005. "Accuracy Assessment of Ocean Tide Models around Antarctica." *Geophysical Research Letters* 32 (23). https://doi.org/10.1029/2005GL023901.

Kusahara, Kazuya, and Hiroyasu Hasumi. 2013. "Modeling Antarctic Ice Shelf Responses to Future Climate Changes and Impacts on the Ocean." *Journal of Geophysical Research: Oceans* 118 (5): 2454–75. https://doi.org/10.1002/jgrc.20166.

Kusahara, Kazuya, Hiroyasu Hasumi, and Takeshi Tamura. 2010. "Modeling Sea Ice Production and Dense Shelf Water Formation in Coastal Polynyas around East Antarctica." *Journal of Geophysical Research: Oceans* 115 (C10). https://doi.org/10.1029/2010JC006133.

Liu, Yan, John C. Moore, Xiao Cheng, Rupert M. Gladstone, Jeremy N. Bassis, Hongxing Liu, Jiahong Wen, and Fengming Hui. 2015. "Ocean-Driven Thinning Enhances Iceberg Calving and Retreat of Antarctic Ice Shelves." *Proceedings of the National Academy of Sciences* 112 (11): 3263–68. https://doi.org/10.1073/pnas.1415137112.

Lüpkes, Christof, and Gerit Birnbaum. 2005. "'Surface Drag in the Arctic Marginal Sea-Ice Zone: A Comparison of Different Parameterisation Concepts.'" *Boundary-Layer Meteorology* 117 (2): 179–211. https://doi.org/10.1007/s10546-005-1445-8.

Mack, Stefanie L., Michael S. Dinniman, John M. Klinck, Dennis J. McGillicuddy, and Laurence Padman. 2019. "Modeling Ocean Eddies on Antarctica's Cold Water Continental Shelves and Their Effects on Ice Shelf Basal Melting." *Journal of Geophysical Research: Oceans* 124 (7): 5067–84. https://doi.org/10.1029/2018JC014688.

Malyarenko, A., N. J. Robinson, M. J. M. Williams, and P. J. Langhorne. 2019. "A Wedge Mechanism for Summer Surface Water Inflow Into the Ross Ice Shelf Cavity." *Journal of Geophysical Research: Oceans* 124 (2): 1196–1214. https://doi.org/10.1029/2018JC014594.

Mathiot, Pierre, Bernard Barnier, Hubert Gallée, Jean Marc Molines, Julien Le Sommer, Mélanie Juza, and Thierry Penduff. 2010. "Introducing Katabatic Winds in Global ERA40 Fields to Simulate Their Impacts on the Southern Ocean and Sea-Ice." *Ocean Modelling* 35 (3): 146–60. https://doi.org/10.1016/j.ocemod.2010.07.001.

Mathiot, Pierre, Adrian Jenkins, Christopher Harris, and Gurvan Madec. 2017. "Explicit Representation and Parametrised Impacts of under Ice Shelf Seas in the $z^*$ Coordinate Ocean Model NEMO 3.6." *Geoscientific Model Development* 10 (7): 2849–74. https://doi.org/10.5194/gmd-10-2849-2017.

Mellor, G. L., T. Ezer, and L-Y. Oey. 1994. "The Pressure Gradient Conundrum of Sigma Coordinate Ocean Models." *Journal of Atmospheric and Oceanic Technology* 11 (4): 1126–34. https://doi.org/10.1175/1520-0426(1994)011<1126:TPGCOS>2.0.CO;2.

Meneghello, Gianluca, John Marshall, Jean-Michel Campin, Edward Doddridge, and Mary-Louise Timmermans. 2018. "The Ice-Ocean Governor: Ice-Ocean Stress Feedback Limits Beaufort Gyre Spin-Up." *Geophysical Research Letters* 45 (20): 11,293-11,299. https://doi.org/10.1029/2018GL080171.

Meneghello, Gianluca, John Marshall, Mary-Louise Timmermans, and Jeffery Scott. 2018. "Observations of Seasonal Upwelling and Downwelling in the Beaufort Sea Mediated by Sea Ice." *Journal of Physical Oceanography* 48 (4): 795–805. https://doi.org/10.1175/JPO-D-17-0188.1.

Morlighem, Mathieu, Eric Rignot, Tobias Binder, Donald Blankenship, Reinhard Drews, Graeme Eagles, Olaf Eisen, et al. 2020. "Deep Glacial Troughs and Stabilizing Ridges Unveiled beneath the Margins of the Antarctic Ice Sheet." *Nature Geoscience* 13 (2): 132–37. https://doi.org/10.1038/s41561-019-0510-8.

Morrison, A. K., A. McC Hogg, M. H. England, and P. Spence. 2020. "Warm Circumpolar Deep Water Transport toward Antarctica Driven by Local Dense Water Export in Canyons." *Science Advances* 6 (18): eaav2516. https://doi.org/10.1126/sciadv.aav2516.

Nakayama, Y., D. Menemenlis, M. Schodlok, and E. Rignot. 2017. "Amundsen and Bellingshausen Seas Simulation with Optimized Ocean, Sea Ice, and Thermodynamic Ice Shelf Model Parameters." *Journal of Geophysical Research: Oceans* 122 (8): 6180–95. https://doi.org/10.1002/2016JC012538.

Nakayama, Y., R. Timmermann, M. Schröder, and H.H. Hellmer. 2014. "On the Difficulty of Modeling Circumpolar Deep Water Intrusions onto the Amundsen Sea Continental Shelf." *Ocean Modelling* 84 (December): 26–34. https://doi.org/10.1016/j.ocemod.2014.09.007.

Naughten, Kaitlin A., Katrin J. Meissner, Benjamin K. Galton-Fenzi, Matthew H. England, Ralph Timmermann, Hartmut H. Hellmer, Tore Hattermann, and Jens B. Debernard. 2018. "Intercomparison of Antarctic Ice-Shelf, Ocean, and Sea-Ice Interactions Simulated by MetROMS-Iceshelf and FESOM 1.4." *Geoscientific Model Development* 11 (4): 1257–92. https://doi.org/10.5194/gmd-11-1257-2018.

Naughten, Kaitlin A., Jan De Rydt, S. H. R. Rosier, P. R. Holland, and Jeff K. Ridley. 2021. "Two-Timescale Response of a Large Antarctic Ice Shelf to Climate Change." https://doi.org/in press.

Nicholls, Keith W., Svein Østerhus, Keith Makinson, Tor Gammelsrød, and Eberhard Fahrbach. 2009. "Ice-ocean Processes over the Continental Shelf of the Southern Weddell Sea, Antarctica: A Review." *Reviews of Geophysics* 47 (3).

https://doi.org/10.1029/2007RG000250.

Nøst, O. A., M. Biuw, V. Tverberg, C. Lydersen, T. Hattermann, Q. Zhou, L. H. Smedsrud, and K. M. Kovacs. 2011. "Eddy Overturning of the Antarctic Slope Front Controls Glacial Melting in the Eastern Weddell Sea." *Journal of Geophysical Research: Oceans* 116 (C11). https://doi.org/10.1029/2011JC006965.

Padman, Laurie, and Helen Amanda Fricker. 2005. "Tides on the Ross Ice Shelf Observed with ICESat." *Geophysical Research Letters* 32 (14). https://doi.org/10.1029/2005GL023214.

Richter, Ole, David E. Gwyther, Matt A. King, and Benjamin K. Galton-Fenzi. in review. "Tidal Modulation of Antarctic Ice Shelf Melting." *The Cryosphere Discussions*, 1–32. https://doi.org/10.5194/tc-2020-169.

Rignot, E., Stanley S. Jacobs, J. Mouginot, and B. Scheuchl. 2013. "Ice-Shelf Melting Around Antarctica." *Science* 341 (6143): 266–70. https://doi.org/10.1126/science.1235798.

Roquet, Fabien, Guy Williams, Mark A. Hindell, Rob Harcourt, Clive McMahon, Christophe Guinet, Jean-Benoit Charrassin, et al. 2014. "A Southern Indian Ocean Database of Hydrographic Profiles Obtained with Instrumented Elephant Seals." *Scientific Data* 1 (1): 140028. https://doi.org/10.1038/sdata.2014.28.

Schmidtko, Sunke, Karen J. Heywood, Andrew F. Thompson, and Shigeru Aoki. 2014. "Multidecadal Warming of Antarctic Waters." *Science* 346 (6214): 1227–31. https://doi.org/10.1126/science.1256117.

Schmidtko, Sunke, Gregory C. Johnson, and John M. Lyman. 2013. "MIMOC: A Global Monthly Isopycnal Upper-Ocean Climatology with Mixed Layers." *Journal of Geophysical Research: Oceans* 118 (4): 1658–72. https://doi.org/10.1002/jgrc.20122.

Schnaase, Frank, and Ralph Timmermann. 2019. "Representation of Shallow Grounding Zones in an Ice Shelf-Ocean Model with Terrain-Following Coordinates." *Ocean Modelling* 144 (December): 101487. https://doi.org/10.1016/j.ocemod.2019.101487.

Schodlok, M. P., D. Menemenlis, and E. J. Rignot. 2016. "Ice Shelf Basal Melt Rates around Antarctica from Simulations and Observations." *Journal of Geophysical Research: Oceans* 121 (2): 1085–1109. https://doi.org/10.1002/2015JC011117.

Shchepetkin, Alexander F., and James C. McWilliams. 2003. "A Method for Computing Horizontal Pressure-Gradient Force in an Oceanic Model with a Nonaligned Vertical Coordinate." *Journal of Geophysical Research: Oceans* 108 (C3). https://doi.org/10.1029/2001JC001047.

Sikirić, Mathieu Dutour, Ivica Janeković, and Milivoj Kuzmić. 2009. "A New Approach to Bathymetry Smoothing in Sigma-Coordinate Ocean Models." *Ocean Modelling* 29 (2): 128–36. https://doi.org/10.1016/j.ocemod.2009.03.009.

Silvano, Alessandro, Stephen R. Rintoul, Beatriz Peña-Molino, and Guy D. Williams. 2017. "Distribution of Water Masses and Meltwater on the Continental Shelf near the Totten and Moscow University Ice Shelves." *Journal of Geophysical Research: Oceans* 122 (3): 2050–68. https://doi.org/10.1002/2016JC012115.

Silvano, Alessandro, Stephen Rich Rintoul, Beatriz Peña-Molino, William Richard Hobbs, Esmee van Wijk, Shigeru Aoki, Takeshi Tamura, and Guy Darvall Williams. 2018. "Freshening by Glacial Meltwater Enhances Melting of Ice Shelves and Reduces Formation of Antarctic Bottom Water." *Science Advances* 4 (4): eaap9467. https://doi.org/10.1126/sciadv.aap9467.

Stewart, Andrew L., Andreas Klocker, and Dimitris Menemenlis. 2018. "Circum-Antarctic Shoreward Heat Transport Derived From an Eddy- and Tide-Resolving Simulation." *Geophysical Research Letters* 45 (2): 834–45. https://doi.org/10.1002/2017GL075677.

Stewart, Andrew L., and Andrew F. Thompson. 2015. "Eddy-mediated Transport of Warm Circumpolar Deep Water across the Antarctic Shelf Break." *Geophysical Research Letters* 42 (2): 432–40. https://doi.org/10.1002/2014GL062281.

Tamura, Takeshi, Kay I. Ohshima, Sohey Nihashi, and Hiroyasu Hasumi. 2011. "Estimation of Surface Heat/Salt Fluxes Associated with Sea Ice Growth/Melt in the Southern

Ocean." *SOLA* 7: 17–20. https://doi.org/10.2151/sola.2011-005.

Thoma, Malte, Adrian Jenkins, David Holland, and Stanley S. Jacobs. 2008. "Modelling Circumpolar Deep Water Intrusions on the Amundsen Sea Continental Shelf, Antarctica." *Geophysical Research Letters* 35 (18). https://doi.org/10.1029/2008GL034939.

Timmermann, R., A. Beckmann, and H. H. Hellmer. 2002. "Simulations of Ice-Ocean Dynamics in the Weddell Sea 1. Model Configuration and Validation." *Journal of Geophysical Research: Oceans* 107 (C3): 10-1-10–11. https://doi.org/10.1029/2000JC000741.

Timmermann, Ralph, and Sebastian Goeller. 2017. "Response to Filchner–Ronne Ice Shelf Cavity Warming in a Coupled Ocean–Ice Sheet Model – Part 1: The Ocean Perspective." *Ocean Science* 13 (5): 765–76. https://doi.org/10.5194/os-13-765-2017.

Timmermann, Ralph, and Hartmut Hellmer. 2013. "Southern Ocean Warming and Increased Ice Shelf Basal Melting in the Twenty-First and Twenty-Second Centuries Based on Coupled Ice-Ocean Finite-Element Modelling." *Ocean Dynamics* 63 (9): 1011–26.

Timmermann, Ralph, Qiang Wang, and Hartmut Hellmer. 2012. "Ice shelf basal melting in a global finite-element sea ice/ice shelf/ocean model." *Annals of Glaciology* 53. https://doi.org/Timmermann, R. , Wang, Q. and Hellmer, H. (2012) Ice shelf basal melting in a global finite-element sea ice/ice shelf/ocean model , Annals of Glaciology, 53 (60) . doi:10.3189/2012AoG60A156 <http://doi.org/10.3189/2012AoG60A156> , hdl:10013/epic.40279.

Wåhlin, A. K., N. Steiger, E. Darelius, K. M. Assmann, M. S. Glessmer, H. K. Ha, L. Herraiz-Borreguero, et al. 2020. "Ice Front Blocking of Ocean Heat Transport to an Antarctic Ice Shelf." *Nature* 578 (7796): 568–71. https://doi.org/10.1038/s41586-020-2014-5.

Williams, G. D., L. Herraiz-Borreguero, F. Roquet, T. Tamura, K. I. Ohshima, Y. Fukamachi, A. D. Fraser, et al. 2016. "The Suppression of Antarctic Bottom Water Formation by Melting Ice Shelves in Prydz Bay." *Nature Communications* 7 (August): 12577. https://doi.org/10.1038/ncomms12577.

---

## Author Response (AR2)

**Response to the second revision of "The Whole Antarctic Ocean Model (WAOM v1.0): Development and Evaluation" by Richter et al.**

We thank the editor and reviewer for their remarks. Our response is in blue text. Our line numbers refer to the marked up manuscript.

**Response to Review**

In this revision the authors have substantially improved the manuscript, particularly via an overhaul of the evaluation of the simulated hydrography on the continental shelf. Unfortunately, this improved evaluation reveals serious biases in the continental shelf water masses. Most notably, there is an almost-complete absence of High Salinity Shelf Waters at sites such as the Ross and Weddell continental shelves, and a lack of Circumpolar Deep Water (CDW) circulating in the Amundsen and Bellingshausen Seas. Although the authors argue that the model is ready for process-oriented studies, my perspective is that the water mass biases make its applications rather limited: the model cannot be used to study dense shelf water formation, and is not well suited to studying CDW-driven melt of Antarctica's most rapidly-melting ice shelves. These biases likely stem from the authors' prescription of the ocean surface fluxes, rather than the more conventional approach simulating the evolution of the sea ice under a prescribed atmosphere.

That said, the authors are open about the model's biases and readiness for addressing science questions, and they identify various experiments and analyses for which the model could be used in its present state. Thus, while I think the community would be better served by the authors publishing a further improved version of the model in which the major hydrographic biases had been somewhat ameliorated, I do not see this as a barrier to publication of the model in its current state.

We would like to highlight that the reviewer agrees that the development step that WAOM v1.0 represents is worthy of publication.

Below I have included another round of comments and questions for the authors. Of these comments, the most significant pertain to the equilibration of the model at 4km and 2km resolutions. As many of these comments may require substantial changes to be made, particularly to the figures, my recommendation is that the manuscript be returned to the authors for further major revisions.

We now have included a statistical measure that shows that the model is near to quasi equilibrium at all resolutions. Further, we argue that the remaining model drift is acceptable for the purpose of this study. All comments have been addressed in detail below.

Comments/questions:

L29-32: This statement is correct, but don't all models fall into one of these categories? I am struggling to understand what exactly the authors are aiming to convey with this sentence.

No, there are also pan-Antarctic ocean models without any kind of ice shelf interaction (explicit or parameterised). We have included a clarifying sentence (here shown in bold, L32-34):

*"Many ocean models with pan-Antarctic coverage have either been designed with cavities from the beginning (Beckmann et al., 1999; Timmermann et al., 2002; Hellmer, 2004) or augmented by an ice shelf component at a later stage (e.g. Timmermann et al., 2012; Kusahara and Hasumi, 2013; Dinniman et al., 2015; Schodlok et al., 2016; Mathiot et al., 2017; Naughten et al., 2018b).* **There also exist stand alone ocean models without explicit or even parameterised ice shelf interaction (e.g. Mazloff, Heimbach, and Wunsch 2010) and most earth system models used for state-of-the-art climate projections do not include an ice shelf component (e.g. Griffies et al. 2016; Dinniman et al. 2016)."**

L48: The authors state that model parameters are "often" calibrated, but then cite only a single study in support of this statement. Does this really constitute "often"?

We agree with the reviewer. We are not aware of another study that performed a calibration as extensive as done by Nakayama et al. (2017). We have reworded the sentence without changing its general statement (L51).

*"Model parameters in regional studies  **can be** calibrated (e.g. Nakayama et al., 2017), but to approach similar efforts with large scale models, suitable Antarctic-wide observations need to be compiled first."*

L50: Again "often", this time with just two examples. If these were review articles then this would be reasonable, but these are just regular scientific studies.

We agree. We have rephrased the sentence without modifying its general statement (L54).

*"For this purpose, previous studies have  utilised ice shelf melt rates derived from satellite observations and models of firn processes (e.g. Schodlok et al., 2016), and selected Southern Ocean quantities from observations and reanalysis products (e.g. Naughten et al., 2018b)."*

L97-111: Have the authors tested the model with uniform stratification and no forcing to establish that their topographic smoothing and density Jacobian formulation of the pressure gradient force suppress spurious along-slope currents?

Yes, we have performed a zero-forcing experiment for the 10 km resolution version of the model and included a sentence about the magnitude of the spurious currents as compared with the forced simulation (L106-115):

"*We apply the Mellor-Ezer-Oey algorithm (Mellor et al., 1994), which is well established for bathymetry smoothing (Sikiric et al., 2009). We smooth the bathymetry and water column thickness directly until a maximum slope factor r = (hi − hi+1)/(hi + hi+1) ≤ 0.3 is satisfied for both. The ice draft is then redefined as the superposition of bed and water column thickness.  This is a well established procedure to minimise spurious currents in regional ice shelf-ocean configurations (Galton-Fenzi et al., 2012; Cougnon et al., 2013; Gwyther et al., 2014). An experiment at 10 km resolution with uniform stratification and no forcing produced negligible currents in most regions. The only spurious currents of note on or near the continental shelf are along part of the Amundsen-Bellingshausen shelf break, which may explain some of the discrepancy in hydrographic conditions in this region (described in Sec. 3.4).*"

We refer back to this point when discussion overly mixed conditions on the shelf (L413-417):

"*[...] CDW enters the shelf, but gets mixed away too readily before reaching the ice (Fig. 9c). Indeed, WAOM is overly mixed in many regions (incl. the Bellingshausen Seas; see Fig. 12). **Spurious currents from pressure gradient force errors may explain part of the discrepancy in the Amundsen-Bellingshausen Seas, but not in other regions (see Sec. 2.3).** [...]*"

and future work (L479-481):

"*To improve WAOM v1.0 (focused on accurate sub-ice shelf melting) future development should focus on reducing mixing [...] Finally, the sensitivity of stratification to different slope factors (Haney factors) should be explored. Spurious mixing at steep sloping topography (related to pressure gradient force errors in sigma coordinate models; discussed earlier) is sensitive to the degree of smoothing. Our smoothing procedure is similar to regional studies and the smoothing algorithm has been shown to perform well for a realistic, complex case without ice (The Adriatic Sea, see Sikiri ́c et al., 2009). **However, spurious currents in our model are significant along the shelf break of the Amundsen-Bellingshausen Seas, possibly reducing the stratification on the adjacent continental shelf.**  Other pan-Antarctic studies have chosen different routines and algorithms and do not report overly mixed conditions (Naughten et al., 2018b). [...]*"

L130: I suggest an alternative naming to "heat flux into the ocean". Based on the units and context, I infer that this quantity is some kind of restoring coefficient.

Yes, in ROMS it is actually referred to as surface net heat flux sensitivity to SST.

We have changed the text accordingly (L137 f.):

*"Further, to avoid model drift, the surface ocean is relaxed to the solution from SOSE (Mazloff et al., 2010), using a*  **surface net heat flux sensitivity to SST** *of 40 W m−2 ◦C−1 and … "*

Fig. 2: The 10km run looks, to my eye, to have reached equilibrium, but it would be appropriate to include some statistical measure of this. The 4km run looks like it may still be drifting away from the 10km run - have the authors tried checking whether there is any statistically significant trend in the difference between the 10km and 4km melt rates at the end of the integration period?

The drift in the monthly mean melting for both the 10km and 4km runs is less than 3% across the last year of integration. Further, Dinniman et al. (2015) uses a comparable model to perform sensitivity studies, which is generally more delicate than model development and evaluation. They are satisfied with a remaining interannual melt variability of 1%. Therefore, we rate our remaining model drift as acceptable for the purpose of this study, that is development and evaluation.
However, we acknowledge that 3% drift is not perfectly equilibrated, but rather *close to* steady state. We have changed our wording accordingly and included the above outlined statistical measure in the text.

L149 f.: *"Forcing with single year conditions captures daily to seasonal variability, while allowing us to run the model **close** to quasi-equilibrium with our given supercomputing resources."*

L155-159: *"The 10 km version of the model is integrated for 5 years, before the on shelf ocean*  *is near to a quasi equilibrium and its solution is used to initialise the 4 km run. Analogously, the 4 km run is stepped forward in time for 2 years before the final 2 km simulation is initiated and integrated for another year and three months. **Interannual monthly mean melting at each resolution drifts by less than 3% at the end of the integration period, which we rate as acceptable for the purpose of this study.**"*

L530 f.: * We have brought the model close to equilibrium at 2 km grid spacing."*

Additionally, I would strongly encourage the authors to consider whether other measures of shelf water masses, e.g. mean bottom temperatures or salinities, indicate that the simulations have reached equilibrium.

We thank the reviewer for this suggestion. Ice shelf basal melting, however, is a consolidated representation of continental shelf ocean conditions and the central quantity of this study. Like previous studies with comparable models (see, e.g., Dinniman et al. 2015; Kusahara and Hasumi 2013; Naughten et al. 2018), we are convinced that the evolution of ice shelf melting is sufficient to present the spin up of models of ocean ice shelf interaction.

L187: Why only the summer mean? The winter conditions are arguably more important for some processes, e.g. formation of dense shelf waters.

This is true, but we trust the observational products only for summertime where they would have primarily been sampled. This has been elaborated in the preceding paragraph (L184 f.):

"*WOA18 is most accurate in summer, when sea ice has its minimum extent and the vast majority of observations are taken.*"

No changes have been made to the text.

Fig. 3: As the amplitude of tidal fluctuations varies widely around Antarctica, this figure may be more insightful if the error were normalized by the root-mean-square tidal height fluctuations at each tide gauge.

The reviewer makes a good point here. The key is to produce a figure that best demonstrates the performance of the model - with respect to what the model is designed to do. Modelled ice shelf melting scales approximately linearly with velocity, and thus relative errors of tidal height amplitude are most insightful.

We have replaced the original Figure with the following:

[Figure]

*Figure 3: Spatial distributions of tidal height accuracy. Relative amplitude differences between the model solution and Antarctic Tide Gauge records ([H_WAOM − H_ATG] / H_ATG ) are shown for the major tidal constituents (a) M2, (b) S2, (c) K1 and (d) O1. The colorbar has been truncated at the 95% quantile.*

The respective interpretation has been included at the end of the section (L237-240):

*"Figure 3 shows the relative differences in tidal height amplitude. WAOM systematically overestimates tidal strength of the semi-diurnal constituents in the Ross Sea with differences often exceeding 80 %. In contrast, diurnal tides are generally underestimated and deviations are more balanced around the coast. For the diurnal bands most stations feature differences below 35 %."*

As this bias is not reflected in the melt rates, these findings are not picked up in the discussion.

Finally, we have disregarded the original Figure 3 (absolute errors of complex amplitude differences) to the supplemental material as it only confirms biases, which are well known from barotropic tide models. We have modified the respective paragraph accordingly (L228-232):

*"**Table 2 summarizes the outcomes of the tidal height accuracy analysis.** The model has a combined RMS error of 20 cm, which is within the accuracy of 2D Antarctic tide models (assessed by King and Padman, 2005).  **Similar to these models, most of our** bias comes from  sites at the grounding line deep under the large ice shelves **(see Appendix Fig. D4)**."*

Fig. 4: This figure shows that the 4km and 1km runs are much more similar than the 10km run. Is this simply because the 10km run has been integrated independently of the others for much longer (the 4km run is branched off the 10km run at year 5, then the 2km run is branched off the 4km run in year 7, and analysis is performed in year 8), or due to changes in the resolution of shelf processes?

The impact of independent integration time does not matter when each of the model solutions has equilibrated. That is, if we would disregard our cascading spinup procedure and instead initiate the 2km model from the ECCO2 state and integrate it for 9 years, we would expect to derive the same state. Further, the remaining model drift in mean melting at 2 km resolution has an opposing sign to the one at 4 km resolution, showing that independent integration time does not govern our results.

No changes have been made to the text.

It would also be helpful to include a second x-axis, along the top of the figure, showing the actual model grid spacings.

We agree with the reviewer and have included this detail. Here is the updated Figure 4:

[Figure]

Fig. 8: These plots are difficult to compare because the data are presented as a straightforward scatter, causing many points to overlap one another and thus obscuring many details. I would recommend presenting these T/S diagrams as two-dimensional probability density functions instead. Information about the depths of water masses could be conveyed in separate panels via volumetric weighting of the depths contributing to each water mass bin.

We believe the reviewer's understanding about our visualization method is incomplete, despite an comprehensive and well placed description. Our TS diagrams are not scatter plots. We plot a grid of volume weighted averages with transparent grid cells where bins are empty. It might appear as a scatter with overlap, but, actually, no points overlap and no detail is lost. This kind of visualization has been used before and using it here allows for direct comparison (e.g. against MetROMS and FESOM by Naughten et al., 2018).

No changes have been made to the Figure. The method to produce these plots is outlined in detail in the caption of the figure and, hence, no changes have been made to the text.

Also, the "ISW" label doesn't seem to be indicating any water masses in the left-hand panel.

We now have included a double headed arrow to indicate that the ISW label refers to all water masses below the freezing point (dashed line). Despite more ISW in subfigure b, the ISW label is presented in subfigure a to have all labels in one place.

[Figure]

Also, "AntarcticSurface" -> "Antarctic Surface" in the caption.
We have corrected this mistake in Figure 8 and Figure D2.

L304-305: Aren't the waters below the surface freezing temperatures ice shelf waters?
The reviewer is correct. We have modified the text from (L317-320):

*"Ice Shelf Water (ISW) outside the cavities is only apparent in WAOM."* to:

*"Ice Shelf Water (ISW) in WOA18 remains within 0.25 degC below freezing and is apparent over a wide range of salinities (33.75 to 34.75). In contrast, WAOM features ISW with temperatures of more than 0.5 degC below freezing, but a narrower range of salinities of 34.25 to 34.6."*

This detail has not been picked up later and no further changes to the manuscript are necessary.

L307: Formatting of citation.
We have corrected this mistake (L322 f.).

L311: Format "degC"
We have corrected this (L326).

L325: "temperature salinity transects"
We have corrected this to (L340): *"temperature-salinity transects"*

L327: "hydrology" -> "hydrography"?

The reviewer is right, we did mean hydrography. The term hydrology also includes water on the land surface, while hydrography only refers to ocean water. We have corrected this mistake (L342).

Fig. 9: The color bars for the middle panels are mis-labeled, I think.
The reviewer is right. These are absolute values, no differences. We have corrected this mistake in the label.

Figs. 11-13: The biases in WAOM look to be quite severe: comparable to the variations in T and S in WOA. I suggest adding panels showing the T and S sections in WAOM to better visualize this.
We agree with the reviewer and have included the respective plots. To follow the reviewers' intent in emphasizing the bias, we show WAOM results on the same color scale as WOA transects, rather than desaturating the new plots. We also have modified the captions accordingly. Here are the new Figures and captions:

[Figure]

*Figure 11. Temperature and Salinity transect on the Ross Sea continental shelf (175 °E) compared against observations. (a) and (d) are WOA18 2005-2017 summer mean temperature and salinity,* **(b) and (e) are 2007 summer mean temperature and salinity from our model (WAOM),** *and (b) and (d) are the perspective differences*  *(WAOM - WOA18).*  *WAOM's data has been interpolated to the WOA18 grid using nearest neighbours* **(for b and e in the horizontal; for c and f in the horizontal and vertical).**

[Figure]

*Figure 12. As Fig. 11, but for a transect across the Amundsen Seas along 107 ∘W.*

[Figure]

*Figure 13. As Fig. 11, but for a transect in Prydz Bay (Davis Sea continental shelf) along 70 ∘E.*

No more changes to the text were necessary, since the additional figures better visualize already discussed biases.

Fig. 15: Same comment as for Fig. 8.
We kindly refer to the same answer as given for the comment for Figure 8 above.

L367: "accurate" -> "accurately"
We have corrected this mistake (L383).

L384: citation formatting
We have corrected this mistake (L400).

L473-475: Has this issue been discussed earlier in the main text. I see it noted in Fig. D4, but not elsewhere.

The point the reviewer is referring to was discussed under L491-493, that we repeat here:

*"In third place, the boundary effects in the Eastern Ross Sea should be addressed. Introducing a sponge layer is difficult, since tides are also forced at the open boundary. Instead we recommend an adjustment to the model boundary locations to avoid intersection with ACC jets at shallow angles."*

This issue has been introduced earlier in the discussion (L433-437):

*"We also have reported CDW intrusions onto the continental shelf of the Eastern Ross Sea (see Fig. 9) and this is likely related to boundary effects. Where ACC jets cross the domain's boundary in shallow angles, artificial currents can arise. We have reduced these effects by making the boundary conditions outflow dominant (see Methods), but some artificial currents remain in the Ross Sea (see Fig. D3). We hypothesise that these currents drive CDW onto the shelf by affecting the slope of the isopycnals close to the shelf break."*

No changes have been made to the text.

**References**

Dinniman, Michael S., Xylar Asay-Davis, Benjamin Galton-Fenzi, Paul Holland, Adrian Jenkins, and Ralph Timmermann. 2016. "Modeling Ice Shelf/Ocean Interaction in Antarctica: A Review." *Oceanography* 29 (4): 144–53. https://doi.org/10.5670/oceanog.2016.106.

Dinniman, Michael S., John M. Klinck, Le-Sheng Bai, David H. Bromwich, Keith M. Hines, and David M. Holland. 2015. "The Effect of Atmospheric Forcing Resolution on Delivery of Ocean Heat to the Antarctic Floating Ice Shelves." *Journal of Climate* 28 (15): 6067–85. https://doi.org/10.1175/JCLI-D-14-00374.1.

Griffies, Stephen M., Gokhan Danabasoglu, Paul J. Durack, Alistair J. Adcroft, V. Balaji, Claus W. Böning, Eric P. Chassignet, et al. 2016. "OMIP Contribution to CMIP6: Experimental and Diagnostic Protocol for the Physical Component of the Ocean Model Intercomparison Project." *Geoscientific Model Development* 9 (9): 3231–96. https://doi.org/10.5194/gmd-9-3231-2016.

Kusahara, Kazuya, and Hiroyasu Hasumi. 2013. "Modeling Antarctic Ice Shelf Responses to Future Climate Changes and Impacts on the Ocean." *Journal of Geophysical Research: Oceans* 118 (5): 2454–75. https://doi.org/10.1002/jgrc.20166.

Mazloff, Matthew R., Patrick Heimbach, and Carl Wunsch. 2010. "An Eddy-Permitting Southern Ocean State Estimate." *Journal of Physical Oceanography* 40 (5): 880–99. https://doi.org/10.1175/2009JPO4236.1.

Nakayama, Y., D. Menemenlis, M. Schodlok, and E. Rignot. 2017. "Amundsen and Bellingshausen Seas Simulation with Optimized Ocean, Sea Ice, and Thermodynamic Ice Shelf Model Parameters." *Journal of Geophysical Research: Oceans* 122 (8): 6180–95. https://doi.org/10.1002/2016JC012538.

Naughten, Kaitlin A., Katrin J. Meissner, Benjamin K. Galton-Fenzi, Matthew H. England, Ralph Timmermann, Hartmut H. Hellmer, Tore Hattermann, and Jens B. Debernard. 2018. "Intercomparison of Antarctic Ice-Shelf, Ocean, and Sea-Ice Interactions Simulated by MetROMS-Iceshelf and FESOM 1.4." *Geoscientific Model Development* 11 (4): 1257–92. https://doi.org/10.5194/gmd-11-1257-2018.